# Processes contributing to cloud dissipation and formation events on the North Slope of Alaska

Joseph Sedlar[1,2], Adele Igel[3], Hagen Telg[1,2]

[1]Cooperative Institute for Research in Environmental Sciences, University of Colorado Boulder, Boulder, CO, USA
[2]NOAA Global Monitoring Laboratory, Boulder, CO, USA
[3]University of California Davis,

*Correspondence to*: Joseph Sedlar (joseph.sedlar@colorado.edu)

**Abstract.** Clear sky periods across the high latitudes have profound impacts on the surface energy budget and lower atmospheric stratification, however an understanding of the atmospheric processes leading to low-level cloud dissipation and
formation events is relatively limited. A method to identify clear periods at Utqiagvik (formerly Barrow), Alaska, during a five-year period (2014-2018) is developed. A suite of remote sensing and in situ instrumentation from the high-latitude observatory are analysed; we focus on comparing and contrasting atmospheric properties during low-level (below 2 km) cloud dissipation and formation events to understand the processes controlling clear sky periods. Vertical profiles of lidar backscatter suggest that aerosol presence across the lower atmosphere is relatively invariant around the clear period
bookends, which suggests that a sparsity of aerosol is not frequently a cause for cloud dissipation on the North Slope of Alaska. Further, meteorological analysis indicates two active processes ongoing that appear to support the formation of low clouds after a clear sky period: namely, horizontal advection which was dominant in winter and early spring, and quiescent air mass modification which was dominant in the summer. During summer, the dominant mode of cloud formation is a low cloud or fog layer developing near the surface. This low cloud formation is driven largely by air mass modification under
relatively quiescent synoptic conditions. Near-surface aerosol particles concentrations changed by a factor around summer cloud formation events. Thermodynamic adjustment and increased aerosol presence under quiescent atmospheric conditions are hypothesized as an important mechanism for fog formation.

## 1 Introduction

Over the Arctic clouds are ubiquitous (e.g., Herman and Goody, 1976; Curry et al., 1996). Studies of cloud occurrence from satellite report large cloud fractions over the full annual cycle (Wang and Key, 2005; Kay et al., 2016). Detailed observations of the vertical structure of Arctic clouds from remote sensing "supersites" document the frequent presence of lower tropospheric clouds (e.g., Shupe et al., 2011). These clouds frequently contain both water and ice particles, known as mixed-phase clouds, which can persist for hours to days in a near homogeneous state (Shupe, 2011). Liquid-bearing clouds have
been observed at temperatures as cold as -34 °C (Intrieri et al., 2002), but liquid is most common during the warmer, summer

months (Shupe et al., 2011). Clouds strongly modulate the incoming and outgoing radiative fluxes; over sea ice, longwave radiation dominates the radiative energy budget at the surface (Walsh and Chapman, 1998; Shupe and Intrieri, 2004; Sedlar et al., 2011).

While clear sky periods are relatively rare, their impact on surface radiation and thermodynamic structure are also considerable. So-called radiative states are dominant features of the Arctic atmosphere, alternating between radiatively clear and radiatively opaque states (Stramler et al., 2011). The Arctic atmosphere is relatively dry and cold, limiting the atmospheric greenhouse effect when clouds are absent. The surface longwave warming associated with Arctic clouds is a crucial component of the surface energy budget in the Arctic (Walsh and Chapman, 1998; Shupe and Intrieri, 2004; Sedlar et

al., 2011). Under cloud free conditions with low solar elevations, effective infrared cooling from the surface results in near-surface temperatures to drop (Pinto et al., 1997). As a result, strong surface-based temperature inversions commonly form (Kahl, 1990), and the turbulent mixing in the surface layer is inhibited. The Arctic boundary layer tends to remain relatively shallow following the lack of buoyant mixing because stratocumulus cloud-top generated turbulence is absent during clear skies (Shupe et al., 2008; Shupe et al., 2013; Sedlar and Shupe, 2014; Sotiropoulou et al., 2014; Brooks et al., 2017;

Tjernström et al., 2019). Over sea ice, when skies are clear, the surface energy deficit can lead to anomalies in the ice growth or melt depending upon season (Sedlar and Devasthale, 2012). Global climate models that fail to properly represent the seasonal occurrence of low-level Arctic stratocumulus also fail to match the observed relationships amongst net surface radiative forcing and static stability (Pithan et al., 2014).

In the Arctic, the transition between the radiative states is controlled by the evolution of lower tropospheric clouds (Morrison et al., 2012). As such, there is great interest in understanding the processes and mechanisms crucial to the formation, maintenance, and dissipation of lower tropospheric Arctic clouds. Their persistence seems counterintuitive since mixed-phase clouds are microphysically unstable (Morrison et al. 2012). Few studies have examined the processes active during dissipation and formation of these clouds.

Atmospheric conditions have a critical role in supporting the formation of lower troposphere Arctic clouds. Surface longwave cooling, relative lack of humidity, and subsequently the stratification are important processes contributing to the transformation of an air mass towards saturation (e.g., Wexler, 1936; Curry et al., 1996). The presence of a cold but emissive ice and snow-covered surface, especially over the central Arctic pack ice, provides an additional constraint on the air mass

transformation process (cf. Herman and Goody, 1976; Tjernström et al., 2015; 2019). Further, the Arctic atmosphere is not stationary; synoptic forcing, changes in the free tropospheric subsidence strength, frontal passages, and storms are observed during all seasons (e.g., Stramler et al., 2011; Sotiropoulou et al., 2016; Persson et al., 2017; Vessey et al., 2020). Such active disturbances may provide the forcing needed to transition from clear sky to cloudy, or vice versa (Kalesse et al., 2016). Large eddy simulations have shown that mixed phase cloud lifecycle is very intricately connected to the free

tropospheric subsidence (e.g., Young et al., 2018), further highlighting the important role of synoptic forcing on cloud evolution.

While dynamic forcing likely controls most transitions between clear and cloudy states, an increasing body of work is pointing towards the possible role of aerosol particles in this process. Simulations of Arctic clouds consistently show that

enhanced ice nuclei (IN) or ice crystal concentrations can lead to mixed-phase cloud glaciation (Harrington et al. 1999; Jiang et al. 2000; Avramov and Harrington, 2010; Morrison et al., 2011), as ice precipitation acts a net sink of cloud mass (cf. Solomon et al., 2011; Forbes and Ahlgrimm, 2014). Using observations from the central Arctic sea ice, Mauritsen et al. (2011) reported on a cloud condensation nuclei (CCN) limited cloud regime; they found that pristine air with very small CCN concentrations actually inhibited the formation of cloud even under supersaturated conditions. Model simulations also

suggest that very low cloud condensation nuclei (CCN) or cloud droplet number concentrations (~10 cm$^{-3}$ or less) are an efficient mechanism to transition the cloud lifecycle, initiating cloud dissipation (Birch et al., 2012; Hines and Bromwich, 2017; Loewe et al., 2017; Stevens et al., 2018). Based on observations from the North Slope of Alaska (NSA) and complementary simulations, Silber et al. (2020) found that clouds forming under low aerosol concentration regimes are incapable of producing the cloud-top turbulence necessary to maintain cloud persistence. The results from these studies

suggest microphysical changes, such as local increases in IN or decreases in CCN, may be responsible for the dissipation of Arctic mixed-phase clouds.

A detailed analysis of one observed Arctic cloud dissipation event suggested an array of complex processes contributed to the cloud decay (Kalesse et al., 2016). Observed changes in aerosol number and scattering properties were found to be

associated with a large-scale change in air mass that advected through the NSA region. Their case study revealed how transient atmospheric dynamics were responsible for changing the thermodynamic structure, coinciding with a response in the cloud microphysical properties. They suggest that the interaction of aerosol-modified cloud microphysical properties with dynamic and thermodynamic processes could be important for driving dissipation. The results of Kalesse et al. (2016) are far from the steady state, idealized modelling studies that typically focus on how changes in aerosol or cloud particle

concentrations impact cloud lifecycle and suggest the need for further investigation of the impact of aerosol on the cloud lifecycle in the Arctic.

Missing from case studies of cloud dissipation or formation events, such as in Kalesse et al. (2016), is a climatological understanding of the causes and physical processes responsible for the dissipation or formation of low-level Arctic clouds. In

this paper, we examine the characteristics of clear-cloudy sky transitions, or vice versa, in the Arctic. More specifically, we assess whether the aerosol and the general meteorological variability provide clues to the processes that are important for lower troposphere, below 2 km, cloud dissipation and cloud formation events. By comparing and contrasting the variability of such properties around cloud dissipation (start of clear period) and around cloud formation (end of clear period) events,

we aim to learn how changes in aerosol number, aerosol vertical structure, and atmospheric thermodynamics contribute to
formation and cessation of clear sky periods in the Arctic. Measurements and retrievals from a range of in situ and remote sensing instruments during the course of a five-year period from 2014 to 2018 on the NSA at Utqiagvik (formerly Barrow) are utilized. Rather than exploit individual cases, we assess the role of aerosol, synoptic variability and near-surface meteorology using a statistical approach for these prolonged clear sky periods over these five years.

## 2 Instruments at Utqiagvik

The observatory at Utqiagvik is an ideal location for understanding the contribution of meteorological and aerosol processes to Arctic cloud dissipation and formation. Generally, cloud fractions are high, typically between 60 and 95%, and lower tropospheric clouds were common, especially during sunlit months (Shupe et al., 2011; Sedlar, 2014). Having a relatively large cloud occurrence makes the NSA a viable location to further study the process that actually led to the formation or
cessation of an infrequent clear sky period. Utqiagvik is at a coastal site, located within 2 km of the coast line along the NSA. Seasonal climatologies of the back-trajectory footprint of air masses reaching the observatory were predominantly from the high Arctic Ocean, and to a lesser extent from the continent to the south (Freud et al., 2017). Pollution from the oil fields around Prudhoe Bay did not regularly lead to changes in background aerosol or cloud microphysical properties at Utqiagvik (Maahn et al., 2017). However, wildfires may sporadically influence the background aerosol concentrations and
chemical composition across the NSA during active fire seasons (Creamean et al., 2018).

The Vaisala CL31 ceilometer is an operationally robust instrument measuring the vertical profile of backscattered light due to aerosol and cloud particles (Ravila and Räsänen, 2004). The lidar instrument operates fully automatically by emitting a pulsed laser with a wavelength of 910 nm. The backscattered signal is processed by onboard software, producing retrievals
of cloud presence and the vertical level of up to 3 cloud base heights. When the signal is attenuated but a cloud base height could not be retrieved, the retrieval software assumes the obscuration in the backscatter is due to a surface-based cloud or fog layer and therefore reports the vertical visibility.

The high spectral resolution lidar (HRSL, Eloranta, 2005) was designed to separate the molecular scattering signal from the
geophysical (aerosol, cloud) scattering signals at the 532 nm laser wavelength. As a result, vertical profiles of aerosol and cloud hydrometeor backscatter are robustly characterized by the instrument retrieval software. The profiles were available from the first valid range gate, approximately 101 m AGL, through the full troposphere. Profiles of particulate (aerosol + cloud) backscatter and depolarization ratio are used to aid in identification of clear sky profiles, in addition to examining the vertical distribution of aerosol during clear sky periods. Because the HSRL operates in the visible light portion of the
spectrum, the signal typically becomes attenuated once the cloud optical thickness reaches ~ 3 to 4. Considering this

limitation, HSRL backscatter is only analyzed during periods determined to be completely cloud free by analyzing all available active remote sensing measurements.

Vertical distributions of cloud layers were derived from the zenith-viewing Ka-Band (KAZR) cloud radar (e.g., Moran et al., 1998). The KAZR measures the spectra of backscattered power (reflectivity) as a function of Doppler velocity of the cloud and precipitation particles in the atmospheric column above the radar. The millimeter wavelength (35 GHz) provides high sensitivity and signal to noise ratio allowing the radar to observe cloud droplets, although some concentrations may be missed (de Boer et al., 2009). The ARSCL (Active Remote Sensing of Cloud Layers) algorithm (Kollias et al., 2016) yields processed KAZR cloud property retrievals based on best-estimate radar moments including reflectivity, Doppler velocity, and spectrum width. Cloud top height and signal to noise ratios from the ARSCL data products are examined here. While the KAZR is capable of observing concentrations of small droplets, its measurement is sensitive volume squared and therefore the signal may be attenuated in by the presence of ice crystals which are typically larger than droplets (e.g., de Boer et al., 2009).

At the surface, a TSI 3010 condensation particle counter (CPC) measures the concentration of particles ranging in diameter from 10-3000 nm present within a volume of air. Air is continuously pumped through the instrument, where it is supersaturated with n-butyl alcohol which results in condensational growth of individual particles. The grown particles scatter sufficient light so they can be detected and counted by an optical particle detector. Additionally, a TSI nephelometer measures the total light scattering of aerosol particles. From this, particle scattering coefficients are computed.

Near-surface measurements of air temperature, relative humidity, and wind speed and direction were observed from a weather station deployed on the NSA. Downwelling and upwelling longwave radiation measurements were made from upward- and downward-viewing Eppley Precision Infrared Pyrgeometers. These instruments have factory stated uncertainties of about 2-5%. The longwave fluxes are further scrutinized using the Radiative Flux (e.g., Long and Turner, 2008) processing retrievals.

Atmospheric profiles of thermodynamics and winds were made by radiosoundings launched from the NSA. Radiosoundings were launched nominally every 12 hours, although intermittent periods exist when the frequency was either higher or lower. The measurements of temperature, specific humidity, and pressure were used to compute profiles of equivalent potential temperature.

**3 Methods**

Measurements from the instrumentation described above are analysed to characterize and better understand radiative, aerosol, and thermodynamic characteristics of clear sky periods on the NSA during all seasons from 2014 through 2018. The
identification of clear sky periods is first and foremost dependent upon continuous measurements from the ceilometer. Periods of continuous ceilometer detection status equal to zero (zenith-viewing clear sky) were earmarked as potentially clear. To avoid broken cloudiness being classified as clear periods, clear sky periods were required to be at least 2 hours in duration. If the 2-hr temporal requirement was met, the end of the clear period was determined as the time when the ceilometer once again detected cloud overhead and the cloud persisted for at least 2 consecutive hours. The start and end
times of the clear periods meeting these criteria were logged. Clear periods were scrutinized further by ensuring at least 96% of the ceilometer detection status during the clear period were actually reported as clear sky. If intermittent cloudiness occurred and this condition was not met, the clear period was discarded from further analysis. Finally, clear periods are required to be bookended by clouds below 2 or 3 km (depending on the analysis below) or less in order to focus on the dissipation and formation of low-level clouds.


Start and end times of the clear periods were compiled monthly for further cloud screening. Measurements from the HSRL and cloud radar during identified clear periods were exploited to further modify start and end times of clear periods based on their sensitivities to cloud hydrometeors. Vertical profiles of HSRL backscatter and depolarization ratio during each clear period were scrutinized to modify start and end times if backscatter and depolarization ratio exceeded threshold values
typical of aerosol particles in the Arctic (Shupe, 2007). Signal to noise ratios and minimum detectable reflectivity flag indicators from the cloud radar were used to further remove times of intermittent cloud and/or precipitation signals during the clear periods. An identified clear period having a start or end time that transitioned between adjacent months was considered for analysis in both months.

An example clear sky period from 14 August 2014 is shown in Fig. 1. Prior to the start of the clear sky period, a low cloud with a base and top at 100 m and 400 m AGL, respectively, was present. The clear period began shortly before 04:00UTC and persisted for nearly 7.5 hours before intermittent, very low cloud signatures were observed by the cloud radar, HSRL, and ceilometer (Fig. 1a-b). The transition from cloudy to clear caused marked transitions in the net surface radiative fluxes, especially in the net longwave (LWN) which dropped by nearly 80 W m$^{-2}$ (Fig. 1c); a similar abrupt transition in LWN
occurred together with the low cloud formation shortly before 13:00UTC. The abrupt changes in LWN in Fig. 1c are representative examples of the radiative states governing the Arctic (Stramler et al., 2011; Morrison et al., 2012; Engström et al., 2014). The inset of Fig. 1c shows the equivalent potential temperature profiles from radiosoundings during the event at 05:30 (blue) and 13:15UTC (yellow), revealing changes in mixed layer depth depending upon whether or not the cloud was present; decreases in mixed layer depth are evident in the shallower layer depth of aerosol backscatter from the HSRL during

the clear period (Fig. 1b). Evolution of near-surface meteorology showed temperature increased and dew point temperature decreased following the dissipation in connection with a slight change in wind direction (Fig. 1d). During the clear period, winds and thermodynamics remained quasi-constant until cloud formation when wind direction slightly changed again. Likewise, near-surface particle concentrations exhibited the largest variability near the start and end of the clear period (Fig. 1e).


The following sections explore the statistical variability of aerosol and meteorology associated with clear sky periods on the NSA from 2014 through 2018.

## 4 Results

### 4.1 Clear sky periods

The number of periods meeting the clear-sky criteria are relatively few during the 2014-2018 period, owing to the vast persistence of Arctic cloudiness. Clear periods were most frequent during the dark winter and spring months, with as many as 25 individual periods during the five-year period. Increased cloudiness limited clear sky periods to as few as six during summer and autumn (Fig. 2) based on the definition of a clear sky period here. The annual distribution of monthly clear sky

frequency follows the annual trends of cloudiness in the high Arctic reported in the literature (Curry et al., 1996; Wang and Key, 2005; Shupe, 2011), where more clear periods are found during the seasons with relatively lower cloud fractions.

Figure 2 also shows the number of clear sky periods that ended due to a low cloud (magenta, cloud base below 400 m) or a fog (blue) formation event. A seasonal cycle is evident in the emergence of both low clouds and fog. These cloud formations

dominate after clear periods from spring through early autumn, occurring for approximately 60 to 90% of all cloud formation events during these seasons. Oppositely, few of the formation events during winter were connected to a fog or cloud with a low base height. Subsequent sections focus on these low cloud and fog forming cases in order to understand the processes supporting the formation of these very low clouds after a clear sky period.

### 4.2 Aerosol characteristics at clear period bookends

### 4.2.1 HSRL aerosol backscatter during clear periods

Lower tropospheric Arctic clouds require available aerosol to act as cloud condensation nuclei and ice nuclei. Statistical distributions of HSRL aerosol backscatter during clear periods are examined to determine whether the vertical structure of aerosol may provide clues to processes supporting dissipation or formation of clouds. Because lidar backscatter is largely

attenuated by cloud hydrometeors, HSRL backscatter profiles are only analyzed during periods determined to be completely cloud free using a combination of measurement streams from the lidar, ceilometer, and cloud radar.

The vertical structure of aerosol backscatter retrieved from the HSRL during all clear periods as a function of month is presented in a climatological fashion. From Fig. 3, it is found that aerosol backscatter has a very dynamic structure, with variability changing both vertically and temporally (Kafle and Coulter, 2013). A pronounced decrease in backscatter across a relatively shallow layer near the surface, ranging from 100 to 1000 m, is observed during all clear periods. The variability across the lower 1000 m is overlayed by a reduction in the backscatter gradient with height, marking the transition towards free troposphere background aerosol. The depth of the transition, as well as variability in its gradient with height, is intimately connected to season. For example, the summer and early autumn (g-k) mean backscatter decrease happens over a shallower layer above the surface and is more abrupt than during winter and spring (a-f). Many processes may contribute to the depth of an enhanced aerosol backscatter layer, including horizontal advection, long-range transport often largest during winter (Klonecki et al., 2003; Willis et al., 2018), and lower atmosphere stratification and boundary layer mixing; the large variability across the lowest kilometer is related to a combination of these characteristics.

To better connect vertical structure of aerosol backscatter to potential impacts on cloud dissipation and formation, backscatter profiles are normalized by mean cloud top height retrieved from the ARSCL processing of cloud radar profiling. Only cases where a mean cloud top below 2 km AGL 60-min before cloud dissipation and after cloud formation are examined; these cloud top heights are used to normalize backscatter profiles in a window 1-hr after (before) a cloud dissipation (formation) event. Relative frequency distribution (RFD) profiles of seasonal backscatter on the normalized vertical grid are presented (Fig. 4). If any cloud hydrometeor returns within the 1-hr period were sensed by the HSRL, KAZR or ceilometer, these times were flagged and removed from the subsequent analysis.

In the hour after cloud dissipation, aerosol backscatter shows a decrease with height from near the surface ($z_n$=0) to the prior cloud top level ($z_n$=1) (Fig. 4a-d); the median decreasing backscatter profile is less evident during spring (b) and summer (c) compared to winter (a). These profile shapes are similar to the decrease with height found for the full clear period profiles of backscatter for the winter and spring months (Fig. 3). Similar to summer, the backscatter profiles are less variable with height during autumn (d). The distributions also indicate individual cases with enhanced aerosol at and below the previous cloud top (c-d). Values of backscatter larger than $10^{-7}$ $m^{-1}$ $sr^{-1}$, a threshold value determined as pristine (Shupe, 2007), at all heights suggests that aerosol concentrations remained relatively large below the previous cloud level and especially across the lower atmosphere. Therefore, aerosol particles were available throughout the lower atmosphere even after the cloud dissipated (or perhaps because the cloud dissipated). These distributions suggest a lack of particles was not the likely cause for dissipation.

Preceding cloud formation, the backscatter distributions and median profiles below $z_n=0.5$ (e-h) are typically smaller than observed directly after cloud dissipation (a-d). Backscatter continues to decrease with height towards the newly formed cloud top level (e-h); the decrease is more evident during winter and spring (e-f), while the RFD in summer (g) has significantly less variability with height. As the cloud top height is approached, backscatter medians are similar between the hour after dissipation and hour before formation, for all seasons. There is no evidence of enhanced backscatter prior to formation, and the backscatter across the lower levels is often smaller than just after cloud dissipation. Considering cloud was observed shortly after, these features prior to formation do not show any evidence of enhanced aerosol transport into the lower atmosphere. Furthermore, if we assume the aerosol backscatter just after dissipation (a-d) was likely high enough to support clouds (e.g., Shupe, 2007), the smaller aerosol backscatter prior to cloud formation (e-h) was probably not small enough to inhibit cloud either.

Seasonal profile statistics of HSRL backscatter just after cloud dissipation and just before cloud formation for low cloud formation cases only (cloud base under 400 m) are further examined (Fig. 5). To connect the vertical variability in aerosol distribution with cloud formation, Fig. 5a-d shows median and interquartile ranges of backscatter normalized to low cloud top height, using the median cloud top height, that ended the clear period, over a 60 min window. Backscatter prior to cloud formation (blue) is largest below the cloud top ($z_n=1$), above which backscatter decreases rapidly with height for all seasons but winter (a). The decrease in backscatter with height reveals a relatively shallow boundary layer where the surface is the likely source of aerosol; in summer and autumn, this transition occurs over the first 300 m (g-h), and increases to near 600 in spring (f). A lack of variability in backscatter above cloud top suggests the upcoming cloud layer may depend upon aerosol within the boundary layer, as aerosol backscatter above cloud top level is limited. Not including winter, backscatter profiles through the layer where low cloud eventually forms (blue) are generally similar, or slightly smaller, than backscatter just after cloud dissipates (black) (Fig. 5b-c). It is therefore unlikely that plumes of increased aerosol were advected into the shallow boundary layer to support subsequent low cloud formation. The situation during winter differs (a, e); backscatter variability is slightly larger below cloud top prior to formation than after dissipation (black/gray). Above the low cloud height, backscatter is larger after dissipation and is concentrated within a layer between 400-800 m AGL (e). Elevated backscatter shortly after dissipation is modestly larger than prior to cloud formation. The magnitude and variability in the median profiles above and below $z_n=1$ suggest vertical transport, such as subsidence, may have resulted in increased aerosol and supported the low cloud formation (a).

**4.2.2 Near-surface aerosol concentrations and clear period boundaries**

Variability in near-surface particle concentrations around the start and end times of clear periods are investigated to complement the lidar analysis. Monthly median and interquartile ranges within 2-hr after cloud dissipation versus before

cloud dissipation (a-d) and within 2-hr of cloud formation (e-h) are shown in Fig. 6. In terms of concentrations measured shortly after and before cloud lifecycle changes, numbers infrequently drop below 100 cm$^{-3}$. There is seasonality evident,

where more particles were observed in summer and early autumn than during winter and spring, broadly in agreement with climatologies from the NSA (Quinn et al., 2002; Lubin et al., 2020); though February and March have obvious outliers with relatively large concentrations for specific events (a-b, e-f). Outside of these monthly outliers, particle concentrations during winter and spring were very similar on either side of the dissipation event (a-b). Concentrations after cloud dissipation tend to be larger than before the dissipation occurred, beginning in summer (c) and continuing through autumn (d). Median

increases after dissipation ranged from marginal to twice as large than before dissipation, and these medians were calculated from significantly different distributions following a Wilcoxon rank-sum significance test (no black marker edge). Having at least the same, or greater, number concentration after the clear period starts suggests that decreasing aerosol concentrations were not driving cloud dissipation.

Similarly, CPC concentrations leading up to, and shortly after, cloud formation (end of clear period) are shown in Fig. 6e-h. Here, only cases when the emerging cloud layer was identified as a low cloud with cloud base below 400 m (circles) or surface fog (squares) are considered; this distinction is an effort to constrain the vertical footprint of the near-surface CPC measurements. Despite some monthly outliers, median particle concentrations were generally similar during the pre-formation and post-formation periods in winter and spring (e-f), even though significance testing indicates significantly

different distributions for the majority of cases within these seasons; clouds that form as fog layers reveal no distinct differences in particle concentrations to low clouds with slightly elevated base heights. By summer, the concentrations have shifted, and medians were frequently twice as large before formation compared to after formation (g). In connection with an increase in the number of fog cases during summer, concentrations associated with fog are further away from the 1:1 line than some of the low clouds. Autumn concentration differences between periods highlight a season in transition (h), shifting

between the enhanced concentrations prior to formation in summer, and the similar concentrations around formation during winter. Increased concentrations connected with new particle formation events have been identified as an important mechanism contributing to numerous, but smaller size, near-surface particle concentrations on the NSA (e.g., Freud et al., 2017).

Distributions for the 550 nm scattering coefficient from the nephelometer (Fig. 7) indicate a general reduction in particle scattering from winter and spring to summer, especially in the hours around cloud formation (Fig. 7e-g). Generally, the scattering coefficient is proportional to the particle size. As a result, particles in the ultra-fine mode typically have a negligible contribution to the scattering coefficient of aerosols in all but the most extreme circumstances (e.g., Telg et al., 2017). Despite the seasonal decline in scattering, prior to summer cloud formation, the scattering was frequently larger than

after cloud formation (Fig. 7g); this is especially true for fog formation events during July and August. Analysis of the Ångström exponent in summer revealed distributions where the exponent was smaller prior to formation and generally larger

after formation (not shown). The Ångström exponent is inversely proportional to particle size. Coupled with the generally larger scattering coefficient, the more numerous particles observed prior to summer formation indicate that changes in particle numbers are not limited to Aitken model particles but include larger particle sizes that can activate. New particle formation events during clear periods may have occurred and evolved during the clear sky period. These results suggest a connection between an abundant presence of aerosol particles and potentially these particles serving as the sources of CCN activation in the lower atmosphere. The fact that particle concentrations drop after cloud formation, in some cases by over a factor of two (see summer medians in Fig. 6g), supports the mechanism of a conversion of a fraction of these particles to low cloud/fog droplets.

## 4.3 Meteorology and its relationship to clear periods

The previous analyses did not identify major changes in the vertical distribution or surface concentration of aerosols surrounding cloud dissipation and formation events; increased surface particle concentrations before low cloud formation compared to after during summer were the most significant change. The results imply that cloud-free periods may not be driven by significant changes in aerosol presence alone, consistent with conclusions drawn from an Arctic dissipation case examined in detail (Kalesse et al., 2016). Here we investigate meteorological processes to understand their role in driving cloud dissipation and formation, as well as their role in modulating surface aerosol concentrations. In this section, emphasis is placed on understanding the processes supporting low cloud (base below 400 m) and fog formation as these are the dominant cloud types emergent after clear sky periods during much of the year (Fig. 2).

### 4.3.1 Clear skies, cloudy skies and lower tropospheric stability

Arctic stratocumulus clouds exert a critical influence on the static stability near the surface, where these clouds often modulate the stratification due to cloud top radiative cooling and induced turbulence (Shupe et al., 2008, 2013; Sedlar, 2014; Sedlar and Shupe, 2014; Brooks et al., 2017). A metric to explore the influence of clouds on stratification is through the relationship between lower tropospheric stability (LTS) and net longwave (LWN) radiation. This parametric relationship has the potential to identify coupled modes in the observations since LWN is primarily proportional to cloud infrared emissivity (which asymptotes at liquid water paths between 30-50 g m$^{-2}$ (e.g., Shupe and Intrieri, 2004)) and the effective temperature difference between the cloud (or clear sky) and surface. The difference in equivalent potential temperature between the surface and 950 hPa pressure level provides a value on the static stability of the lower troposphere (Sedlar et al., 2020). The 950 hPa level is generally around 500 m AGL in the Arctic, which frequently encompasses all, or a fraction of, the Arctic atmospheric boundary layer and the sub-cloud mixed layer (Shupe et al., 2013; Sedlar and Shupe, 2014).

The strong dependence of LWN on the presence or absence of clouds (see Fig. 1c), and the strong linkage between cloud and stratification (LTS) is evident in the seasonal frequency distributions of Fig. 8a-d. The dominant peak in the seasonal

distributions occurs for LWN near -10 Wm$^{-2}$ and LTS ranging from 0 to 2.5 K, corresponding to near-neutral to slightly
stable stratification; this mode represents the canonical overcast Arctic with cloud-generated turbulence producing mixing
across the boundary layer and sub-cloud mixing layer (Sedlar et al., 2020).

The red symbols correspond to instances in the LWN-LTS parameter space when a radiosounding was launched during a
clear period. These symbols correspond to a far less frequently occurring distribution mode occurring under clear skies, with
larger (< -40 Wm$^{-2}$) LWN deficits and correspondingly greater positive LTS. The surface is cooling effectively to space, and
together with the lack of mixing from the absence of low-level liquid-bearing clouds, an enhanced stable stratification is
maintained across the lower troposphere. Differences in the magnitudes of both the clear-sky LWN and LTS modes by
season are connected to thermodynamic constraints dependent upon annual cycle. For example, LWN deficits are
considerably larger in summer than winter and spring because the land surface at Utqiagvik emits infrared radiation at a
much higher temperature. Positive LTS for clear-sky conditions are smaller in magnitude during summer than winter and
spring because shortwave radiation represents a strong surface energy forcing, dependent upon surface albedo and solar
elevation.

RFDs describing the relationship between surface condensation particle counts (CPC) per LTS are shown in Fig. 8e-h. CPC
distributions for winter and spring (e-f) are invariant to the stratification, indicating that near-surface aerosol numbers are
largely independent of sky condition (clear or cloudy). The spread in CPC concentrations increases during summer and
autumn, where an order of magnitude span in the distributions are observed (g-h). During summer and autumn, it is evident
that CPC concentrations were consistently larger during clear sky periods (red symbols) than during cloudy conditions
(concentrations corresponding to the peak mode in the RFD with LTS < 2.5 K). These seasonal and sky condition
differences in particle concentrations suggest different processes are responsible for aerosol numbers near the surface, such
as the potential for new particle formation events during summer (Freud et al., 2017).

### 4.3.2 Meteorological contributions to cloud formation

To examine the potential role of near-surface air mass modification in supporting cloud formation (e.g., Tjernström et al.,
2015), the seasonal relationship between 2-hr tendencies in near-surface air temperature and relative humidity are examined
(Fig. 9). Following mean air temperatures (e.g., Korolev and Isaac, 2006) during these individual clear periods, relative
humidity (RH) trends are calculated with respect to ice (RHI) for November through May, and with respect to liquid June
through October.

The covariability between temperature and RH reveals distinct seasonal differences, owing to different processes impacting
the evolution of near-surface thermodynamics during the final 2 hours of the clear periods. Temperature tendencies during
winter (a) were both positive and negative, and changes to RHI were frequently below ±2% hr$^{-1}$. Having temperature

changes of both sign together with little change to RHI indicates that air mass modification, primarily through surface longwave emission, is not a dominating process; this is especially true for the cases with a positive temperature tendency.

During clear periods, the atmosphere is largely transparent to longwave radiation emitted from the surface, and the lack of clouds to re-emit longwave back to the surface would cause a drop in temperature (Fig. 1c-d). If air mass modification through quiescent cooling was the only process occurring, relative humidity would have a positive trend. Instead modest humidity changes coinciding with temperature changes suggests thermodynamic advection may be playing a larger role in the transition from clear to cloudy. In spring, negative temperature tendencies were more common than positive tendencies

(b); decreasing temperatures were almost exclusively connected with increasing RHI, leading to an increase in R-values compared to winter. The majority of low cloud formation cases (red circles) group into this regime, suggestive of cooling and moistening through quiescent air mass transformation. While the majority of fog formation events (magenta squares) also group in this regime, there are a handful of fog cases connected with positive temperature trends and variable changes in RHI leading to a correlation of R=0.45.


While winter, and to a lesser extent spring, revealed thermodynamic changes likely resulting from air mass changes through advection, summer tendencies reveal a distinguished negatively-sloped correlation (Fig. 9c). Nearly all low cloud (red circles) and fog (magenta squares) formation events were observed under cooling and increasing RH trends. A statistically significant R=0.91 for fog events during summer was found. This relationship is consistent with quiescent longwave cooling

leading to an increase in RH near the surface, subsequently conditioning for the formation of a fog. Transitioning to autumn, relative humidity tendencies returned to relatively small values hovering around zero, while temperature trends were slightly negative for fog cases and slightly positive for low cloud cases (d). Despite changes in the temperature, little change to the humidity suggests that thermodynamic advection may be a more influential process than quiescent air mass transformation during autumn.


The variability in near-surface wind direction and wind speed at the start and end of the clear periods as a function of season is shown in Fig. 10. Analysis is restricted to only clear periods that were followed by the formation of a low cloud or fog layer. From spring through autumn (b-d), wind direction distributions within a 1-hr period just after dissipation (solid blue) and 1-hr just prior to cloud formation (solid red) indicate little change in the air mass origination near the surface. A

dominant east-northeast wind prevailed through summer during these clear periods, while autumn winds were influenced by an enhanced southerly component. Spring and summer near-surface winds predominantly have an ocean footprint, which is likely influenced by sea ice cover during spring and more open water during summer. Wind direction variability was considerably larger during winter between the beginning and end of the clear periods (Fig. 10a); large wind shifts in winter are representative of synoptic scale variability and frontal passages. Included are the wind direction RFDs for the 1-hr prior

(dotted blue) and 1-hr post fog formation (dotted green). These wind direction distributions are very similar, especially for spring through autumn (b-d).

Wind speed distributions (insets in Fig. 10) were relatively consistent between the start and end of clear periods in terms of the peak wind speeds. Relatively constant wind direction and wind speed at the start and end of the clear periods further support the finding of persistent flow during spring and summer. During spring and summer, wind speed RFDs for the fog formation events are shifted slightly towards slower wind speeds compared to all low cloud cases (dotted lines in insets of b-c); the slower speeds lend support to relatively calm conditions supporting to fog formation. A lack of wind variability in spring and summer indicates more persistent flow patterns for the duration of the clear periods. This suggests that large-scale synoptic fronts are not likely the driving force for cloud dissipation and subsequent cloud formation during these seasons.

Despite relative consistency in near-surface winds during clear periods, larger-scale atmospheric dynamics may be the mechanism governing cloud dissipation and formation events (Kalesse et al., 2016). To determine the presence and strength of large-scale advective forcing, tendencies in geopotential thickness between two atmospheric pressure layers before cloud dissipation and before cloud formation are analysed. Geopotential thickness between pressure levels is proportional to the mean temperature and mean moisture content of the layer, and therefore are indicators of change in layer temperature, moisture, or both.

Theoretically, geopotential tendency is related to both vorticity advection and geopotential advection (resulting from thermal advection) through quasi-geostrophy (e.g., Holton, 1992). In practice, we can estimate the general vertical structure of geopotential by computing the geopotential thickness profiles at Utqiagvik for two atmospheric layers: 1) the 500-700 hPa layer, and 2) the adjacent 700-850 hPa layer. Comparing the tendencies in these two layers are then useful for identifying differential thermodynamic advection which can be linked to the instability and the vertical coherency of dynamic forcing. Layer geopotential thickness tendencies were originally computed using consecutive radiosounding profiles from the NSA nearest to a cloud dissipation or formation event. However, radiosoundings are released only nominally every 12 hours at Utqiagvik, and therefore the temporal connection to clear sky changes was under sampled. To analyze thickness tendencies on an increased temporal frequency, we use hourly geopotential height profiles from the European Centre for Medium Range Weather Forecasts ERA5 reanalysis (Hersbach et al., 2020) nearest to in time to the cloud dissipation event and compute the change in geopotential thickness from 4 hours prior to this time (m hr$^{-1}$). In a similar fashion, thickness tendencies are computed from geopotential height profiles for 4 hours prior cloud formation time (end of clear period). The use of geopotential height profiles from reanalysis allowed the ability to compute the 4-hr consecutive layer tendencies for each season over the 5-year period. From this, the seasonal mean and standard deviation in layer geopotential tendency could be computed. The seasonal variability is used to identify the strength of thickness tendencies associated cloud dissipation and formation events relative to a seasonal climatology.

Figure 11 shows the seasonal relationships between 500-700 hPa layer and 700-850 hPa layer geopotential tendencies for all cloud dissipation (a-d) and cloud formation (e-h) events; formation events for low cloud formation (red circles) and fog formation (magenta squares) are again differentiated. The relationship between layer tendencies follows a positive slope for all seasons, but with variable linear regressions and associated correlation coefficients. Geopotential tendencies having the same sign are representative of barotropic-like thickness increases/decreases across the lower- to mid-troposphere (Holton,

1992); in these instances, thermal advection influences the two atmospheric layers in a similar manner. Hence, some degree of larger-scale synoptic forcing is present, but to varying magnitudes, which will impact the local thermodynamics within a 4-hr period prior to cloud shift events. Larger tendencies are observed during winter and spring than during summer, prior to both dissipation and formation events. For winter and spring, approximately 58% of all dissipation events were within the range of seasonal variability (dashed blue lines), and even fewer, 38%, for autumn (d). During summer the tendencies were

frequently (74%) within the range seasonal variability for dissipation (c) and formation (g) events. Prior to springtime cloud dissipation (b), a number of events are clustered near the origin like in summer. This clustering reveals a mode of tendencies associated with weaker synoptic forcing in connection with cloud dissipation.

With cloud formation (Fig. 11e-h), the type of forming cloud (elevated, low or fog) varied with season and synoptic setting.

Winter low cloud formations (e, red circles) were associated with relatively small thickness tendencies while tendencies for fog formation (magenta squares) were scattered and large. In contrast, spring and summer (b-c) fog formation events were associated with relatively weak geopotential tendencies which clustered around the origin. During summer, correlation coefficients dropped to 0.11 for fog formation events, indicating a near zero relationship between the thickness tendencies across the two layers. The low cloud formation events were frequently (approximately 75%) observed within the bounds of

seasonal variability. Relatively weaker tendencies remained for cloud formation into autumn (h). The seasonal transition towards weak layer thickness tendencies across spring, summer and into autumn in connection with low and fog cloud formation is consistent with a reduced synoptic forcing as the primary cause for these specific cloud changes.

## 5 Discussion

Little changes in the vertical structure of aerosol from the HSRL after cloud dissipation and before cloud formation events indicates sharp variation or change in aerosol presence was not the predominant process controlling the cloud changes. Aerosol backscatter was always largest across the lower atmosphere near the surface, despite seasonal variability in the lower tropospheric stability. The complicated nature of boundary layer mixing processes in the Arctic due to a lack of ground-based convection and stable stratification further enforce this gradient structure in HSRL backscatter (Di Pierro et al.,

2013; Kafle and Coulter, 2013). HSRL backscatter during the clear periods was always above backscatter levels reported for

very pristine Arctic conditions (Shupe, 2007); this indicates enough aerosol were likely available to sustain cloud had the environmental conditions supported their presence. Near-surface particle concentrations before and after cloud dissipation events were very similar, providing further evidence that the absence of aerosol was not driving the fate of the cloud layer. The processes leading to cloud dissipation are different from Mauritsen et al. (2011) over the central Arctic sea ice, where very pristine air severely limited the number of particles available to become cloud condensation nuclei.

Near-surface meteorology, however, did show variation around cloud dissipation and formation events. In winter, wind direction changes between the start and end of a clear period were substantial. Likewise, the largest variability in the layer geopotential height tendencies was observed during winter and spring; these tendencies, however, subsided in magnitude in spring during the lead up to cloud formation. Furthermore, tendencies in winter near-surface temperature varied between warming and cooling trends ranging between 0.5-1°C hr$^{-1}$ leading up to cloud formation. At the same time, the relative humidity tendencies were often clustered around zero. The lack of change in relative humidity while temperature is changing indicates that changes to absolute humidity must also be ongoing; tendencies in near-surface specific humidity, while small, confirmed that advection of moister or drier air was an ongoing process during winter (not shown). Taken altogether, the evolution of clear periods in winter are more so dominated by large-scale thermodynamic advection rather than quiescent air mass transformation. The study by Kalesse et al. (2016) from the NSA found that dissipation of a low-level cloud was associated with converging air masses from different origins, consistent with the dissipation results here.

Dissipation and formation events during spring reflect a transition in the processes controlling the evolution of cloud free periods. Layer thickness tendencies varied between being as large as in winter, but also indicated a regime where tendencies were relatively small across both atmospheric layers. Leading up to cloud formation, especially low cloud and fog formation events, nearly all thickness tendencies were relatively weak and clustered around zero. Near-surface wind directions between the start and end of the clear periods were also consistently from the same general east-northeast direction. The relationship between temperature and relative humidity tendencies prior to formation were scattered, but a general negative correlation began to emerge. Thus, relative humidity began responding to changes in the near-surface temperature, likely during times when the synoptic forcing was weak and longwave cooling at the surface dominated the thermodynamic response.

By summer, a negatively correlated relationship between relative humidity and temperature became even more apparent. Layered thickness tendencies, while not indicative of a completely stagnant atmosphere, were small and clustered around zero. The geopotential thickness changes for the 500-700 and 700-850 hPa layers were weak relative to those during winter, indicating rapid, large-scale atmospheric forcing was predominantly missing in the hours leading up to summertime low cloud and fog formation. Furthermore, near-surface particles increased prior to formation, at the same time little change in near-surface wind directions were observed. Given that relative humidity was observed to increase while temperature decreased further reveals that local thermodynamic evolution was governed more by local cooling via net longwave deficit

than abrupt synoptic change. Such quiescent conditions prior in the final stages of the clear sky period provide a consistent process of air mass cooling towards saturation, with an abundant availability of particles with which to serve as nuclei for fog droplet formation.

Because significant synoptic variability was primarily non-existent in the lead up to summer low cloud and fog formation
events, it is unlikely the increased particle concentrations observed prior to formation were associated with abrupt air mass changes. However, particle number concentrations were 1.5 to 2 times greater in the two hours prior to low cloud and fog formation than in the two hours after. An analysis of near-surface particle size distributions from a number of Arctic observatories identified smaller Aitken-mode particles that dominated the distribution compared to the accumulation mode in summer (Freud et al., 2017); new particle formation events were attributed to the formation and growth of the smaller
aerosol mode. The enhanced concentrations and the optical properties of these particles observed on the NSA in summer are consistent with the new particle formation process during these clear sky, quiescent periods. Despite the dominance of the Aitken mode, the decrease in aerosol concentration after fog formation is most likely a result of aerosol activation.

## 6 Conclusions

A suite of in situ and remote sensing measurements and data products from the NSA have been analysed to determine the processes contributing to low cloud dissipation and formation events. The triggering mechanisms that support the cloud dissipation and formation events are important because they effectively commence or end a clear sky period. These clear sky periods have a profound impact on the surface energy budget, which further impacts the stratification of the lower troposphere. Improved understanding on Arctic clear period evolution has impacts on scales relevant to local weather and
climate.

We conclude that the onset of clear sky periods, and subsequently the end of clear periods, are primarily responsive to transient atmospheric forcing. While we report that all months are subjected to synoptic disturbances, the magnitude of the forcing is weakest during summer and strongest in winter, with transitions in the forcing strength occurring during spring
and autumn. Relatively homogeneous near-surface thermodynamics and winds during clear sky periods lends support to predominant quiescent conditions during the summer months. The weaker forcing promotes the near-surface temperature to drop through infrared radiative cooling to space, causing the relative humidity to increase in response to the thermodynamic adjustment.

At the same time, a nearly constant two-fold increase of aerosol particles near the surface was observed, suggestive of particle size growth in response to the new particle formation process. These processes provide the ingredients necessary for

the environment to support condensation and the development of fog. Air mass changes are likely not the cause for increasing near-surface aerosol concentrations since the thermodynamics and winds during the summer time clear-sky periods revealed little variability. Instead, enhanced stable stratification resulting from a lack of low cloud cover supports the

pooling of aerosols in a shallower boundary layer closer to the surface.

The mechanisms leading to cloud dissipation are less apparent. Comparison of aerosol backscatter profiles from the HSRL after cloud dissipation and before cloud formation were not statistically different. Backscatter showed that aerosol remained present and relatively consistent both for after dissipation and for before formation events. Because of this consistency, it is

unlikely that a paucity in aerosol presence caused the dissipation of the cloud layer. Relatively large geopotential thickness tendencies were observed prior to dissipation during winter, spring, and autumn. Together with larger wind shifts, dissipation of clouds in winter were commonly connected to an active synoptic setting. A frontal passage or air mass trajectory change, like that reported to have caused the dissipation in Kalesse et al. (2016), is consistent with our findings. Currently, we are examining the potential validity of aerosol changes in causing Arctic cloud dissipation with the help of cloud resolving

model simulations that incorporate detailed aerosol physics. Detailed case studies will be explored to address the impacts of varying aerosol number, vertical structure partitioning, and hygroscopic properties on cloud dissipation, and furthermore on the formation of low-level clouds or fog, which have been shown in this study to be the predominant Arctic cloud type following prolonged clear-sky periods.


**Acknowledgements**

This research has been supported by the U.S. Department of Energy's (DoE) Office of Biological and Environmental Research (BER) Atmospheric Systems Research grant no. DE-SC0019073. The authors thank the four reviewers for the engagement in our manuscript and for their extensive suggestions to improve the scope of this study.


**Data Availability**

All observations analysed in this study are freely available to the user community by following the links provided here to their respective repositories. The ceilometer measurements are accessible from the ARM Data Archive:

https://adc.arm.gov/discovery/#/results/s::nsaceilC1.b1. The HSRL observations are accessible from the ARM Data Archive: https://adc.arm.gov/discovery/#/results/s::nsahsrlC1.a1. The cloud boundaries derived from the ARSCL processing algorithms from the KAZR are accessible from the ARM Data Archive: https://adc.arm.gov/discovery/#/results/s::nsaarsclkazr1kolliasC1. The. RadFlux surface radiation measurements and data products are accessible from the ARM Data Archive:

https://adc.arm.gov/discovery/#/results/datastream::nsaradflux1longC1.c1. The radiosoundings are accessible from the ARM

Data Archive: https://adc.arm.gov/discovery/#/results/s::nsasondewnpnC1.b1. Near-surface meteorology measurements are accessible from the ARM Data Archive: https://adc.arm.gov/discovery/#/results/s::nsametC1.b1. Finally, near-surface CPC measurements are accessible from the NOAA Global Monitoring Laboratory ftp server: https://www.esrl.noaa.gov/gmd/dv/data/index.php?parameter_name=Aerosols&site=BRW. Reanalysis data from the
European Centre for Medium Range Weather Forecasting ERA5 are accessible from the Copernicus Climate Data Store: https://cds.climate.copernicus.eu/cdsapp#!/dataset/reanalysis-era5-pressure-levels?tab=overview.

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

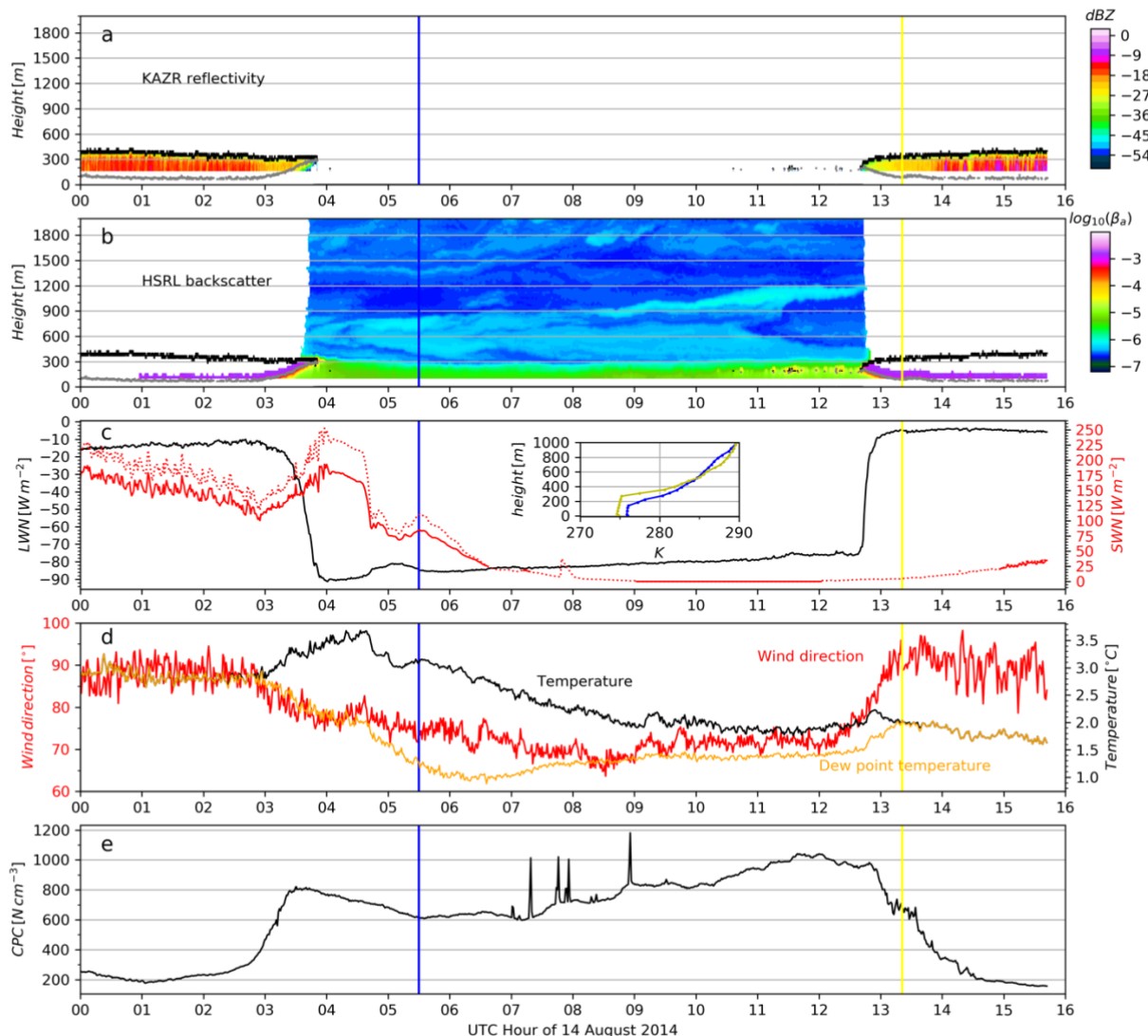

**Figure 1: Temporal evolution [UTC] of cloud dissipation, a clear sky period, and the formation of a cloudy period from 14 August 2014 from the North Slope of Alaska. a) KAZR reflectivity [dBZ, contours] and cloud top (black) and base height (gray) boundaries. b) HSRL backscatter [log₁₀(Ba)] including cloud top and base boundaries; c) Net longwave (black) and net shortwave (red) radiation, including downwelling shortwave (dashed red), all in W m⁻²; the inset includes equivalent potential temperature [K] profiles from radiosoundings at 05:30 (blue) and 13:15 UTC (yellow); the vertical blue and yellow lines in each panel represent the radiosounding launch time. d) Near surface wind direction [degrees, red], temperature [K, black] and dew point temperature [K, orange]. e) Near surface particle concentration [N/cm³].**

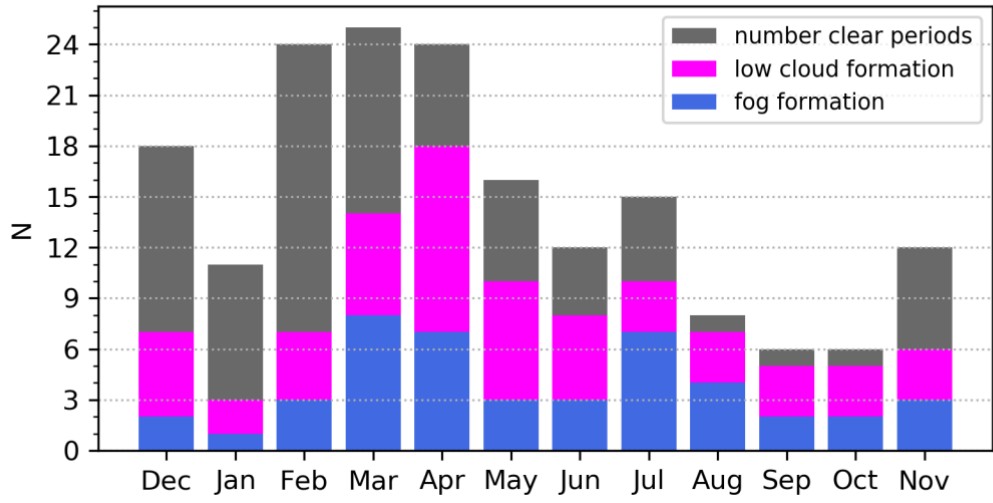

**Figure 2: Monthly occurrences of clear sky periods determined from the remote sensing suite at ARM-NSA during 2014 to 2018 (gray bars). Magenta bars represent the number of clear sky events that ended with the formation of a low cloud layer (cloud base below 400 m AGL) and blue bars for events with fog formation.**

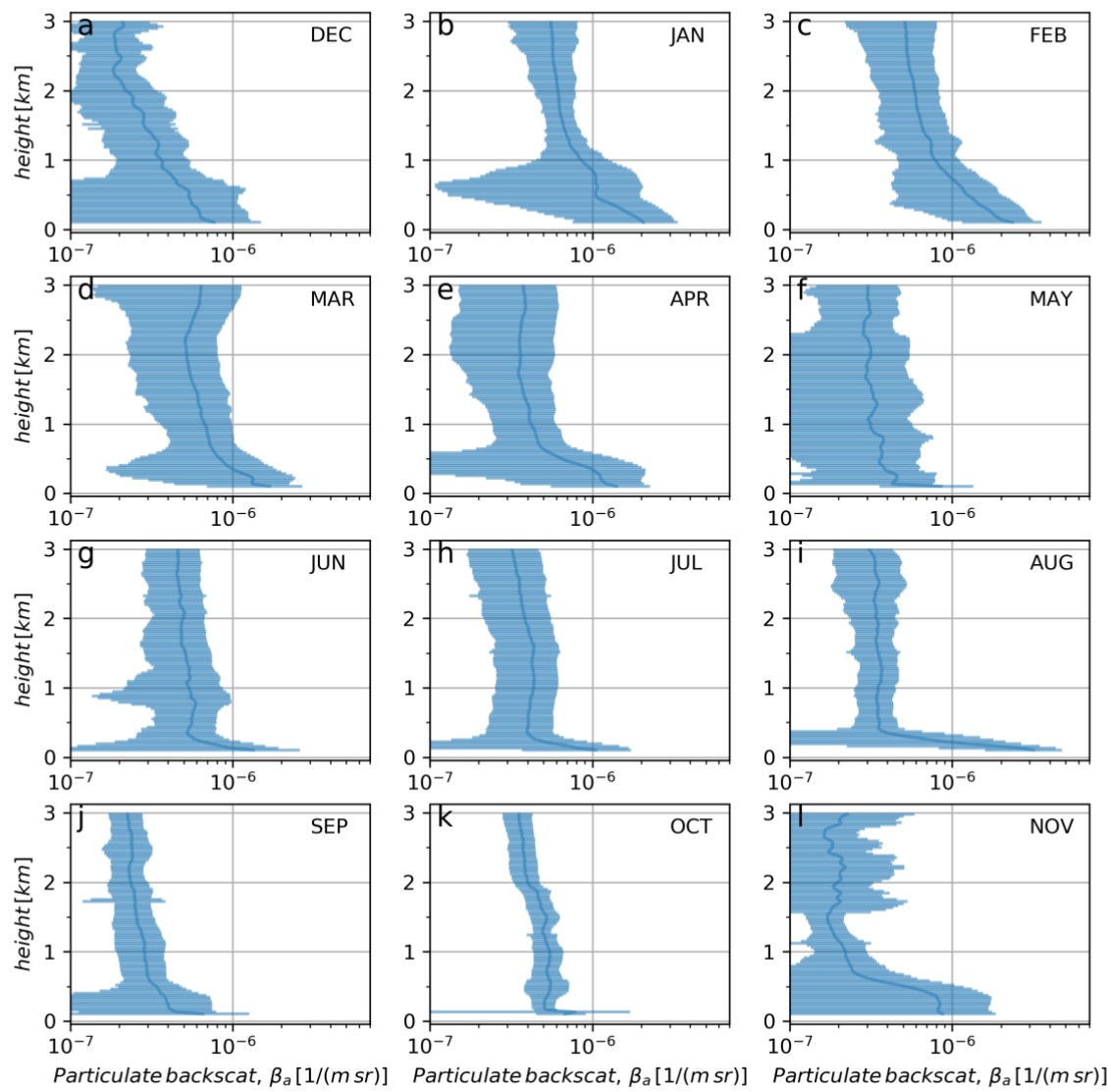

**Figure 3: Monthly mean and 1-sigma HSRL backscatter [1/(m sr)] profiles up to 3 km AGL during clear sky periods. Rows are arranged seasonal from top to bottom: a-c) DJF, d-f) MAM, g-i) JJA, and j-l) SON.**

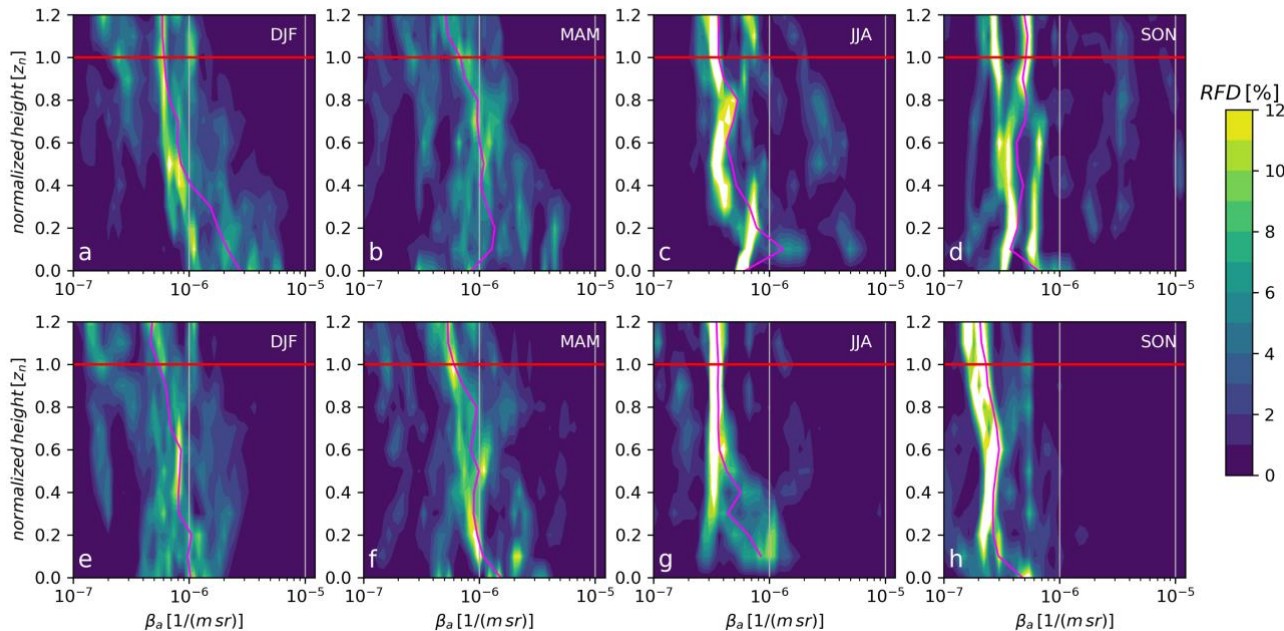

**Figure 4:** Relative frequency distributions (RFDs) [%, colors] of aerosol backscatter as a function of normalized height, $z_n$, where $z_n = 0$ is the surface and $z_n = 1$ is the former/successive mean cloud top height surrounding the clear sky period. All HSRL backscatter profiles after/before 60 minutes of cloud dissipation/formation are combined to create the frequency distributions, which are normalized to 100% at each normalized height range. Seasonal distributions for DJF, MAM, JJA, and SON are shown for a-d) after cloud dissipation (start of clear period), and e-h) prior to cloud formation (end of clear period). Median profiles for each season are given in magenta.

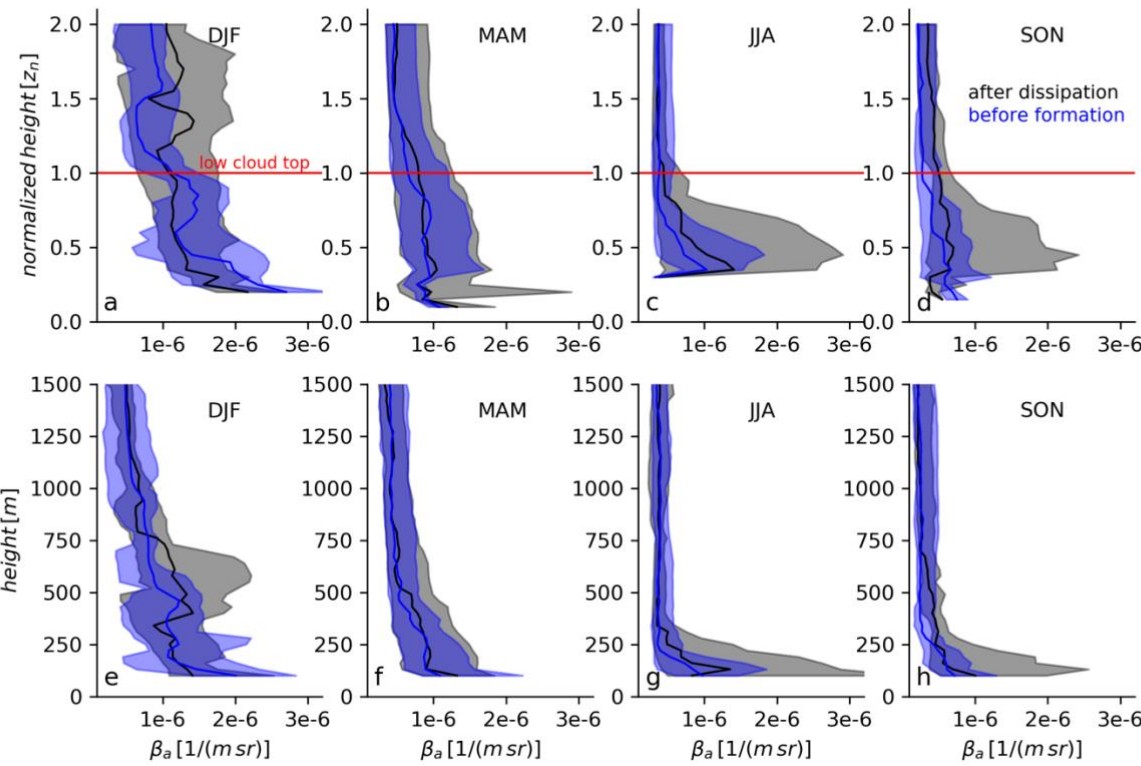

**Figure 5: Seasonal (DJF, MAM, JJA, SON) median (solid line) and interquartile range (shading) profiles of clear-sky aerosol backscatter [1/(m sr)] only for clear periods when a low cloud (base < 400 m) or surface fog was observed to form. Black (gray shading) profiles are for backscatter within 30-60 min period after cloud dissipation; blue (light blue shading) profiles are for backscatter within 60 to 30 min prior to low cloud formation. Panels a-d are normalized in height of the height of the forming cloud top, while e-h show the full profile up to 1500 m AGL.**

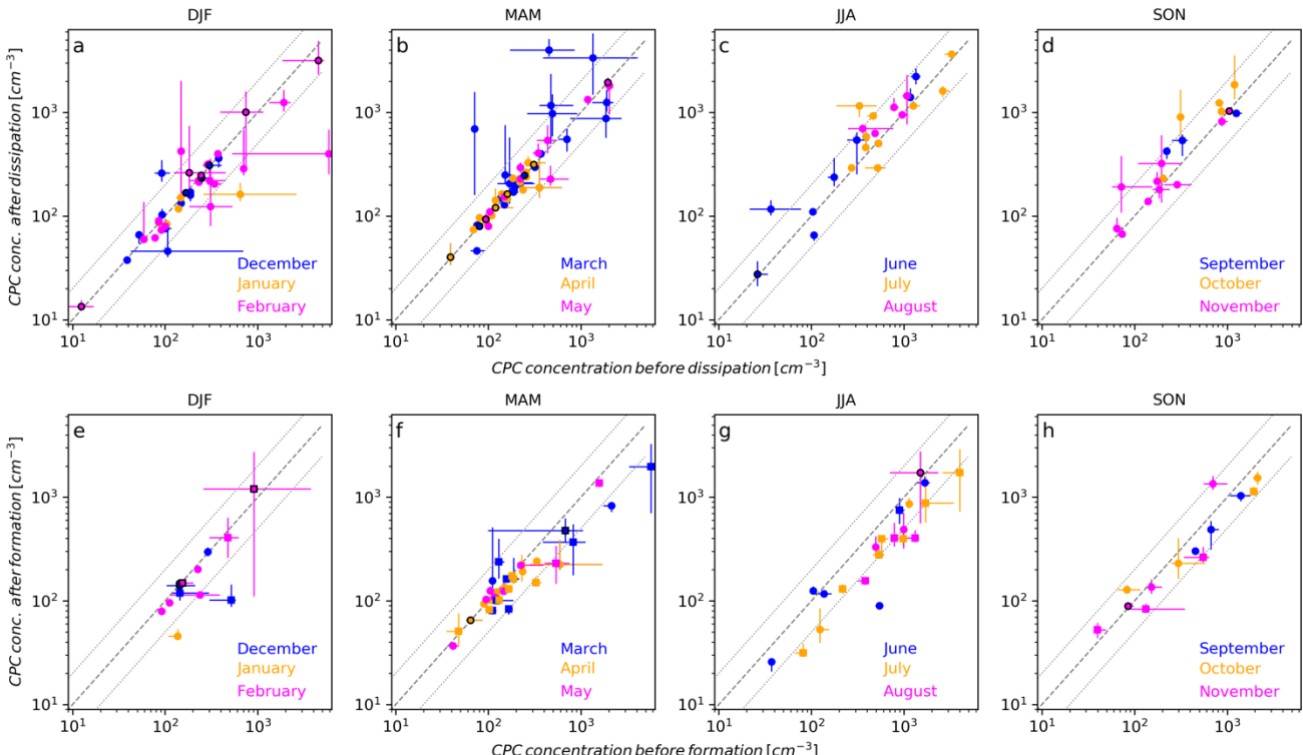

**Figure 6: Median (circles) and interquartile range (lines) of CPC concentrations [cm⁻³] in 2-hr period after cloud dissipation versus 2-hr period before cloud dissipation (a-d), and 2-hr period after cloud formation versus 2-hr period before cloud formation (e-h). Monthly cases are in colors and labeled in each subpanel, with the months grouped by season from left to right: DJF, MAM, JJA, and SON. A Wilcoxon rank-sum statistical significance test was calculated for each CPC distribution prior and post cloud lifecycle change. Events where the distributions around cloud changes were not significantly different at the 95% confidence level have a median symbol outlined in black; a median symbol without black outline indicates significantly different CPC distributions at the 95% level around a cloud lifecycle event. The 1:1 gray dashed line, and 1:2 and 2:1 dotted gray lines, are included as a reference. Note the logarithmic axes.**

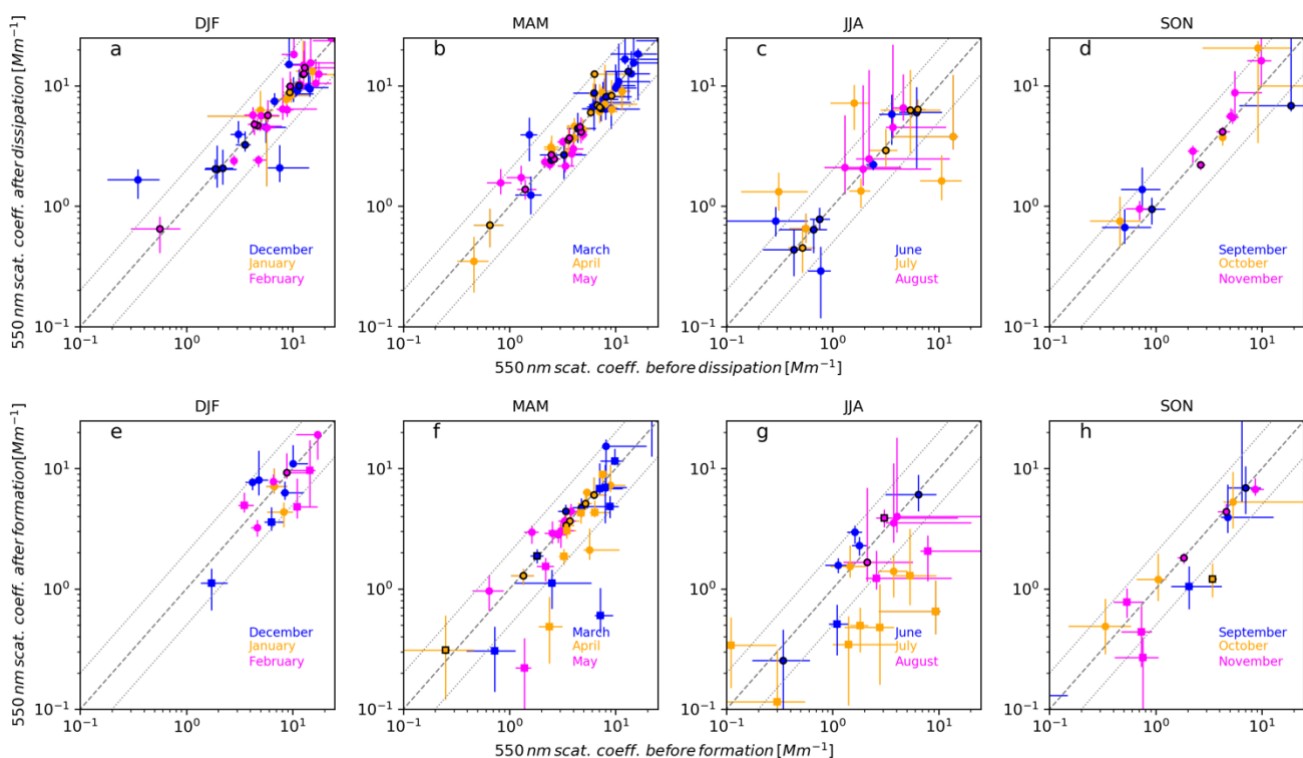

**Figure 7: Same as in Fig. 6, but for distributions of 550 nm scattering coefficient [Mm⁻¹].**

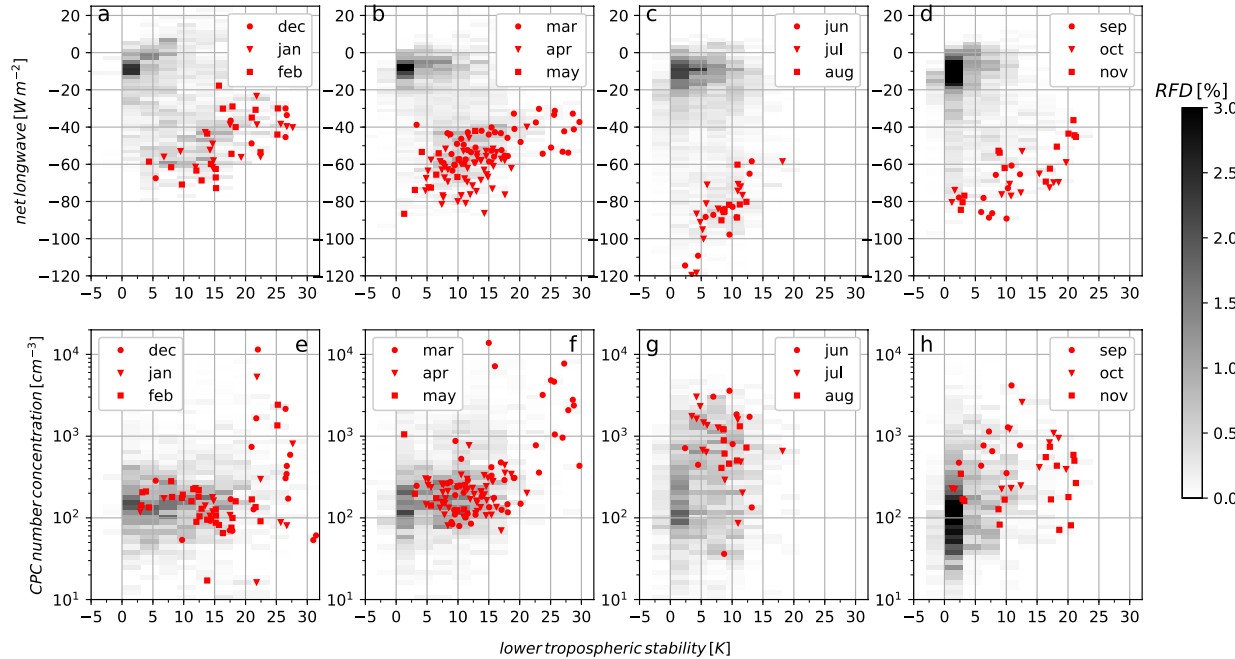

**Figure 8 (7): Relative frequency distributions (RFDs, gray contours) of a-d) net longwave radiation (LWN, [W m⁻²]) as a function of lower tropospheric stability (LTS, [K]) for a) winter, b) spring, c) summer, and d) autumn. E-f) Relative frequency distributions of near-surface CPC concentrations [cm⁻³] as a function of LTS. LWN and CPC concentrations are taken within 10 min of each radiosounding profile used to estimate LTS. Red symbols represent the individual relationships between LTS and LWN/CPC**

**values within 10 min of the radiosounding during the clear sky periods; each month within the season is represented by a different symbol.**

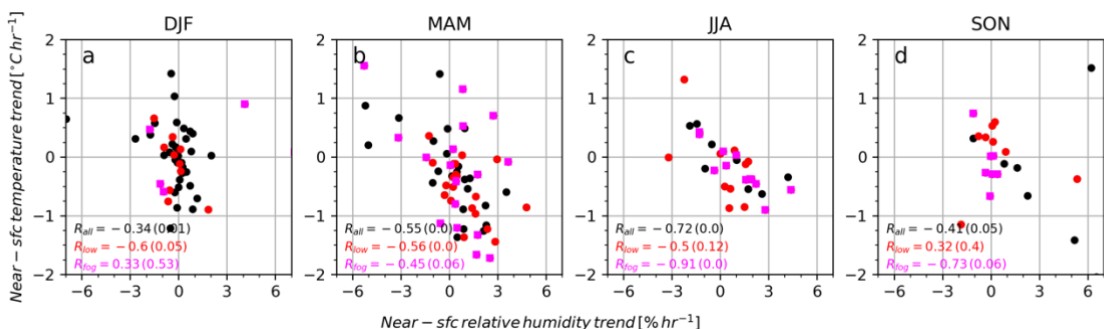

**Figure 9 (8): Seasonal linear trends in near surface air temperature [°C hr⁻¹] versus linear trend in relative humidity [% hr⁻¹]**
computed using linear regression of temperatures and relative humidity in a 2-hr period prior to elevated (black circles), low cloud
(red circles), and fog (magenta squares) formation. Relative humidity was computed with respect to ice for November through
May and with respect to liquid for June through October, based on monthly mean near-surface temperatures. Seasonal Pearson
correlation coefficients and p-values are included for each subset of cloud formation type.

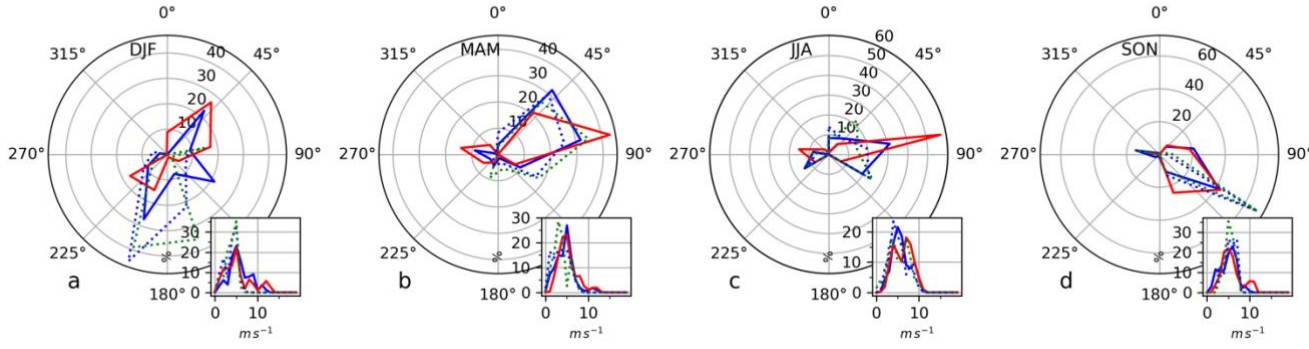

**Figure 10 (9): Seasonal RFDs (radii, [%]) of near-surface wind direction [degrees] within 1-hr after cloud dissipation (solid red) and within 1-hr of low cloud/fog formation (solid blue) for a) DJF, b) MAM, c) JJA, and d) SON. Wind direction RFDs within 1-hr surrounding fog formation events only are shown as dashed lines (blue is 1-hr prior and green is 1-hr after fog formation). Insets in each panel show the RFD of wind speed [m s⁻¹].**

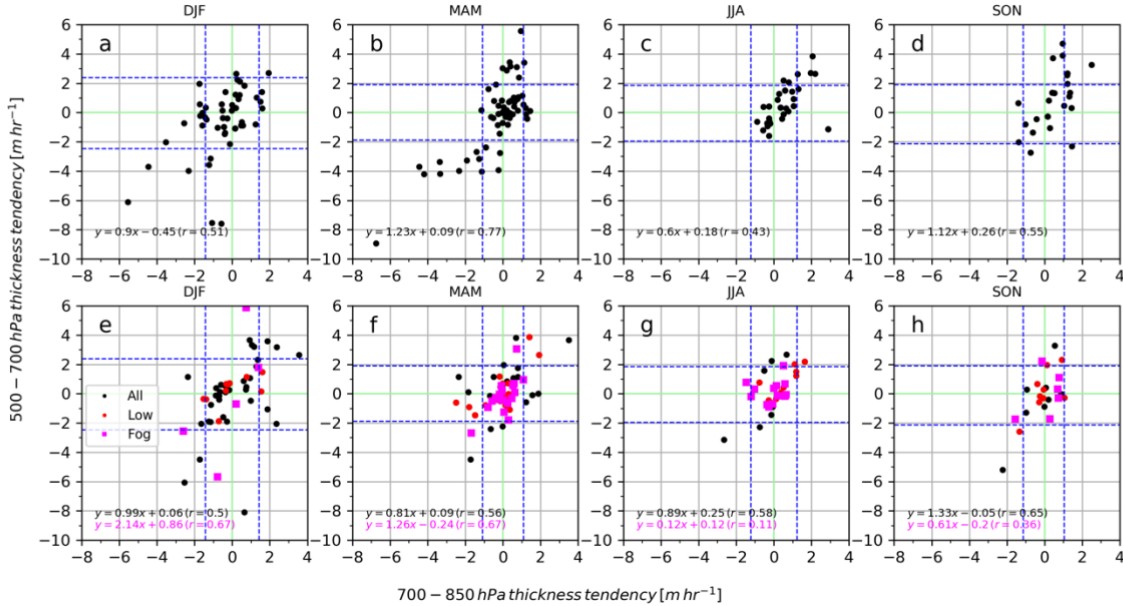

**Figure 11 (10): Geopotential height thickness tendencies [m hr⁻¹] of two atmospheric layers, 500-700 hPa and 700-850 hPa leading up to cloud dissipation (a-d) and cloud formation (e-h). Tendencies are computed from ERA5 layer thicknesses in a 4-hr period prior to cloud dissipation and before cloud formation. In e-h, black circles represent elevated cloud formation events, red circles for low cloud formation events, and magenta squares for fog formation events. The dashed blue lines show the seasonal mean ± one standard deviation computed from consecutive 4-hr layer thickness tendencies for each season. Seasonal linear regressions and associated Pearson correlation coefficients are provided in each panel.**
