# Peer review of "Processes contributing to cloud dissipation and formation events on the North Slope of Alaska"

_Atmospheric Chemistry and Physics, 2020_

## Referee Comment (RC1) · Anonymous Referee #1 · 31 Aug 2020

This manuscript describes an analysis of clear-sky periods following cloud dissipation and prior to cloud formation over Utqiagvik, Alaska, with a focus on low clouds. The authors use a rather comprehensive set of ground-based measurements spanning over 5 years to draw insights on the processes contributing to cloud formation and dissipation. Different clear-sky period properties as a function of season are examined, from which the authors suggest differences in the impact of synoptic-scale forcing on cloud formation and dissipation. The authors also find that a scarcity of aerosol particles is likely not the dominating cause for cloud dissipation, and postulate based on their analysis and the literature that cloud formations from late spring to early autumn largely initiates at or near the surface.

[Figure]

The manuscript is generally well written and I found the analysis description intriguing. I appreciate the amount of information that the authors were able to extract from the ground-based measurements and think that this manuscript provides a new analysis of an atmospheric phenomenon, which is often left without being properly examined, namely, clear sky periods. While I agree and/or find sense in most of the authors' interpretation and conclusions, I have a few concerns regarding the methodology and the analysis description, as well as a high number of rather minor comments, which I think the authors should address before this study can be accepted for publication in ACP. I, therefore, recommend major revisions.

Major comments: 1. Definition of a strict clear-sky period – I find the methodology rather robust. However, to my understanding, once the 2-h clear-sky threshold is met, intermittent clouds can be detected, as long as the total duration of cloudy periods does not exceed 4%, for example, in the case of a 10 h clear sky period, the last hour may contain the only 24 minutes of (broken) clouds. A similar example is provided in Fig. 1, in which I cannot agree with the authors' description in the text (l. 145-157) that the period between 10:30-12:45 UTC is strictly a clear sky period; that is, the KAZR, HSRL, and LW measurements all suggest the intermittent presence of a cloud layer (e.g., at ∼11:30 UTC), obviously a tenuous one, and hence, the weak LWN signature, but this is still definitely a cloudy period. Now, I understand that the data analysis here requires a binary definition of either a "clear" or "cloudy" period and that an addition of an intermediate class period would likely introduce multiple inconsistencies. However, with the current methodology settings and constraints (duration of a clear/cloudy period, altitude limit for cloud occurrence, etc.): a. The clear sky and cloudy period portioning results in a very high overlap with the "radiatively clear/cloudy" states coined by Stramler et al. (2011), which is essentially the only way to argue that in Fig. 1 there is a 9-h long clear-sky period rather than ∼7.0-7.5-h period. The authors should address this point here and other places in the text where it is applicable. b. Clear sky periods can actually be cloudy, so I think that the authors should omit the use of the "strict" clear sky period definition throughout the text, which can become rather subjective, among other

reasons, because of the multiple variable thresholds in this study (e.g., one could argue that only the periods where the thresholds mentioned in l. 236-238 correspond with a "strict" clear sky definition). I recommend the authors to consider terming the clear sky periods in conformance with their effective partitioning of the dataset, for example, I would suggest using the term "prolonged clear sky periods" (corresponding to the duration requirement while remaining objective by not introducing subjective criteria), which precede/follow "persistent cloud occurrence periods".

2. Inconsistent duration thresholds - why were 2 hours used for the analysis in fig. 6 (CPC counts), unlike the rest of the data analysis? How much would the results change with 1 -hour windows? Why were 4-hour windows used in the temperature trend analysis (sec. 4.3.2) instead of consistently working with 1-hour windows? Also, could the 1-hour window allow some separation of fog events (necessarily positive Td depression) from other low cloud events (potentially all depression values possible)? For both the CPC and temperature trend analyses, could there be an influence of intermittent cloudy periods just before (after) cloud dissipation (cloud formation) being classified as part of a clear sky period? How different do the scatter plots look using 1-hour windows instead of the utilized 2/4 hour duration windows?

3. Synoptic forcing methodology and analysis: a. I understand that quasi-geostrophic flow occurs further away from the surface, but if largely low clouds are examined here (cloud base up to 400 m AGL in the main data subset; 3 km for the full dataset), why aren't tendencies in a near-surface layer thickness (e.g., 850-950 hPa) examined here (using a similar or different methodology), being more representative and consistent with the presented analysis thus far? It is not obvious to me how much analyzing such a low atmospheric layer could impact the results and discussion throughout this section, for example: - The conclusion that cloud dissipation events are impacted by relatively homogeneous thermal advection across the lower to mid-troposphere (l. 392-393). - "we identify that the height level where cloud formation events occur may be influenced by a weaker synoptic setting from December through May" (l. 413-414). - "Larger-scale

differential advection is almost always ongoing prior to cloud dissipation, and as such it is assumed that different air mass origin and thermodynamic properties are likely to go in unison with changing aerosol properties" (l. 427-428). b. Point measurement nature of the ground-based data influencing the interpretation - I understand that the results indicate a consistent/inconsistent wind regime in different seasons (e.g., l. 356-360). Could the consistency of the low-level wind direction examined here be the result of a strong micro-meteorology, e.g., prevalent sea-breeze over Barrow (during sunlit periods; hence, the narrowest distribution during summer when SZA is lowest), which masks synoptic forcing, which could still have a significant influence on a mesoscale? By the same token, high variability during winter is influenced by the synoptic-scale flow (e.g., l. 353-354), but that signature could be enhanced (relative to other seasons) by weak/lack of micro-meteorological sources (e.g., during dark periods). This is an additional degree of freedom in the data that the authors need to consider (e.g., using reanalysis or nearby surface stations) in order to support their conclusions in l. 500-502, 506-508.

4. Given the relatively small effective dataset, statistical significance tests could have a large impact on the discussion and interpretation of the results. The authors should perform such tests and refer to them throughout the discussions for which they are relevant (e.g., l. 262-264, l. 360-362, l. 391-392)

Minor comments: - There is an occasional change of tense throughout the manuscript (e.g., l. 16-20, 196-215, 311-319, 387-398). I recommend the authors to be more consistent from this aspect, as I think that it improves the manuscript's readability.

- "Barrow" should be replaced with "Utqiagvik" throughout the text (except for in the abstract and introduction).

- l. 24 – Because the essence of this first sentence is elaborated below, I suggest removing the first reference or adding a few more references (e.g., Curry et al., 1996, https://journals.ametsoc.org/jcli/article/9/8/1731/36313), as Herman and Goody (1974)

only discussed summertime clouds, in which SW radiation plays a role.

- l. 27 - add "water" before "particles"

- l. 96 - suggest modifying "the signal becomes" to "the signal typically becomes"

- l. 101 - I do not think that this is necessarily true over the Arctic. There are numerous examples of cases in which the droplet size and/or concentrations are too small to be detected by ground-based Ka-Band radars such as KAZR (e.g., first hour in Shupe, 2011, fig. 1, where the radar echoes are below cloud base). I agree that voxel-wise KAZR is indeed capable of detecting (in its high-sensitivity mode) most hydrometeor echoes, but many tenuous liquid-bearing clouds (which are common in the Arctic) can remain undetected by radar. I suggest rewording this sentence to address this general misconception.

- l. 113-115 - Note that Long and Turner (2008) only analyzed the downwelling LW and only during clear-sky periods, in which the downwelling fluxes are relatively lower, and found that the 4 Wmˆ-2 value holds for only 2/3 of the NSA cases. The LW flux uncertainties are likely larger and contain a flux percentage component, as also suggested by the ARM handbook for these pyrgeometers (see Table 6 in https://www.arm.gov/publications/tech_reports/handbooks/sirs_handbook.pdf). I recommend the authors to update this discussion accordingly.

- Fig 1. – The title for panel a says "MMCR reflectivity" instead of "KAZR reflectivity".

- l. 147 - following major comment #1, suggest changing to "nearly 7 hours" or elaborate accordingly.

- l. 191-194 – This sentence is slightly confusing. I suggest rewording and or breaking it in two.

- l. 165-166 - I suspect that these are 24 individual periods per month? Or is it a MAM seasonal mean? In which year?

- l. 187-189 - This is hard to interpret in the logarithmic scale used in Fig. 3.

- Fig. 3 - Because the aerosol beta signal is largely concentrated on a single order of magnitude, I think that the logarithmic scale on the x-axis makes the figure more difficult to interpret, especially with regards to the shaded sigma, which may become misleading depending on the mean value. I suggest setting the x-axis scale to linear and/or plotting profiles of the SD absolute value and/or fraction relative to the mean value.

- l. 189-190 - I cannot agree that this is the case in some months, e.g., DEC, FEB, MAR.

- l. 190-191 - Difficult to say that for panel f

- l. 192 - "depth of the enhanced" –> "depth of an enhanced"

- l. 204 - suggest removing "It is interesting that"

- l. 212 - "therefore impossible" –> "therefore it is impossible"

- l. 213-215 - To my understanding of the text description, if the transition phase (sloped part of the profile) occurs at or above cloud base levels, then this statement doesn't hold, because there is a certain depth (either just at cloud base or above) where the aerosol profile appears to be more similar to the aerosol profile below, as also suggested from Fig. 1, so based on the HSRL beta measurements alone, I could argue that the aerosols in the cloud layer are very similar to those at the surface and that the surface is actually representative. I agree that the surface aerosol properties are likely often unrepresentative of aerosol properties at the cloud level, but only because of previous dedicated aerosol studies (some with in-situ measurements) and because surface or near-surface inversions/stable layers are so common over the Arctic (e.g., Tjernström and Graversen, 2009, https://rmets.onlinelibrary.wiley.com/doi/abs/10.1002/qj.380).

- l. 217-219 - That is a nice and significant observation. Do the authors think that this

conclusion holds for other Arctic regions?

- l. 221-222 - what is the time range before/following dissipation/formation that the authors use to determine whether the data corresponds with the 2 km threshold?

- l. 226-227 - the lidar signal could be fully attenuated by cloud, but is not necessarily fully attenuated by cloud. Perhaps the authors can simply say that using a subset of cases without full lidar attenuation before dissipation/after formation would result in very few samples to analyze (number of examined samples is already rather low), also because of the data filtering (see major comment #1a).

- l. 227-230 - Indeed, precipitating hydrometeors typically dominated the aerosol signal, but occasionally there are cases in which the ice number concentration is so low, that it is barely detectable in lidar measurements. I recommend the authors to add "typically" or "largely" to the text. Also, please change "drizzle droplets" to "drizzle drops".

- l. 241-242 - I am not sure this can be said without forward calculations of aerosol properties (given that the backscatter is proportional to the surface area, which is proportional to particle size in addition to concentration), which requires some information not available with this dataset. I would be hesitant to postulating that.

- l. 242-244 - such findings were first reported up to a few decades ago and should be cited here as well, e.g., Curry et al. (1996, https://journals.ametsoc.org/jcli/article/9/8/1731/36313) and references therein, Jiang et al., (2001, https://agupubs.onlinelibrary.wiley.com/doi/abs/10.1029/2000JD900303), Fridlind et al. (2012, https://journals.ametsoc.org/jas/article/69/1/365/27245).

- l. 244-246 - I cannot tell whether this sentence agrees with the data depiction in Fig. 5 because the cloud base height was not considered in the height normalization.

- l. 247 - by flatter do the authors mean less variable?

- l. 248 - define RFD

- l. 248-249 – Not sure I understand the authors' intention here. Perhaps "typically agrees in magnitude with the profiles prior to cloud dissipation"?

- Fig 5 and the associated discussion - Fig. 5 is hard to interpret because: a. There is no normalized height for cloud base. b. The small dataset combined with the interpolated shading can be misleading. While I agree with the general conclusions of this discussion, currently it is rather difficult to evaluate and follow the different arguments. I recommend using larger RFD bin widths and adding a normalized height for cloud base (e.g., at 0.5).

- l. 256-260 –The authors analyze here specific periods, which makes me wonder whether these seasonal patterns agree with the published literature of aerosol/CCN/CN seasonal variability over the NSA (e.g., Quinn et al., 2002, https://agupubs.onlinelibrary.wiley.com/doi/10.1029/2001JD001248; Lubin et al., 2020, https://journals.ametsoc.org/bams/article/101/7/E1069/345559), or are these seasonal signals muted in the bulk statistics reported in the literature?

- l. 257-260 - This sentence reads awkwardly - suggest rewording.

- l. 262-264 - Unlike the formation plots in panels e-h, I do not see any major difference between these panels (a-b, c-d). I think that the authors need some statistical significance tests here to convince a reader (see major comment #4).

- l. 266 - define CPC

- l. 276 - I do not think that the analysis failed to identify drastic signal changes, it simply did not identify significant changes, which I think is a nice observation on its own. I suggest rephrasing this sentence and remove "Therefore" in the following sentence.

- l. 287 - "sub-cloud mixing driven by cloud-top turbulence" - suggest changing to "cloud-top radiative cooling" or "cloud-induced turbulence".

- l. 289-291 - the effective cloud temperature is rather important as well, i.e., the LWN is primarily proportional to the effective temperature

differences between the cloud (cloud emissivity profile considered) and the surface, e.g., compare the LWN histogram in Stramler et al., (2011, https://journals.ametsoc.org/jcli/article/24/6/1747/32737) to that in Silber et al., 2019 (https://agupubs.onlinelibrary.wiley.com/doi/full/10.1029/2018JD029471).

- l. 291-292 - Generally speaking, utilization of the equivalent potential temperature variable to examine static stability is only valid in moist processes (see Ch. 4 in Emanuel, 1994), e.g., in the case of a fog cloud extending from the surface up to 950 hPa. Otherwise, the virtual potential temperature or liquid water potential temperature should be used. Alternatively, the authors may use the equivalent potential temperature only in liquid-containing heights. I suspect that addressing this comment as suggested above would not significantly change the results in this section, and therefore, consider this a minor comment.

- l. 302-310 - That is a nice discussion. The authors should also explain the apparent correlation between LWN and LTS (e.g., stronger stratification may indicate higher downwelling LW due to relatively higher temperature at 950 hPa or so, strongly depending of course on sufficiently high q_v at the 950 hPa level or so).

- l. 309 - suggest "warmer temperature" –> "higher temperature"

- Fig. 7 & 10 - Please add the season to each panel, as already shown in other figures. It helps to follow the text.

- Fig. 8 - suggest using constant y-axis limits - evaluation of the figure can be confusing at the moment (especially with regards to panel b).

- l. 325 - suggest "strongly" –> "largely"

- l. 331 - "Dew point depressions are then computed" - suggest rewording this part of the sentence.

- l. 358-362 - "insert/s" –> "inset/s"

- l. 367-369 - Isn't the moisture content a second-order term relative to mean temperature? If so, then I suggest focusing on the temperature or providing some observational evidence (e.g., via references), about temperature and moisture advection occurring commensurately.

- l. 372 - does "quasi-geostrophy" refers to "quasi-geostrophic flow"? Also, please provide a chapter or page number for Holton, 1992.

- l. 382-385 - I do not find this argument convincing (e.g., in panels b, f), and the current version of Fig. 10 where x- and y-axis limits are inconsistent between one-another and between panels depicts a misleading picture. Also, given that one should expect larger tendencies at higher altitudes (lower pressure levels), then what scale of tendencies in each layer would be considered small?

- l. 387-389 - This sentence is rather confusing. I suggest rewording. Also, I think that labeling the figure quadrants might really help a reader follow this section and quickly understand that quadrant x represents thermodynamic structure change y, and so on.

- l. 391-392 - with such a low number of samples, I would not consider r=0.44 or r=0.5 for that matter to represent moderate correlation (explaining not more than a quarter of the thickness co-variability). In any case, statistical significance for each month should be examined and provided here as well (see major comment #4).

- l. 395-398 and throughout the discussion concerning Fig. 10 - the authors should consistently use (in the text and figure) height or thickness; to my understanding, the authors only refer to thickness as a reference height is not provided.

- l. 401-403 - Does this nice observation also agrees with the fact that frontal clouds are often observed during winter, as suggested by the wind direction analysis?

- l. 413-414 – I recommend replacing "weaker" with "shallower".

- l. 428 - suggest changing "is almost always ongoing" to "often occurs"

- l. 431-432 - what do the authors mean by "aerosol vertical partitioning"?

- l. 452-453 - suggest removing "of the lower atmosphere".

- l. 483 - redundant "the"

- l. 483-484 - given the implemented methodology, the authors should explicitly state in this sentence that they refer to low-clouds (obviously, not necessarily fog).

- l. 514-515 - again, this is often the case, not always. As suggested above, the authors can say that when omitting the fully attenuated cases, the data subset becomes too small to be meaningful.

- l. 519-521 - I would be hesitant to claim that the dominant Arctic cloud type following clear sky periods are low clouds, even though it could be tentatively suggested by the liquid/mixed cloud RFDs presented in Shupe (2011). My main concern here is about the subjectiveness of a clear-sky definition based on the methodology (see major comment #1b). To correspond with the methodology (e.g., 2-h clear sky and cloud occurrence period thresholds) I could agree with "formation of persistent low-level clouds or fog, which have been shown in this study to be the dominant Arctic cloud type following prolonged clear-sky periods."

- l. 739 - Fig. 6 caption - should be (e-h)

- l. 744 - RFD should be first defined in the caption of Fig. 5 where it is first mentioned.

- l. 759 - insert –> inset

- l. 764-766 - "e-g" –> "e-h"
* * *

---

## Referee Comment (RC2) · Ian Brooks (Referee) · 7 Sep 2020

Review of "**Processes contributing to Arctic cloud dissipation and formation events that bookend clear sky periods**" by J. Sedlar, A. Igel, and H. Telg.

Ian M. Brooks

**Overview**

This paper aims to shed some light on the processes that control the dissipation and formation of low cloud in the Arctic. Such cloud is near-ubiquitous, but infrequent cloud-free conditions are important because of the large contrast in the surface radiation budget between clear and cloudy conditions. Models fail to adequately represent Arctic boundary-layer cloud and (operational forecast models) often fail to reproduce observed cloud free conditions. There is thus a definite need for improved understanding of the processes controlling these clouds.

The approach taken here is to utilise 5 years of measurements from a long-term measurement site at Utqiagvik, on the north coast of Alaska. The measurements include lidar backscatter (a proxy for aerosol concentration profiles), cloud radar, radiosondes, surface meteorology, and surface measurements of total aerosol concentration. Most of the analysis focuses on the ~1 hour period following cloud dissipation or preceding cloud formation that 'bookend' periods that are entirely cloud-free.

The analysis first considers the relationships between cloud dissipation/formation and aerosol profiles, comparing the profiles immediately after/before the transition with those for the clear periods as a whole (broken down by month), going on to consider the surface aerosol concentration either side of cloud transitions, and relationships between aerosol and lower tropospheric stability under both clear and cloudy conditions. This analysis provides no significant evidence for a causal link between aerosol properties and cloud dissipation/formation at the measurement site.

The analysis then considers thermodynamic and dynamic processes. This analysis leads the authors to conclude that "*the onset of clear sky periods, and subsequently the end of clear periods, are primarily responsive to transient atmospheric forcing*". For the onset of cloud they essentially conclude that under clear skies radiative cooling causes a fall in temperature and associated increase in relative humidity; ultimately saturation point is reached and provided there are sufficient aerosol present low cloud or fog will form. No firm conclusions are drawn about the processes resulting in the dissipation of cloud, other than the association with '*transient atmospheric forcing*'.

These conclusions are rather generic and unlikely to help improve modelling of Arctic cloud.

The results remain of interest in providing a picture of typical conditions and some seasonal variations thereof, for periods of clear air bookended by low level clouds. There is considerable scope to improve this picture, however, and I recommend major revision before publication is considered.

**General/major comments**

While the aim of the paper is very worthwhile, I feel it ultimately fails to deliver robust conclusions. In part this is a, perhaps inevitable, result of the limitations of the data set. The aim is to understand what the processes are that lead to cloud dissipation/formation – transient events that are inherently linked to changes in local air mass properties over time. Measurements from a fixed site are, however, unable to distinguish between temporal evolution of the air mass properties resulting from in situ processes and the simple advection of a pre-existing spatial gradient in properties past

the measurement site. This is a perennial problem for intensive, and/or long-term measurements. The authors attempt, but I think ultimately fail, to work around this problem by studying the statistics of an ensemble of cases. This provides correlations between measured properties associated with cloud transitions, and the hope is that probable processes can be inferred from these correlations. It is quite possible that observed behaviours might only be explicable by specific processes, and a fairly robust conclusion may be drawn. Sadly I don't think that is the case here.

**Aerosol Analysis**

The analysis of links with aerosol properties is quite extensive, but ultimately finds no causal links with cloud dissipation/formation. The extensive initial focus on aerosol is (I assume) prompted by results from the central Arctic Ocean where very low aerosol concentrations (< 10 cm$^{-3}$) have been found to result in clear sky conditions even when the boundary layer is saturated, and several modelling studies have found that it is essential to accurately represent the aerosol in order to effectively represent the cloud and boundary layer structure. (as a side note, I find it odd that while the authors cite 3 modelling studies, all of which utilise the same observed case from the ASCOS project, they don't cite the original observational paper that first documented such CCN limited conditions and on which Sedlar is a co-author).

The CCN limited conditions in the central Arctic, are from a very different environment from the coastal site used here. The surface aerosol measurements in figure 6 and 7 show that concentrations rarely fall much below ~100 cm$^{-3}$, and are often much higher – far too high for aerosol to be the limiting factor on cloud formation. I think this possibility could have been ruled out much more easily by simply evaluating the surface concentrations (and perhaps relating them to the lidar profiles) for clear sky cases, without the need for the extensive analysis presented here.

The aerosol backscatter profiles show a consistent decrease with altitude through the boundary layer and across the top of the boundary layer and (former) cloud top. This is consistent with a surface source of aerosol. A surface source such as wind-blown dust would include some quite large particles with a significant sedimentation velocity, this would result in the sort of decrease with altitude observed here. No modification of aerosol concentrations by cloud is required.

**Dynamics/thermodynamics analysis**

The analysis in figure 8 reveals an interesting difference in thermodynamic behaviour in the hours prior to cloud formation between summer months (May-August) and the rest of the year. In the summer a decreasing trend in temperature (cooling) prior to cloud formation is accompanied by a decrease in dew point suppression – an increase in relative humidity. No such association is found for the rest of the year, where dewpoint suppression is more or less constant regardless of trends in temperature. The potential link to cloud formation in the summer is clear – increasing relative humidity will eventually result in saturation and condensation. The lack of change in dew point suppression in winter is ascribed to the cooling temperature trend resulting from advection (of increasingly dry air) rather than local cooling. No additional evidence is provided to support this supposition, and it is not clear why there should be a seasonal separation between local cooling and advection of cooler airmasses. Another possibility is that during the winter months the temperature is below freezing and the humidity of air is controlled by the saturation vapour pressure with respect to ice not water. Cooling will enhance this, resulting in growth of ice/frost by vapour deposition and keep the relative humidity with respect to water suppressed.

It is not clear that radiative cooling at the surface will necessarily explain cloud formation – cooling at the surface will tend to lead to increasing stable stratification, suppressing turbulent mixing and keeping the cooling localised to a shallow layer close to the surface. Air aloft might remain unaffected and at constant temperature. Eventually we might expect cooling to result in fog formation, but the formation of an elevated low level cloud depends on more than just surface cooling – mixing sufficient to maintain a more or less well mixed layer that cools as a whole, and an adiabatic profile so that the upper part of the layer saturates first. No attempt is made to distinguish fog and elevated cloud layers in the analysis, although this would seem to be an important distinction from the perspective of the process for cloud/fog formation.

The analysis of geopotential layer thickness trends I find wholly unconvincing. The data points in Figure 10 are mostly very scattered, and in most cases it would be hard to make out a convincing trend by eye. A line can always be fit to the points, but does not imply a robust relationship.

Further, I have serious doubts about whether the calculated tendencies are meaningful, even on a case by case basis. The trends are calculated from 2 consecutive radiosonde profiles prior to the cloud transition. This means, usually, over a 12-hour interval. The example clear sky case shown in figure 1 is barely 9 hours long. The 2 closest sondes preceding the onset of cloud at the end of the clear event actually span the dissipation of the preceding cloud. The later of the two sondes is 1.5 hours after the dissipation, and about 7 hours prior to cloud formation. I would suggest that the geopotential height trend calculated here is more relevant to the dissipation event than to the formation event to which it is actually applied.

Given that we have both clearing and cloud formation both occurring within an interval less than that over which a single geopotential height trend estimate is calculated, that rather suggests that any correlation between the two is suspect at best, and potentially entirely spurious. To make a really meaningful evaluation a much higher time resolution is required for the geopotential height trends. Maybe the output from an operational forecast model would provide a better measure here.

**Detailed comments**

Line 57: Hines & Bromwich (2017, 10.1175/MWR-D-16-0079.1) also model this case, with similar conclusions to Birch et al.

Line 61: pedantic gammar point 'a myriad of complex processes' should be just 'myriad complex processes' (myriad = countless, so 'there are countless processes' not 'there are a countless of processes'. Or classically myriad = 10,000, with similar implications for the grammar)

Line 106: '…measures the number of particles present within a volume of air…' -> 'measures the concentration of particles…'

Line 173: the authors note how low cloud and fog can be distinguished here, but never use this to separate out the cases, which I think is relevant for some of the process identification.

Line 189: the authors note a peak in the variability in backscatter between a few 100 metres and ~1km. This is presumably a result of variability in BL top, and the associated gradient in aerosol & backscatter across it. This is not mentioned here, and throughout the discussion of figure 3 the profiles are discussed in isolation from any consideration of BL depth. I found this frustrating – there are several places where a feature of these profiles is discussed and some inference made, where my first reaction was that this was a result of variation in BL depth and this point was apparently being missed (see notes below). Same with figure 4. Only much later, at figure 5 is this point acknowledged, and profiles normalised to cloud top

height. Given the importance of cloud/BL top in relation to aerosol profiles I think too much is made of the results from figures 3 and 4, when it could be stated up front that to properly interpret the profiles they need to be plotted against altitude normalised to BL top – maybe both true and normalised heights are needed to fully interpret them, but the issue needs acknowledging up front.

Line 201: 'most obvious is a reduction in backscatter in November just before cloud formation (Fig. 4d)' – this doesn't apply at all altitudes, only 200-600m. This might result from, say, subsidence causing BL depth to decrease – change is then not in situ, but movement of layers. It is also not clear that this reduction is relevant to the subsequent cloud formation since we are given no information as to what altitude that cloud/fog formed at.
It is perhaps also worth noting that there are only 6 cases for analysis in November, so a single strong case may dominate the statistics.

Line 204: '*It is interesting that the level where backscatter transitions to its quasi-constant value is at or above where low cloud formation (base < 400 m or surface fog) occurred*'
a) this is exactly what we would expect for any scalar quantity with a surface source (eg water vapour in marine environment)…so reassuring rather than interesting?
b) to properly assess this you need to plot against a normalised altitude – you know where cloud top was/will-be so don't need to approximate to 'at or above where low cloud occurred'.

Lines 206-209. "*Consistency in aerosol backscatter structure from start to end of these clear periods seems to mimic the behaviour of a residual layer of relatively well-mixed aerosol trapped across the lowest few hundred meters of the atmosphere. This mixed layer may have been an artifact of the previous sub-cloud mixed layer prior to dissipation.*"
a) it is not clear what altitude the authors refer to here – assuming they refer to the 'quasi constant' value from 2 lines up, then they refer to the layer above the BL/cloud, i.e. in the free troposphere. Here aerosol profiles depend mostly on advection and conditions upwind, perhaps far upwind. The reference to a previous subcloud layer then seems rather spurious. And again, you know where the cloud layer was (and will be) so you can pin point this, you don't need to speculate. Normalised altitudes would help again.
If the reference is really to within the BL, then this needs making clear.

Line 208: "*since the transition to a quasi-constant value is occurring at or above cloud base*" – physically we expect the transition to quasi-constant free-troposphere values at cloud top, the rather vague, and physically misleading, phrasing 'at or above cloud base' would be unnecessary if the profiles were assessed against a normalised altitude.

The following statement "*the data suggest that suface aerosol properties such as number concentration are likely often unrepresentative of aerosol properties at cloud level*" I agree with, but not because the '*transition to a quasi-constant value is occurring at or above cloud base*' but because there is a general decrease in backscatter with altitude in the lowest levels.

Line 224 & figure 5: only Feb-May are shown in figure 5 '*because these months had the most frequent clear sky periods*'. This is irritating, since it omits November, the one month in figure 4 which showed a behaviour distinct from the other months shown, and which might be explained by the normalised altitude used here. In general, given the very sparse data set, the limiting of data shown to specific months seems counter productive – better to use all of it all the time – combine months to reduce issues with poor stats in single months. Define

season boundaries rather than using whole months to better group consistent seasonal behaviour. If you insist on using only a subset, then at least be consistent and use the same subset throughout.

While the full 2D RFD in figure 5 is useful – it really highlights the variability and that this is clustered (on individual cases?) rather than uniform, it isn't easy to directly compare these plots with figures 3 and 4. The addition of median profiles would help.

Line 233: the words 'and above (fig. 5a-d)' don't fit grammatically with any of the rest of this sentence.

Line 237: '…*cutoff between aerosol and clear sky (Shupe, 2007)*' – here 'clear sky' appears to be being used to mean something different than every other occurrence…a complete (?) lack of aerosol? I would rephrase or risk this being interpreted as just 'cloud free'.

Line 241: "*Being that aerosol backscatter near and above cloud top (zn=1) was at a minimum suggests that low aerosol concentrations near cloud top could have played a role in its dissipation*" – only aerosol below cloud top are directly relevant to its properties, those above can't affect its microphysics. They can only play a role if entrained into cloud, but since the measurements are obtained after dissipation, aerosol above the former cloud top clearly were not entrained. This contradicts the statement on line 239 and is again contradicted (or at least…amended) on line 245.

Line 266-274: The discussion of aerosol concentration at the surface needs more nuance.
In the case of low cloud - formation should not impact aerosol concentration at the surface - CCN lifted above LCL will nucleate a droplet, but if the drop is moved down again it will evaporate leaving the aerosol particle - number of particles is conserved.
Loss of particles requires:
i) coalescence of droplets - evap would then tend to consolidate all the original aerosol into a single large particle.
ii) scavenging of aerosol by droplets - evap as in (i)
iii) precip - loss of CCN & scavenged aerosol to surface.
All these are possible, but not discussed.

In fog the CPC might undercount total particles, even when conserved, if droplets don't make it through the inlet into counter (quite probable).
Again, it would be useful here to distinguish between low (but elevated) cloud and fog.

Line 290: "*LWN is primarily proportional to cloud liquid…*" – only for liquid water paths below the black body limit of ~50 g m$^{-2}$, above that there is little impact on LW radiation.

Line 318: "*These seasonal and sky condition differences in particle concentrations suggest different mechanisms are responsible for aerosol numbers near the surface*" – this is interesting. Is this simply a result of having an exposed local surface during summer, which may be a strong source or aerosol, and a snow covered or frozen surface for the rest of the year?

Line 391: "*least squares linear regression of the tendencies between the layers reveal a moderate agreement to the monthly cases*" – 'with the monthly cases' or 'for' the monthly cases depending on your intended meaning.

Line 418: "*The statistical analyses presented fail to identify a definitive signal in aerosol vertical profiles indicating changes in aerosol partitioning are the primary cause for cloud dissipation.*" – poor phrasing, this is easily misread as meaning "*changes in aerosol partitioning are the primary cause for cloud dissipation*" rather than "*fail to identify a definitive signal in aerosol vertical profiles that would support changes in aerosol partitioning being the primary cause for cloud dissipation*"

Line 480: "*Here, a similar transformation process has been identified on the northern edge of NSA*" – I'm not sure one has been *identified*, only inferred as a potential mechanism.

Line 488: 'morphology' is not a verb!

Line 505: '*increased pooling of aerosol particles near the surface*' – I'm not sure that an 'increase' in pooling is demonstrated. And none is needed, concentrations rarely fall low enough for aerosol to limit cloud formation, so no pooling of aerosol is required to 'provide the ingredients' for cloud formation.

---

## Referee Comment (RC3) · Anonymous Referee #3 · 17 Sep 2020

In this study the authors attempt to explore reasons for dissipation and formation of low clouds in the Arctic, using a multitude of data from the ARM site in Utqiagvik (Barrow). They first isolate clear-sky periods using a ceilometer and refine these with additional data. They then proceed to analyze data from lidar aerosol backscatter and from in-situ surface measurements of aerosols, radiation and basic meteorology as well as indicators of atmospheric tendencies from soundings. They do this using composites of data for four years.

Their effort is ungrateful in the sense that it turns out to be very difficult to tease out any solid relationships. This is, while of course frustrating, in itself not a reason to reject

a paper; a negative result is also a result, and it all rests with how this is handled. However, the paper could be better organized and more clearly written. I recommend that the paper is accepted after major revision focusing more on the structure and language of the paper, more than on the results themselves.

Major comments: This is an original way to analyze data, and the approach is interesting. I commend the use of more than cases studies; while this is likely a reason for the lack of clear results, it represents a way to obtain more general results. Anyone can dig out a single case and speculate about reasons for a given outcome, but this is close to useless in a more general sense unless it can be shown that results are more general.

While this is a strong case for this paper it is also a bit of a weakness in the present manuscript. The background to the problem and the motivation for the method is presented in a very hand-waiving fashion; the current introduction reads more like a list of previous studies and suggestions than an organized argument. Many examples of suggested aerosol influence is listed, but isn't it quite clear why. While aerosols are certainly important, different clouds form mainly because of dynamics than by aerosol constraints. Different types of clouds form in different situations and differently at different locations because of different predominant dynamics; low clouds in the Arctic Ocean, frontal clouds in extratropical cyclones and deep convection in the tropics. All of this is modified but not determined by aerosols.

Hence, I wish that the authors more deeply criticize and discuss the problem of representatively, as a motivation to stay away from case studies, and then present more clearly the hypotheses they are attempting to test including potential effects of atmospheric dynamics. As it stands, I get the impression they throw whatever data they can lay their hands on, on this problem in the hope that something might show up. I also miss the motivation to why four years of data is used; why not five – or ten?

The paper – even its title– makes a big deal of the clear periods, but if one is interested in cloud dissipation or formation, presumably the happenings before and after

the shoulder times are the interesting things; not the clear period per see. Isn't the clear period in between in itself sort beside the point? Also, when clouds are dissipated, presumably new clouds will form at some later time, hours or days later; the formation of the new clouds at the end of the clear period may have absolutely nothing to do with the dissipation of the other clouds hours or days earlier. Calling these "bookends" is misleading in that the reader is lead to think of this as a coupled sequence of events; they may in fact be entirely different. Hence the focus should have been on either cloud dissipation or cloud formation – or both but separately – and then focusing on before and after cloud dissipation/formation.

This constitutes a problem with the lidar, since it is difficult or even impossible to obtain aerosol backscatter in the presence of low clouds, attenuating the lidar signal. This is just a fact of life and is discussed on lines 226-227, as in the passing; this information should be given and discussed up front. The results in Figure 3 should therefore be discussed in the context if being clear skies; not in the context of not being cloudy, since that contrast just isn't there. Of course it may still have some value to look at aerosol backscatter directly after dissipation and directly before formation in a statistical sense, as in Figure 4, but this caveat should be discussed up front; that the one set of plots represent after dissipation has happened while the other set is before cloud formation. Without knowing what the structure was before dissipation and after formation of clouds, the information value is limited. And BTW, is this really cloud dissipation/formation; isn't it just a hole in the cloud layer advected past the viewer? Maybe this is why its so hard to get statistically robust results?

At the end of the discussion section a hypothesis is formulated, almost like in passing; I'm sorry, but I don't get it. It builds on the Tjernström et al (2019) air-mass transformation hypothesis. But a central tenet in that hypothesis is the fact that over melting sea ice, the surface temperature is locked constant at the freezing point; here there is no analogy. So is cloud dissipation leading to surface cooling, then aerosol pooling, followed by fog formation, fog deepening and lifting to clouds? That would in essence

mean that cloud dissipation leads to cloud formation? If this chain of events is really happening, it should be a testable hypothesis; temperature should drop while aerosol concentrations rise with time, followed by fog formation and cloud base rising from zero to some height; in gact, the very same set of data used here could be used to test this hypothesis. Instead the hypothesis is not even clearly repeated in the conclusions, but brushed over with many words in paragraph two and beginning of paragraph three. If you want to pose a hypothesis, do it; else don't!

Finally, the language is sometimes what I would – in lack of a better description – call "flowery". It is important to have a capturing narrative, but unnecessarily complicated sentence structures sometimes lead to confusion and misunderstanding. So maybe sometimes be a bit less imaginative.

Minor comments

Line 28: Drop "even".

Line 29: Please rephrase; the temperature of low clouds do not reach "as cold as -34 °C" in "all seasons".

Line 14: Unnecessarily complicated. Suggest "While clear sky is less frequent than clouds" or even "While clear skies are rare".

Line 38: Lack of what? "longwave warming" or "Arctic clouds"?

Lines 39-40: Only true when the sun is absent or the albedo is high; over bare land and in summer, clear skies usually leads to a surface warming. Even in the Arctic.

Lines 41-44: A prime example of when there are too many ideas in the same sentence. Exactly what is it that "is currently understood". I know all this so I understand what you mean, but please rephrase anyway.

Line 43: "stratocumulus and also"

Lines 50-51: I would move up "in the Arctic" in that sentence, or it sounds like the

transition everywhere is controlled by Arctic clouds.

Line 59-60: So opaque liquid clouds would form out of what? Optically thin ice clouds?

Line 71: In what regard is that?

Line 71-72: This is a sentence where the narrative is that clouds dissipate and form at the beginning and end of the clear period, as if the dissipation and the formation where reverse analogs.

Lines 74-77: Here is a completely different take; now the formation clear period is at focus, not the dissipation of formation of the clouds.

Lines 106-107; what has "a diameter of 10 to 3000 nm"; the volume of the air or the partciles? I know the answer of course, but the sentence is rather unclear.

Line 107: Do all cloud-relevant aerosols absorb alcohol, or do we miss some?

Line 129: Grater than identically zero?

Line 136: How is the agreement on clouds between the ceilometer and the HSRL?

Line 146: I assume the base is at 100 m and the top is at 400 m; neither is between 100 and 400 m.

Lines 172-174: Another long sentence with more than one idea confusing the other. Is there any other way a clear period can end than by the emergence of a cloud? And is the ceilometer ever operating in anything but vertical mode?

Line 188: Not all months have a clear elevated "level of maximum variability". Figure 4: Why one hour?

Lines 226-227: This is really important information to have before looking at Figure 3 & 4.

Line 239: What type of aerosol particle would not come from "below"; what aerosols do not have an origin at the surface except for those emitted by aircraft?

Line 278: "agrees" with what?

Figure 6: Why now 2 hours; earlier it was one?

Line 279: You are not exploring "phenomena"; you are exploring variables and trying to infer "phenomena".

Line 325: "strongly transparent"? Better say "almost opaque".

Lines 342-344: Not sure I get this; if the dew-point deficit has a positive trend (is increasing) and the temperature has a negative trend (is decreasing), does that necessarily mean RH is increasing? Could the dew point not decrease so much more than temperature that RH stays constant or even decrease?

Line 383: "in flux"? Maybe chose a different wording?

Line 423: About the source of aerosols again; isn't this trivial? Moreover, I think aerosols are defined as "airborne ... particles" so there's one "airborne" to many here.

Line 424: "general stable stratification" is probably incorrect, or

Line 364-365: This is a bold sentence, supported by only one reference. I'm not necessarily disagreeing, but still.

Line 383: "in flux"; is this a good choice of words?

Line 423: Here are the aerosol sources again; I'm no expert but unless you emit them from an aircraft, don't they have to come from the surface?

Line 424: The statement on "general stable conditions" is probably inaccurate or at the very least debatable. Studies have shown that the most common near surface stratification over the whole year is near-neutral, but that stably stratified conditions prevail in clear conditions especially in the winter when they are also deep and strong. Additionally, is there no ground based convection over Alaska or at Barrow; I get over the ocean but this is on land?

Lines 485-489: Here's that hypothesis; I would have much liked to have the hypothesis at the front and the paper about testing it, or at the end as a bridge to the next study. Here it isn't even a conclusion; reading a bit hasty one could have missed it.

Line 511: Maybe avoid the word "transparent" in this context, as it is so intimately linked to other things in this manuscript.

---

## Referee Comment (RC4) · Anonymous Referee #4 · 22 Sep 2020

Review of "Processes contributing to Arctic cloud dissipation and formation events that bookend clear sky periods" by J. Sedlar et al.

This manuscript presents an analysis of the atmospheric state (including aerosol concentrations) right before and after the onset of cloudy and clear periods at Utqiagvik, Alaska. The main motive of the work is to understand the processes that drive low-level cloud formation and dissipation in an Arctic environment.

I find the overall aim of the study and the analysis of available observations interesting and commendable. However, it seems like the manuscript was put together a bit too hastily; the overview and connection to published literature could be expanded (in particular in terms of Arctic aerosols), the presentation of the instrumentation and methods needs more information and the discussion of the results lacks some clarity and depth. On the data analysis side, I also find some issues with the way that the aerosol data from the CPC are treated. As stated in the manuscript, the data from the CPC will give you the total aerosol number concentration, including aerosols down to 6 nm diameter. This is a problem, at least during summer, when the total aerosol number concentration is dominated by smaller aerosols (nucleation and Aitken mode), which have very little influence on cloud droplet formation. Relating the aerosol concentrations from the CPC with cloud formation is therefore dubious.

**General comments:**
- I would suggest that the authors are a bit more careful when they use the term "the Arctic" or when they refer to certain characteristics of "the Arctic". The Arctic is not a homogeneous region where clouds, meteorology and surface properties are the same. Many of the features that the authors mention, in particular in the introduction, may not be true for the lower-latitude parts of the Arctic and/or land areas. For example, are clouds ubiquitous over the whole Arctic during the whole year? Does the longwave radiation dominate the radiative energy budget everywhere and during the whole year? Under cloud-free conditions, does effective infrared cooling from the surface cause extremely cold temperatures everywhere? I am thinking for example of Siberia where you in the summertime can have very different conditions compared to over the Arctic Ocean.
- Related to the previous comment, how representative is Barrow as a station for "the Arctic" and the type of cloud formation/dissipation events that you study? I think that the idea that aerosols control cloud formation/dissipation has mainly (only?) been presented for high (>80°N) Arctic clouds, i.e. in pristine environments where (accumulation mode) aerosol number concentrations are extremely low. Utqiagvik (or Barrow) has rather high (accumulation mode) aerosol concentrations for an Arctic station (cf. e.g. Freud et al., 2017 or Schmale et al., 2018). It may still be an interesting place to study low-level cloud formation and dissipation, but perhaps not so much from the perspective of an aerosol-limited regime?
- The authors use CPC measurements to relate aerosol concentrations to cloud formation/dissipation events. Firstly, I think that the methodology related to the CPC measurements needs to be better explained. What air is pumped into the instrument? Is it "whole air", "cloudy air" or "clear air"? How are ice crystals and cloud (fog) droplets handled by the instrument? Is the air dried? Does the instrument have any detection limit in terms of number? Secondly, the CPC measures particles down to 6 nm (as stated by the authors). The Arctic is typically dominated by small aerosols in

summer (cf. e.g. Freud et al., 2017) but these small aerosols are not efficient cloud condensation nuclei. Figure 3 in Freud et al. shows that in summer, the accumulation mode particle concentration typically goes down drastically while the total concentration of aerosols goes up as new particle formation and growth controls the aerosol population. Why did the authors not use Scanning Mobility Particle Sizer (SMPS) aerosol size distribution or CCN measurements from Utqiagvik? I think these should be publically available (cf. e.g. Schmale et al., 2017).

- I find the discussion about the vertical structure of geopotential height and "synoptic activity" and their relation to cloud formation and dissipation events confusing. In Section 3 (lines 387-398), the authors say that "From May through summer, differential advection amongst the atmospheric layers becomes a more frequent occurrence." From this, they conclude that cloud dissipation events are often associated with baroclinic activity in summer. I would also assume then that the *synoptic activity* is more frequent in summer during cloud dissipation events. The same is also true for cloud formation events (lines 400-409); these are more frequently associated with synoptic activity in summer compared to winter. But in the discussion section, it is stated that (in association with cloud formation events) "Variable dynamics resulting in differential atmospheric advection is most prominently observed during the winter and early spring. Furthermore, in the conclusions, the authors state (in relation to cloud dissipation events) "While we report that all months are subjected to synoptic disturbances, the magnitude of the forcing is weaker during late spring and through early autumn than during winter and early spring."

**Specific comments:**
*Abstract:*
- Line 2: I would suggest reformulating the sentence including "…lack of downwelling…". It sounds like there is no downwelling radiation at all when the cloud is absent.
- Line 18: I am not sure why you emphasize the link to aerosol concentrations here? Isn't any general change in dynamics/radiative cooling more important?

*1. Introduction*
- Line 27: Are there any other studies than Shupe et al. (2011)? Would be interesting to know.
- Line 27: I suggest changing "These clouds frequently contain concentrations of both…" to "These clouds frequently contain both …".
- Line 54. "Simulations of Arctic clouds consistently show that over-abundant ice nuclei or ice crystal concentration can lead to cloud glaciation". I don't think this statement is completely true – it depends on what the authors mean with "over-abundant" and "Arctic clouds". There are several studies that show that mixed-phase clouds in the high Arctic only glaciate at extremely (i.e. unrealistically) high ice crystal number concentrations, e.g. Stevens et al. (2018), Loewe et al. (2018).
- Line 56: Related to the previous comment, I think a CCN-limited regime has only been suggested for high Arctic clouds?
- Line 61: In this paragraph, it could perhaps also be worthwhile considering the studies by Young et al. (2018) and Dimitrelos et al. (2020) where they point out the importance of large-scale divergence/convergence (and associated free tropospheric moisture supply) in governing the lifetime of Arctic low-level clouds.
- Line 75: When reading the introduction, I was wondering why you focus on atmospheric properties "after cloud dissipation". It would have made more sense to look the atmospheric state before cloud dissipation. In the methods section you then

explain why this is not possible, but I think it could be good to include a short explanation already in the introduction.

*2. Instruments*
- Line 91: The description of the HRSL is very brief and should be expanded. For example, what is the detection limit of the lidar? Is there a limit in terms of how close to the surface the signal can be trusted?
- Line 1010: How small concentrations of small cloud droplets can the cloud radar observe?

*3. Methods*
- General: it would be nice to have a map of the location of the station and also a brief description of the typical conditions (closeness to sea, potential pollution sources etc.)
- Line 130: I'm just curious, why 96%?
- Lines 138-140: I suggest replacing the word "when" with "if".
- Line 146: Why show times as UTC and not local times? Would make it easier to interpret the radiative fluxes.
- Line 154: It is not completely evident to me that the mixed layer (elevated aerosol backscatter) is shallower during the clear period. How do you see this? Maybe it would help to draw a line at the start of the clear and cloudy periods?
- Line 155: "Evolution in near-surface meteorology showed modest changes…". I interpret "modest" as "not pronounced", but maybe this is not what the authors mean. I would say that the change in wind direction is fairly pronounced at the time of cloud formation? And also the change in dew point temperature?
- Line 157: It is quite interesting that the particle concentrations increase so dramatically during the clear period. In summer, new particle formation and/or condensational growth of nucleation mode particles often takes place when there is sunlight and (initially) low background concentrations of aerosols (e.g. Freud et al., 2017). Could this be what is happening? Was this a typical pattern or only a one-time feature? Important here is of course also what air the CPC samples, if it is "whole" air or only cloud-free air.

*4. Results*
- Line 165: Just out of curiosity, was there any difference in length of the clear periods between the seasons?
- Line 170: I assume that the clouds with bases below 400m also could include other clouds than fog and low clouds? For example nimbostratus, cumulus and cumulonimbus.
- Line 188: What is the "1-sigma envelope"?
- Lines 190-194: I have several questions/comments regarding this paragraph.
    - When is the boundary layer backscatter (which should be dependent on the aerosol surface area, so mainly the accumulation mode) the highest/lowest? How does this agree with other in-situ measurements of CCN and/or aerosol size distribution measurements (e.g. Freud et al., 2017; Schmale et al., 2018; Schmeisser et al., 2018)
    - Is it really true that the "transition layer" is the shallowest in summer? October and September looks pretty shallow too?
    - I don't understand the sentence that begins with "Many processes may contribute to …". Shouldn't this layer just be a result of the vertical depth of the boundary layer/mixed layer?
- Line 213: The limitation of the HSRL should be mentioned in Section 2.

- Line 214: Can you really draw this conclusion from looking at averages? I would think that in order to make this statement, you would have to look at the individual profiles and make sure that the transition layer is always below cloud or within the cloud that the clear-sky period bookends?
- Line 221: The selection based on a maximum cloud top height below 2km makes sense and should be done from the beginning.
- Line 236: The cutoff backscatter values should be mentioned in Section 2. But I am also wondering what the authors mean with "clear sky"? I assume there should still be aerosols present, it is just that the instrument cannot detect these low concentrations?
- Line 241: What do the authors mean with the sentence "Being that the aerosol backscatter… was at minimum…"? Where and how do you see this?
- Line 241: Related to the comment above, how low backscatter values would you need in order to have accumulation mode aerosol concentrations below ~10cm-3?
- Line 248: Please define "RFD".
- Lines 257-260. I do not think this argument holds. The backscatter will be dependent on surface area. If the aerosol population is dominated by small particles in summer, then the surface area will not be at its maximum, see also Freud et al. (2017).
- Lines 269-271: This results is interesting as the increased number of particles in spring/summer could be due to new particle formation and growth during clear periods, please see previous comment (Chapter 3, line 157).
- Lines 271-274: Does the CPC measure "whole air" or only "clear air"? If it is "whole air", then why would the concentrations decrase?
- Line 290: I do not think this argument is true. The downwelling LW should also be dependent on the temperature, in particular if the LWP is larger than ~20gm$^{-2}$ (emissivity close to 1).
- Line 293: How is the analysis affected by any presence of a stable surface layer (boundary layer decoupling)?
- Line 297: I think it should be mentioned in Section 3 that you use the soundings to calculate LTS.
- Line 300: Related to figure 7, why is the cooling generally smaller with more stable stratification (for clear sky)?
- Line 318: Which mechanisms are you referring to?
- Line 342: So this means that in summer you mainly have fog formation due to radiative cooling?
- Line 356: Are these results then inconsistent with the geopotential tendencies where you concluded that synoptic activity was more frequent in summer and spring during cloud dissipation events (lines 395-398)?
- Line 365: For the analysis of geopotential tendencies, I think it could also be interesting to look at these from the perspective of large-scale subsidence and convergence as in Young et al. (2018) and Dimitrelos et al. (2020). It would also be interesting to look at vertical profiles of moisture to see if the layer right above the cloud is a source or sink of moisture.
- Line 372: I would suggest inserting a "vertical" before "structure".
- Line 380: How much was the number of cases reduced?
- Line 401: You mean in late spring/summer…?

*5. Discussion*
- Line 430: I am not convinced that differences in horizontal advection is the main reason for the differences in vertical distribution of aerosols, see e.g. Freud et al. (2017).

*6. Conclusions*
- Line 499: I thought the forcing from synoptic disturbances was stronger in late spring through summer (lines 395-398)?
- Line 511: I guess there is also a possibility that the cloud formation and dissipation events does not happen "in-situ" but rather that transport of clouds (and clear air) contribute to the observations made at Utqiagvik?

**References**

Freud et al. Atmos. Chem. Phys., 17, 8101–8128, 2017 https://doi.org/10.5194/acp-17-8101-2017.

Young et al. Atmos. Chem. Phys., 18, 1475–1494, 2018 https://doi.org/10.5194/acp-18-1475-2018.

Dimitrelos et al., 2020. Journal of Geophysical Research: Atmospheres, 125, e2019JD031738. https://doi.org/10.1029/2019JD031738.

Loewe et al. Atmos. Chem. Phys., 17, 6693–6704, 2017 https://doi.org/10.5194/acp-17-6693-2017

Stevens et al. Atmos. Chem. Phys., 18, 11041–11071, 2018 https://doi.org/10.5194/acp-18-11041-2018

Schmale et al., Sci Data 4, 170003 (2017). https://doi.org/10.1038/sdata.2017.3.

Schmale et al. Atmos. Chem. Phys., 18, 2853–2881, 2018. https://doi.org/10.5194/acp-18-2853-2018

Schmeisser et al. Atmos. Chem. Phys., 18, 11599–11622, 2018. https://doi.org/10.5194/acp-18-11599-2018

---

## Author Comment (AC1) · 24 Nov 2020

This manuscript describes an analysis of clear-sky periods following cloud dissipation and prior to cloud formation over Utqiagvik, Alaska, with a focus on low clouds. The au- thors use a rather comprehensive set of ground-based measurements spanning over 5 years to draw insights on the processes contributing to cloud formation and dissipation. Different clear-sky period properties as a function of season are examined, from which the authors suggest differences in the impact of synoptic-scale forcing on cloud forma- tion and dissipation. The authors also find that a scarcity of aerosol particles is likely not the dominating cause for cloud dissipation, and postulate based on their analysis and the literature that cloud formations from late spring to early autumn largely initiates at or near the surface.

The manuscript is generally well written and I found the analysis description intriguing. I appreciate the amount of information that the authors were able to extract from the ground-based measurements and think that this manuscript provides a new analysis of an atmospheric phenomenon, which is often left without being properly examined, namely, clear sky periods. While I agree and/or find sense in most of the authors' interpretation and conclusions, I have a few concerns regarding the methodology and the analysis description, as well as a high number of rather minor comments, which I think the authors should address before this study can be accepted for publication in ACP. I, therefore, recommend major revisions.

We wish to thank the reviewer their detailed revision of our manuscript. As you will find below, we have intently considered each of the reviewer's criticisms, comments and suggestions. We have provided detailed responses to each of the specific comments below (in red).

Major comments: 1. Definition of a strict clear-sky period – I find the methodology rather robust. However, to my understanding, once the 2-h clear-sky threshold is met, intermittent clouds can be detected, as long as the total duration of cloudy periods does not exceed 4%, for example, in the case of a 10 h clear sky period, the last hour may contain the only 24 minutes of (broken) clouds. A similar example is provided in Fig. 1, in which I cannot agree with the authors' description in the text (l. 145-157) that the pe- riod between 10:30-12:45 UTC is strictly a clear sky period; that is, the KAZR, HSRL, and LW measurements all suggest the intermittent presence of a cloud layer (e.g., at ~11:30 UTC), obviously a tenuous one, and hence, the weak LWN signature, but this is still definitely a cloudy period. Now, I understand that the data analysis here requires a binary definition of either a "clear" or "cloudy" period and that an addition of an in- termediate class period would likely introduce multiple inconsistencies. However, with the current methodology settings and constraints (duration of a clear/cloudy period, al- titude limit for cloud occurrence, etc.): a. The clear sky and cloudy period portioning results in a very high overlap with the "radiatively clear/cloudy" states coined by Stram- ler et al. (2011), which is essentially the only way to argue that in Fig. 1 there is a 9-h long clear-sky period rather than ~7.0-7.5-h period. The authors should address this point here and other places in the text where it is applicable.

We understand the reviewer's concern, and we agree with their reasoning. It is true that intermittent cloudiness, by our definition of a "clear period", is allowed to emerge sporadically yet still be considered a clear period. We are looking for consistent cloudiness as observed by the vertically pointing remote sensors to identify the start and end points of a derived clear period. While we understand this is in no way strict, it was necessary in order to have any data points for our study. Further, we use the suite of instruments including the KAZR, HSRL and ceilometer to identify any instances, flag these times, and remove them from further analysis, in the time periods of analysis before or after a dissiapation/formation event. However, we still retain the original start and end points of the event because the general criteria for a clear sky period have been satisfied.

We have revised the wording around the duration of the clear sky period shown in Fig. 1, as suggested by the reviewer.

b. Clear sky periods can actually be cloudy, so I think that the authors should omit the use of the "strict" clear sky period definition throughout the text, which can become rather subjective, among other reasons, because of the multiple variable thresholds in this study (e.g., one could argue that only the periods where the thresholds mentioned in l. 236-238 correspond with a "strict" clear sky definition). I recommend the authors to consider terming the clear sky periods in conformance with their effective partitioning of the dataset, for example, I would suggest using the term "prolonged clear sky periods" (corresponding to the duration requirement while remaining objective by not introducing subjective criteria), which precede/follow "persistent cloud occurrence periods".

As the reviewer suggests, we have removed the description of clear sky periods as "strict". We agree that the original wording was confusing.

2. Inconsistent duration thresholds - why were 2 hours used for the analysis in fig. 6 (CPC counts), unlike the rest of the data analysis? How much would the results change with 1 -hour windows? Why were 4-hour windows used in the temperature trend analysis (sec. 4.3.2) instead of consistently working with 1-hour windows? Also, could the 1-hour window allow some separation of fog events (necessarily positive Td depression) from other low cloud events (potentially all depression values possible)? For both the CPC and temperature trend analyses, could there be an influence of intermittent cloudy periods just before (after) cloud dissipation (cloud formation) being classified as part of a clear sky period? How different do the scatter plots look using 1-hour windows instead of the utilized 2/4 hour duration windows?

In the revised analysis and manuscript, we have changed the results to look at either 1-hr or 2-hr time windows around dissipation/formation events for all analyses except for the large-scale layer thickness tendencies (4 hr). We kept the 1 hr period for the HSRL analysis because this analysis was focused on characterizing the vertical distribution of aerosol backscatter very near in time to the actual cloud dissipation or formation event. The intention was to identify if sharp gradients in backscatter (interstitial aerosol layers) were present and could be connected as an important mechansims contributing to cloud dissipation of formation. The HSRL also provided the highest frequency output (30 s) of the datasets analyzed, and we did not want the characteristics of longer duration clear sky period to emerge in the frequency distribution profiles. All other datasets had a 1-min and in order to compute meaningful tendencies, we extended the analysis period around a cloud lifecycle event by another hour to get a similar number of temporal data points as the HSRL. We do not have a legitimate motivation as to why we chose 4-hr windows for temperature trends; however in the revised manuscript, we have used 2-hr windows prior to cloud formation. Tendencies for layer thicknesses use 4-hr windows because we now rely on 1-hr ERA5 reanalysis to compute the tendencies (see specific comments below).

3. Synoptic forcing methodology and analysis: a. I understand that quasi-geostrophic flow occurs further away from the surface, but if largely low clouds are examined here (cloud base up to 400 m AGL in the main data subset; 3 km for the full dataset), why aren't tendencies in a near-surface layer thickness (e.g., 850-950 hPa) examined here (using a similar or different methodology), being more representative and consistent with the presented analysis thus far? It is not obvious to me how much analyzing such a low atmospheric layer could impact the results and discussion throughout this section, for example: - The conclusion that cloud dissipation events are impacted by relatively homogeneous thermal advection across the lower to mid-troposphere (l. 392-393). - "we identify that the height level where cloud formation events occur may be influenced by a weaker synoptic setting from December through May" (l. 413-414). - "Larger-scale differential advection is almost always ongoing prior to cloud dissipation, and as such it is assumed that different air mass origin and thermodynamic properties are likely to go in unison with changing aerosol properties" (l. 427-428).

The choice of these atmospheric layers are motivated by the fact that we are interested in understanding the background state of the atmosphere. Following synoptic forecasting guidelines (https://www.weather.gov/source/zhu/ZHU_Training_Page/Miscellaneous/Heights_Thicknesses/thickness_temperature.htm), the 500-700hPa/700-850 hPa layers are frequently analyzed in terms of the their thickness tendencies to characterize differential advection. Understanding how the

thickness of these 2 layers covaries was the primary purpose of this analysis. As for the detailed sentences highlighted: these have all been removed in the revised manuscript because of the updated figure and analysis surrounding it.

b. Point measurement nature of the ground-based data influencing the interpretation - I understand that the results indicate a consistent/inconsistent wind regime in different seasons (e.g., l. 356- 360). Could the consistency of the low-level wind direction examined here be the result of a strong micro-meteorology, e.g., prevalent sea-breeze over Barrow (during sunlit periods; hence, the narrowest distribution during summer when SZA is lowest), which masks synoptic forcing, which could still have a significant influence on a mesoscale? By the same token, high variability during winter is influenced by the synoptic-scale flow (e.g., l. 353-354), but that signature could be enhanced (relative to other seasons) by weak/lack of micro-meteorological sources (e.g., during dark periods). This is an additional degree of freedom in the data that the authors need to consider (e.g., using reanalysis or nearby surface stations) in order to support their conclusions in l. 500- 502, 506-508.

The reviewer raises a number of valid arguments, and the simple answer is unfortunately we do not have the data sets to address them. We agree with the reviewer that reanalysis or a spatial analysis may provide some insights to these questions. However, this alone would be easily enough material to comprise a separate study. We feel that the results and conclusions drawn in this paper about the mechanisms contributing to cloud lifecycle changes at Utqiagvik provide a baseline and a framework for further synoptic analyses. Our main conclusion is that finding the atmosphere to be in a complete steady state is rare, and therefore idealized modeling studies only exploring the changes to cloud from aerosol microphysics is not relevant on the North Slope of Alaska.

4. Given the relatively small effective dataset, statistical significance tests could have a large impact on the discussion and interpretation of the results. The authors should perform such tests and refer to them throughout the discussions for which they are relevant (e.g., l. 262-264, l. 360-362, l. 391-392)

The reviewer raises a very valid concern, and we appreciate the suggestion. To examine the distributions of number concentrations from the CPC prior and post cloud dissipation/formation events, we have performed a Wilcoxon rand-sum significance test. This test is used to determine whether the null hypothesis that the median values amongst two populations can be rejected.

For the original dew point depression and temperature trends analysis, a Pearson correlation coefficient and associated p-value for the relationship is presented. These include corrrelations for all cases, low/fog and fog only cases.

For the geopotential height tendencies, 4 hr trends from consecutive 4 hr periods within a month of in geopotential height between the two atmospheric layers during the 5 year period have been computed from ERA5 reanalysis. From these monthly trends, monthly mean and standard deviations in the 4-hr trends are used to identify when 4-hr trends prior to (post) cloud dissipation (formation) events were within, or exceeded, +/- one standard deviation of the seasonal mean. Including the stasistics of seasonal variability highlights our original finding/conclusion that winter and spring cloud dissipation/formation events occur more frequently for larger forcing (geopential height tendencies) than during summer and autumn. During summer, nearly all the low cloud/fog formation events occur in coincidence with small geopotential tendencies.

Minor comments: - There is an occasional change of tense throughout the manuscript (e.g., l. 16-20, 196-215, 311-319, 387-398). I recommend the authors to be more consistent from this aspect, as I think that it improves the manuscript's readability.

- "Barrow" should be replaced with "Utqiagvik" throughout the text (except for in the abstract and introduction).

We have replaced Barrow with Utqiagvik, as suggested.

- l. 24 – Because the essence of this first sentence is elaborated below, I suggest removing the first reference or adding a few more references (e.g., Curry et al., 1996, https://journals.ametsoc.org/jcli/article/9/8/1731/36313), as Herman and Goody (1974) only discussed summertime clouds, in which SW radiation plays a role.

Changed as suggested.

- l. 27 - add "water" before "particles"

We have changed the text to "...water and ice particles..."

- l. 96 - suggest modifying "the signal becomes" to "the signal typically becomes"

Changed as suggested.

- l. 101 - I do not think that this is necessarily true over the Arctic. There are numerous examples of cases in which the droplet size and/or concentrations are too small to be detected by ground-based Ka-Band radars such as KAZR (e.g., first hour in Shupe, 2011, fig. 1, where the radar echoes are below cloud base). I agree that voxel-wise KAZR is indeed capable of detecting (in its high-sensitivity mode) most hydrometeor echoes, but many tenuous liquid-bearing clouds (which are common in the Arctic) can remain undetected by radar. I suggest rewording this sentence to address this general misconception.

Following the reviewer's concern, we have updated the sentence to the following:

The millimeter wavelength (35 GHz) provides high sensitivity and signal to noise ratio allowing the radar to observe cloud droplets, although some may be missed (de Boer et al., 2009).

- l. 113-115 - Note that Long and Turner (2008) only analyzed the downwelling LW and only during clear-sky periods, in which the downwelling fluxes are relatively lower, and found that the 4 Wm^-2 value holds for only 2/3 of the NSA cases. The LW flux uncertainties are likely larger and contain a flux percentage component, as also suggested by the ARM handbook for these pyrgeometers (see Table 6 in https://www.arm.gov/publications/tech_reports/handbooks/sirs_handbook.pdf). I rec- ommend the authors to update this discussion accordingly.

The uncertainty estimate has been updated as listed in the SIRS handbook provided by the reviewer, thank you.

- Fig 1. – The title for panel a says "MMCR reflectivity" instead of "KAZR reflectivity".

Changed as suggested.

- l. 147 - following major comment #1, suggest changing to "nearly 7 hours" or elaborate accordingly.

Updated as suggested by the reviewer.

- l. 191-194 – This sentence is slightly confusing. I suggest rewording and or breaking it in two.

This section has been updated during the revision to better reflect the seasonal variability and how its magnitude varies with season.

- l. 165-166 - I suspect that these are 24 individual periods per month? Or is it a MAM seasonal mean? In which year?

The reviewer is correct. Figure 2 identifies the total number of clear periods during a particular month over the 5 year period. The text has been rephrased to clarify this point.

- l. 187-189 - This is hard to interpret in the logarithmic scale used in Fig. 3.

Please see response to the next comment below.

- Fig. 3 - Because the aerosol beta signal is largely concentrated on a single order of magnitude, I think that the logarithmic scale on the x-axis makes the figure more difficult to interpret, especially with regards to the shaded sigma, which may become misleading depending on the mean value. I suggest setting the x-axis scale to linear and/or plotting profiles of the SD absolute value and/or fraction relative to the mean value.

Considering the reviewer's suggestion here, as well as the in the comment prior, we have changed the x-axis to linear scale.

- l. 189-190 - I cannot agree that this is the case in some months, e.g., DEC, FEB, MAR.

In the revised manuscript, the statements that caused the reviewer to disagree have been removed.

- l. 190-191 - Difficult to say that for panel f

This statement is no longer included in the manuscript.

- l. 192 - "depth of the enhanced" –> "depth of an enhanced" - l. 204 - suggest removing "It is interesting that"

These changes have been made as suggested.

- l. 212 - "therefore impossible" –> "therefore it is impossible"

This statement is no longer included in the manuscript.

- l. 213-215 - To my understanding of the text description, if the transition phase (sloped part of the profile) occurs at or above cloud base levels, then this state- ment doesn't hold, because there is a certain depth (either just at cloud base or above) where the aerosol profile appears to be more similar to the aerosol pro- file below, as also suggested from Fig. 1, so based on the HSRL beta mea- surements alone, I could argue that the aerosols in the cloud layer are very simi- lar to those at the surface and that the surface is actually representative. I agree that the surface aerosol properties are likely often unrepresentative of aerosol prop- erties at the cloud level, but only because of previous dedicated aerosol stud- ies (some with in-situ measurements) and because surface or near-surface inver- sions/stable layers are so common over the Arctic (e.g., Tjernström and Graversen, 2009, https://rmets.onlinelibrary.wiley.com/doi/abs/10.1002/qj.380).

We agree with the reviewer, in particular the wording in the original manuscript was confusing. This has been addressed by updating the figure which includes both normalized height (by the low cloud formation top height) and the full profiles up to 1.5 km. The lines the reviewer is referring to are not included in the revised manuscript. Instead, the new figure with profiles normalized to the top height of following low cloud layer formation (a-d) now show the aerosol backscatter 1)

decreased up to the cloud top normalization height; and 2) from spring through summer the new figure shows that aerosol did not increase compared to the magnitude in the same height layers after cloud dissipation. From this analysis, we conclude that interstitial aerosol being advected across the lower troposphere were unlikely to be processes supporting low level cloud formation. The text has been revised to reflect these changes.

- l. 217-219 - That is a nice and significant observation. Do the authors think that this conclusion holds for other Arctic regions?

We agree in that the similarity in HSRL backscatter suggests that aerosols remained present and they did not drop to a concentration critical for sustaining cloud. This has been emphasized in the revised manuscript surrounding the updated figures. We believe that limiting aerosol concentration as a mechanisms for dissipating or inhibiting cloud formation are not present on the NSA, and likely only relevant over the central Arctic sea ice where a lack of particles sources exist.

- l. 221-222 - what is the time range before/following dissipation/formation that the authors use to determine whether the data corresponds with the 2 km threshold?

We use the 60 min window prior/post dissipation/formation to come to the mean cloud top height. This is now included in the revised manuscript.

- l. 226-227 - the lidar signal could be fully attenuated by cloud, but is not necessarily fully attenuated by cloud. Perhaps the authors can simply say that using a subset of cases without full lidar attenuation before dissipation/after formation would result in very few samples to analyze (number of examined samples is already rather low), also because of the data filtering (see major comment #1a).

The reviewer is absolutely correct. However, we have removed this discussion from the revised manuscript.

- l. 227-230 - Indeed, precipitating hydrometeors typically dominated the aerosol signal, but occasionally there are cases in which the ice number concentration is so low, that it is barely detectable in lidar measurements. I recommend the authors to add "typically" or "largely" to the text. Also, please change "drizzle droplets" to "drizzle drops".

Again, the reviewer is absolutely correct. We have removed this discussion from the revised manuscript.

- l. 241-242 - I am not sure this can be said without forward calculations of aerosol properties (given that the backscatter is proportional to the surface area, which is pro- portional to particle size in addition to concentration), which requires some information not available with this dataset. I would be hesitant to postulating that.

We agree with the reviewer's concern. In connection with Reviewer 2's comment stressing the importance of entrainment and the inability to assume entrainment is ongoing due to the lack of a cloud layer presence, we have removed this statement from the revised manuscript.

- l. 242-244 - such findings were first reported up to a few decades ago and should be cited here as well, e.g., Curry et al. (1996, https://journals.ametsoc.org/jcli/article/9/8/1731/36313) and references therein, Jiang et al., (2001, https://agupubs.onlinelibrary.wiley.com/doi/abs/10.1029/2000JD900303), Fridlind et al. (2012, https://journals.ametsoc.org/jas/article/69/1/365/27245).

We appreciate the additional references connected to importance of cloud top entrainment as a scalar source to the cloud layer. In lieu of the removal of role of free atmosphere aerosol source in

the revised manuscript, the inclusion of these additional references is no longer relevant to the revised manuscript.

- l. 244-246 - I cannot tell whether this sentence agrees with the data depiction in Fig. 5 because the cloud base height was not considered in the height normalization.

This statement is no longer included in the revised manuscript.

- l. 247 - by flatter do the authors mean less variable? - l. 248 - define RFD

Correct, we referred to a reduction in variability; this has been updated as suggested. The definition of RFD has been included earlier in the manuscript, as suggested.

- l. 248-249 – Not sure I understand the authors' intention here. Perhaps "typically agrees in magnitude with the profiles prior to cloud dissipation"?

This statement has been removed from the revised manuscript.

- Fig 5 and the associated discussion - Fig. 5 is hard to interpret because: a. There is no normalized height for cloud base. b. The small dataset combined with the inter- polated shading can be misleading. While I agree with the general conclusions of this discussion, currently it is rather difficult to evaluate and follow the different arguments. I recommend using larger RFD bin widths and adding a normalized height for cloud base (e.g., at 0.5).

We have addressed this by combined monthly data into seasons. This has increased the representativeness of the frequency distributions by including more data in each subpanel. We have also included the seasonal median profiles to help distinguish how a reduced number of cases (for example in JJA) influences the distributions. We decided to not normalize by the cloud base for two reasons: 1) We have extended the analysis original Fig. 4 to look at the profiles of seasonal median/$25^{th}$-$75^{th}$ profiles of all low cloud and fog cases, which in effect focuses the analysis truly on the lowest 300-500 m of the atmosphere; and 2) because cloud base heights, as discussed with the number of fog and low cloud cases in section 4.1 limited the number of valid cloud bases to normalize the height grid. Furthermore, the depth of the layer between observed cloud base and cloud top is relatively shallow in these low Arctic clouds. As such the normalization would become dominated by specific cases (i.e., the RFD would be biases by individual cases).

- l. 256-260 –The authors analyze here specific periods, which makes me wonder whether these seasonal patterns agree with the published literature of aerosol/CCN/CN seasonal variability over the NSA (e.g., Quinn et al., 2002, https://agupubs.onlinelibrary.wiley.com/doi/10.1029/2001JD001248; Lubin et al., 2020, https://journals.ametsoc.org/bams/article/101/7/E1069/345559), or are these seasonal signals muted in the bulk statistics reported in the literature?

Thank you to the reviewer for pointing us towards these relevant studies. We have included their citations as they agree with the general seasonality in number concentrations that we observed.

- l. 257-260 - This sentence reads awkwardly - suggest rewording.

During the revision of this section, this original sentence has been removed.

- l. 262-264 - Unlike the formation plots in panels e-h, I do not see any major differ- ence between these panels (a-b, c-d). I think that the authors need some statistical significance tests here to convince a reader (see major comment #4).

We agree with the reviewer that differences were sometimes difficult to assert in the original plot. While in the process of adding significance testing, we determined the original figure contained a bug. Instead of the bars showing the 25-75th percentile range, the 25th and 75th percentiles were being added/subtracted to the median values, essentially being treated as error bars. We have fixed this figure to properly show the interquartile ranges, and this helps to better distinguish differences amongst the distributions.

Further, we have included the 2:1 and 0.5:1 lines (in addition to the 1:1 line) to better identify visually how the median distributions have changed depending up time period prior to, or post, cloud lifecycle change. A Wilcoxon rank-sum significance test was performed to test whether the distributions that the medians are computed from were significantly different. Since the majority of the distributions were able to reject the null hypothesis that the distributions were equal at the 5% level, only cases where statistical significance was less than 5% were highlighted with a black marker edge color.

- l. 266 - define CPC

We have now defined CPC (as Condensation Particle Counter) in Section 2, Instruments.

- l. 276 - I do not think that the analysis failed to identify drastic signal changes, it simply did not identify significant changes, which I think is a nice observation on its own. I suggest rephrasing this sentence and remove "Therefore" in the following sentence.

We have followed the reviewer's suggestion and rephrased the opening paragraph of Section 4.3 as follows:

The previous analyses did not identify major changes in the vertical distribution or surface concentration of aerosols surrounding cloud dissipation and formation events. Increased surface particle concentrations prior to low cloud formation during summer were the most significant finding. The results imply that cloud-free periods may not be driven by significant changes in aerosol presence alone, consistent with conclusions drawn from an Arctic dissipation case examined in detail (Kalesse et al., 2016).

- l. 287 - "sub-cloud mixing driven by cloud-top turbulence" - suggest changing to "cloud-top radiative cooling" or "cloud-induced turbulence".

- l. 289-291 - the effective cloud temperature is rather important as well, i.e., the LWN is primarily proportional to the effective temperature differences between the cloud (cloud emissivity profile considered) and the surface, e.g., compare the LWN histogram in Stramler et al., (2011, https://journals.ametsoc.org/jcli/article/24/6/1747/32737) to that in Silber et al., 2019 (https://agupubs.onlinelibrary.wiley.com/doi/full/10.1029/2018JD029471).

We completely agree with the point raised by the reviewer. To address this, we have changed the text to the following:

…LWN is primarily proportional to cloud infrared emissivity (which asymptotes at liquid water paths between 30-50 g m$^{-2}$ (e.g., Shupe and Intrieri, 2004)) and the effective temperature difference between the cloud and surface, …

- l. 291-292 - Generally speaking, utilization of the equivalent potential temperature variable to examine static stability is only valid in moist processes (see Ch. 4 in Emanuel, 1994), e.g., in the case of a fog cloud extending from the surface up to 950 hPa. Otherwise, the virtual potential temperature or liquid water potential temperature should be used. Alternatively,

the authors may use the equivalent potential temperature only in liquid-containing heights. I suspect that addressing this comment as suggested above would not significantly change the results in this section, and therefore, consider this a minor comment.

We agree that equivalent potential temperature is valid in a moist environment. The Arctic is frequently very high in relative humidity, even though specific humidity magnitudes can be relatively low (e.g. Tjernström et al. 2012, ACP, https://doi.org/10.5194/acp-12-6863-2012). Further since the absolute humidity if relatively low, it typically has little influence on the equivalent potential temperature calculation. In this low absolute humidity environment, even if the equivalent potential temperature at 950 hPa changes by 1-2 degrees, this will not impact the LTS stability metric that we have computed. As the review mentions, the results would not significantly change, and therefore we have not changed the analysis to potential temperature.

- l. 302-310 - That is a nice discussion. The authors should also explain the apparent correlation between LWN and LTS (e.g., stronger stratification may indicate higher downwelling LW due to relatively higher temperature at 950 hPa or so, strongly de- pending of course on sufficiently high $q\_v$ at the 950 hPa level or so).

We are confused by the reviewer's request to explain the correlation between LWN and LTS. There are not many instances in the RFDs where the LTS is strongly positive and the LWN is relatively small, as suggested by the reviewer. In Sedlar et al. (2020), this mode in the RFD is discussed as being a consequence of warm, moist advection that becomes "trapped" in a vertical sense near the surface (e.g. Tjernström et al., 2015, GRL). Ultimately, we decided that this discussion does not add any scientific explanation to the major results of the figure, which are meant to show the separation between the "radiatively opaque" and "radiatively clear" modes in the LWN-LTS parameter space.

- l. 309 - suggest "warmer temperature" –> "higher temperature"

Changed as suggested.

- Fig. 7 & 10 - Please add the season to each panel, as already shown in other figures. It helps to follow the text.

To keep consistency throughout, seasonal titles have been included in the title of each panel, as suggested by the reviewer.

- Fig. 8 - suggest using constant y-axis limits - evaluation of the figure can be confusing at the moment (especially with regards to panel b).

Figure 8 has been updated in the revised manuscript, including using constant x- and y-axis limits, as suggested.

- l. 325 - suggest "strongly" –> "largely"

Changed as suggested.

- l. 331 - "Dew point depressions are then computed" - suggest rewording this part of the sentence.

This section has been revised, and this statement is no longer included.

- l. 358-362 - "insert/s" –> "inset/s"

- l. 367-369 - Isn't the moisture content a second-order term relative to mean temperature? If so, then I suggest focusing on the temperature or providing some observational evidence (e.g., via references), about temperature and moisture advection occurring commensurately.

In principle, the reviewer is correct. However, the modest specific humidity magnitudes typically found over the polar regions mean that only small changes to the absolute moisture can have a relatively large impact in the moist static energy through an atmospheric layer (e.g., Naakka et al., 2019, Int. J. Climatol. doi:10.1002/joc.5988)

- l. 372 - does "quasi-geostrophy" refers to "quasi-geostrophic flow"? Also, please provide a chapter or page number for Holton, 1992.

- l. 382-385 - I do not find this argument convincing (e.g., in panels b, f), and the current version of Fig. 10 where x- and y-axis limits are inconsistent between one-another and between panels depicts a misleading picture. Also, given that one should expect larger tendencies at higher altitudes (lower pressure levels), then what scale of tendencies in each layer would be considered small?

We thank the reviewer for suggesting to hold the x- and y-axis limits to the same bounds for each subpanel. Further, in light of Reviewer #2's concern with the geopotential height tendencies being computed from 12-hour soundings (the time scale is too long), we have used hourly ERA5 reanalysis profiles of geopotential height. From these data, we compute 4-hr tendencies (going back 4 hours from a dissipation or formation event) in order to increase the temporal resolution and focus on atmospheric dynamics in a time window closer to the actual dissipation/formation event. Because we are now using reanalysis data, in order to place the observed layer tendencies into context, we have computed consecutive 4-hr tendencies for all months within a season for the full five year period. This provides a measure of the variability of layer tendencies (a mean and +/- one standard deviation) with which we now compare the tendencies around a dissipation/formation event against.

- l. 387-389 - This sentence is rather confusing. I suggest rewording. Also, I think that labeling the figure quadrants might really help a reader follow this section and quickly understand that quadrant x represents thermodynamic structure change y, and so on.

The original wording was likely overcomplicated, as the reviewer indicates; in the revised manuscript, we have removed this statement. This section of the manuscript has been completely revised, including using ERA5 reanalysis to compute layer geopotential thickness tendencies computed over a 4-hr window prior to cloud dissipation or formation events. Application of reanalysis thickness tendencies provided more robust results than the nominal 12-hr radiosoundings prior to cloud dissipation and formation events. The better temporal resolution permits the focus of the synoptic activity, or lack thereof, on the actual cloud lifecycle event. From this analysis, we find a robust consistency across all seasons that represented a barotropic-type forcing amongst the 500-700 and 700-850 hPa layers; when plotting the two layer tendencies as a scatter plot, this barotropic signature emerges with a positively sloped relationship.

- l. 391-392 - with such a low number of samples, I would not consider r=0.44 or r=0.5 for that matter to represent moderate correlation (explaining not more than a quarter of the thickness co-variability). In any case, statistical significance for each month should be examined and provided here as well (see major comment #4).

We agree with the reviewer. However, the lack of correlation coefficients in the linear regressions are actually a consistent signature of one of the main conclusions of this analysis. Correlations were generally smallest during summer, especially prior to fog formation events. We assert that a lack of synoptic activity is an important mechanism that allowed the near surface to adjust

thermodynamically to the large net longwave radiation deficit, promoting saturation and fog formation. The small correlations are representative of the layer tendencies clustering around zero (the origin in Fig. 10), with no real relationship in the sign and magnitude of the layer tendencies. Further, to account for variability, we have computed the seasonal mean and one standard deviation (1-sigma) of the thickness tendencies for each layer (values exceeding the blue dashed lines in Fig. 10). This variability allows us to determine that winter and spring dissipation/formation events more often were associated with thickness tendencies exceeding the seasonal 4-hr climatology, indicating significantly large synoptic forcing was more common in winter/spring than summer.

- l. 395-398 and throughout the discussion concerning Fig. 10 - the authors should consistently use (in the text and figure) height or thickness; to my understanding, the authors only refer to thickness as a reference height is not provided.

We have revised the manuscript, including the figure caption, to consistently refer to geopotential thickness tendencies rather than height tendencies.

- l. 401-403 - Does this nice observation also agrees with the fact that frontal clouds are often observed during winter, as suggested by the wind direction analysis?

This statement has been removed in the revised manuscript, but we use the consistent results from the thickness tendencies together with wind direction changes during winter to argue that winter dissipation and formation events are influenced by larger synoptic activity than summer.

- l. 413-414 – I recommend replacing "weaker" with "shallower".

Shallower is not the context we were trying to describe, as this could easily be interpreted as a measure of a disturbance's vertical scale. Regardless, this statement is no longer included in the revised manuscript.

- l. 428 - suggest changing "is almost always ongoing" to "often occurs"

This statement has been removed from the revised manuscript.

- l. 431-432 - what do the authors mean by "aerosol vertical partitioning"?

Aerosol vertical partitioning refers to the general gradient structure of aerosol across the lower troposphere, being largest near the surface and generally decreasing with height. The vertical partitioning also describes any interstitial layers with enhanced or diminished aerosol backscatter signatures.

Following the reviewer's questioning of the terminology, we have replaced "aerosol vertical partitioning" with "aerosol vertical structure" throughout the manuscript.

- l. 452-453 - suggest removing "of the lower atmosphere".

This statement has been removed from the revised manuscript.

- l. 483 - redundant "the"

We have removed the discussion surrounding the air mass transformation process from the revised manuscript. Instead we focus on our main results that show a seasonal difference in synoptic forcing, near-surface cooling and transition towards saturation, and an enhancement in the particle concentration near the surface during summer – all results which suggest the summer, and to some extent spring, fog formation mechanism differ from that of winter.

- l. 483-484 - given the implemented methodology, the authors should explicitly state in this sentence that they refer to low-clouds (obviously, not necessarily fog).

See the response directly above.

- l. 514-515 - again, this is often the case, not always. As suggested above, the authors can say that when omitting the fully attenuated cases, the data subset becomes too small to be meaningful.

We do agree with the reviewer, but for the sake of clarity and the overall length of the manuscript, we have removed this discussion in its entirety in the revised manuscript.

- l. 519-521 - I would be hesitant to claim that the dominant Arctic cloud type following clear sky periods are low clouds, even though it could be tentatively suggested by the liquid/mixed cloud RFDs presented in Shupe (2011). My main concern here is about the subjectiveness of a clear-sky definition based on the methodology (see major com- ment #1b). To correspond with the methodology (e.g., 2-h clear sky and cloud occur- rence period thresholds) I could agree with "formation of persistent low-level clouds or fog, which have been shown in this study to be the dominant Arctic cloud type following prolonged clear-sky periods."

We respect the reviewer's concern regarding the general representativeness of our results. We have changed the text as the reviewer has suggested.

- l. 739 - Fig. 6 caption - should be (e-h)

Revised as suggested.

- l. 744 - RFD should be first defined in the caption of Fig. 5 where it is first mentioned. - l. 759 - insert –> inset

Both points have been revised as suggested.

- l. 764-766 - "e-g" –> "e-h"

Revised as suggested.

---

## Author Comment (AC2) · 24 Nov 2020

Review of "Processes contributing to Arctic cloud dissipation and formation events that bookend clear sky periods" by J. Sedlar, A. Igel, and H. Telg.

Ian M. Brooks

Overview

This paper aims to shed some light on the processes that control the dissipation and formation of low cloud in the Arctic. Such cloud is near-ubiquitous, but infrequent cloud-free conditions are important because of the large contrast in the surface radiation budget between clear and cloudy conditions. Models fail to adequately represent Arctic boundary-layer cloud and (operational forecast models) often fail to reproduce observed cloud free conditions. There is thus a definite need for improved understanding of the processes controlling these clouds.

The approach taken here is to utilise 5 years of measurements from a long-term measurement site at Utqiagvik, on the north coast of Alaska. The measurements include lidar backscatter (a proxy for aerosol concentration profiles), cloud radar, radiosondes, surface meteorology, and surface measurements of total aerosol concentration. Most of the analysis focuses on the ~1 hour period following cloud dissipation or preceding cloud formation that 'bookend' periods that are entirely cloud-free.

The analysis first considers the relationships between cloud dissipation/formation and aerosol profiles, comparing the profiles immediately after/before the transition with those for the clear periods as a whole (broken down by month), going on to consider the surface aerosol concentration either side of cloud transitions, and relationships between aerosol and lower tropospheric stability under both clear and cloudy conditions. This analysis provides no significant evidence for a causal link between aerosol properties and cloud dissipation/formation at the measurement site.

The analysis then considers thermodynamic and dynamic processes. This analysis leads the authors to conclude that "the onset of clear sky periods, and subsequently the end of clear periods, are primarily responsive to transient atmospheric forcing". For the onset of cloud they essentially conclude that under clear skies radiative cooling causes a fall in temperature and associated increase in relative humidity; ultimately saturation point is reached and provided there are sufficient aerosol present low cloud or fog will form. No firm conclusions are drawn about the processes resulting in the dissipation of cloud, other than the association with 'transient atmospheric forcing'.

These conclusions are rather generic and unlikely to help improve modelling of Arctic cloud.

The results remain of interest in providing a picture of typical conditions and some seasonal variations thereof, for periods of clear air bookended by low level clouds. There is considerable scope to improve this picture, however, and I recommend major revision before publication is considered.

We are grateful for the detailed review of our manuscript provided by the reviewer. We have responded with detailed replies to each criticism, comment and suggestion made by the reviewer below (in red).

General/major comments

While the aim of the paper is very worthwhile, I feel it ultimately fails to deliver robust conclusions. In part this is a, perhaps inevitable, result of the limitations of the data set. The aim is to understand what the processes are that lead to cloud dissipation/formation – transient events that are inherently linked to changes in local air mass properties over time. Measurements from a fixed site are, however, unable to distinguish between temporal evolution of the air mass properties resulting from in situ processes and the simple advection of a pre-existing spatial gradient in properties past the measurement site. This is a perennial problem for intensive, and/or long-term measurements. The authors attempt, but I think ultimately fail, to work around this problem by studying the statistics of an ensemble of cases. This provides correlations between measured properties associated with cloud transitions, and the hope is that probable processes can be inferred from these correlations. It is quite possible that observed behaviours might only be explicable by specific processes, and a fairly robust conclusion may be drawn. Sadly I don't think that is the case here.

As the reviewer understands, due to the sparse, detailed observing networks in the polar region, we are limited to specific locations or time periods to study processes criticial to cloud lifecycle changes. To avoid the trap of "case studies", we used 5 years of observations and statistical processing to identify features that are linked to the dissipation and/or formation process of clouds on the North Slope of Alaska (NSA). Using these statistics, we respectfully disagree with the reviewer about our study's lack of delivering conclusions. While cloud dissipation events have been studied in greater detail, the processes leading to the reemergence of lower tropospheric clouds has received considerably less attention. Following the reviewer's suggestions below, we have applied more focus on the separation of the type of forming cloud (base above 400 m, base below 400 m, or fog). This separation and evaluation of vertical aerosol distributions, near surface thermodynamics and winds, and larger-scale transient synoptic distributions has led to an understanding of forming Arctic clouds that has not been reported in the literature. While we cannot state that all the relevant physical processes have been explored, we have documented that on the NSA, the variation in aerosol has little impact on cloud dissipation; instead large-scale atmospheric forcing (exceeding the background seasonal variability in climatological forcing; revised figure and analysis – see detailed comments below and section 4.3.2) has not been reported previously; we feel this is an important result emerging from this study.

Aerosol Analysis

The analysis of links with aerosol properties is quite extensive, but ultimately finds no causal links with cloud dissipation/formation. The extensive initial focus on aerosol is (I assume) prompted by results from the central Arctic Ocean where very low aerosol concentrations (< 10 cm$^{-3}$) have been found to result in clear sky conditions even when the boundary layer is saturated, and several modelling studies have found that it is essential to accurately represent the aerosol in order to effectively represent the cloud and boundary layer structure. (as a side note, I find it odd that while the authors cite 3 modelling studies, all of which utilise the same observed case from the ASCOS project, they don't cite the original observational paper that first documented such CCN limited conditions and on which Sedlar is a co-author).

The CCN limited conditions in the central Arctic, are from a very different environment from the coastal site used here. The surface aerosol measurements in figure 6 and 7 show that concentrations rarely fall much below ~100 cm$^{-3}$, and are often much higher – far too high for aerosol to be the limiting factor on cloud formation. I think this possibility could have been ruled out much more easily by simply evaluating the surface concentrations (and perhaps relating them to the lidar profiles) for clear sky cases, without the need for the extensive analysis presented here.

The aerosol backscatter profiles show a consistent decrease with altitude through the boundary layer and across the top of the boundary layer and (former) cloud top. This is consistent with a surface source of aerosol. A surface source such as wind-blown dust would include some quite large particles with a significant sedimentation velocity, this would result in the sort of decrease with altitude observed here. No modification of aerosol concentrations by cloud is required.

We believe that it is important to document that aerosol processes controlling the dissipation of clouds over the central sea ice are, as reported here, very different than at the NSA. For this reason, we felt it necessary to highlight that cloud dissipation did not connect with aerosol changes. That because the surface CPC measurements remained high does not mean that changes in aerosol backscatter – proportional to the cross sectional area of aerosol concentrations – would not show indications of sharp gradients in the profile; for example a density gradient of enhanced/diminished aerosol backscatter across the boundary layer or above. This was not observed in the statistics. However, many of the LES and cloud resolving modeling studies referenced in the introduction attempt to emulate changes in background aerosol by varying CCN/IN numbers, conversion efficiency, and sedimentation processes through precipitation. The results from our study suggest that such processes are not of first order importance in determining whether a cloud should dissipate, especially during winter. However, during summer, cloud formation, especially fog formation, is frequently associated with relatively calm synoptic forcing. Enhanced concentrations of aerosol, with some particles still large enough to influence the scattering, (see responses to specific comments below and Figs. 6-7 in the revised paper) and therefore be efficient CCN should conditions permit nucleation to droplets, has not been reported previously. Our study found that the relatively calm synoptic forcing led to thermodynamic adjustment near the surface, in the presence of more particles, supporting fog.

Dynamics/thermodynamics analysis

The analysis in figure 8 reveals an interesting difference in thermodynamic behaviour in the hours prior to cloud formation between summer months (May-August) and the rest of the year. In the summer a decreasing trend in temperature (cooling) prior to cloud formation is accompanied by a decrease in dew point suppression – an increase in relative humidity. No such association is found for the rest of the year, where dewpoint suppression is more or less constant regardless of trends in temperature. The potential link to cloud formation in the summer is clear – increasing relative humidity will eventually result in saturation and condensation. The lack of change in dew point suppression in winter is ascribed to the cooling temperature trend resulting from advection (of increasingly dry air) rather than local cooling. No additional evidence is provided to support this supposition, and it is not clear why there should be a seasonal separation between local cooling and advection of cooler airmasses. Another possibility is that during the winter months the temperature is below freezing and the humidity of air is controlled by the saturation vapour pressure with respect to ice not water. Cooling will enhance this, resulting in growth of ice/frost by vapour deposition and keep the relative humidity with respect to water suppressed.

Following the reviewer's comment, we agree that the dew point depression was not the meaningful tendency. We have revised this analysis to explore the change in relative humidity with respect to temperature changes. We computed relative humidity with respect to ice for the months November through May, and with respect to liquid for June through October; we based these calculations on the mean monthly temperatures. The revised figure and analysis is a better method to explore

changes in both absolute humidity and temperature changes and removes the potential for the results to be controlled by vapor deposition to the surface during the very cold winter and spring seasons.

It is not clear that radiative cooling at the surface will necessarily explain cloud formation – cooling at the surface will tend to lead to increasing stable stratification, suppressing turbulent mixing and keeping the cooling localised to a shallow layer close to the surface. Air aloft might remain unaffected and at constant temperature. Eventually we might expect cooling to result in fog formation, but the formation of an elevated low level cloud depends on more than just surface cooling – mixing sufficient to maintain a more or less well mixed layer that cools as a whole, and an adiabatic profile so that the upper part of the layer saturates first. No attempt is made to distinguish fog and elevated cloud layers in the analysis, although this would seem to be an important distinction from the perspective of the process for cloud/fog formation.

Following the reviewer's suggestion, the revised figures and analysis surrounding them separates low cloud and fog cloud formation events. Please see the detailed responses related to this suggested revision below.

The analysis of geopotential layer thickness trends I find wholly unconvincing. The data points in Figure 10 are mostly very scattered, and in most cases it would be hard to make out a convincing trend by eye. A line can always be fit to the points, but does not imply a robust relationship.

Further, I have serious doubts about whether the calculated tendencies are meaningful, even on a case by case basis. The trends are calculated from 2 consecutive radiosonde profiles prior to the cloud transition. This means, usually, over a 12-hour interval. The example clear sky case shown in figure 1 is barely 9 hours long. The 2 closest sondes preceding the onset of cloud at the end of the clear event actually span the dissipation of the preceding cloud. The later of the two sondes is 1.5 hours after the dissipation, and about 7 hours prior to cloud formation. I would suggest that the geopotential height trend calculated here is more relevant to the dissipation event than to the formation event to which it is actually applied.

Given that we have both clearing and cloud formation both occurring within an interval less than that over which a single geopotential height trend estimate is calculated, that rather suggests that any correlation between the two is suspect at best, and potentially entirely spurious. To make a really meaningful evaluation a much higher time resolution is required for the geopotential height trends. Maybe the output from an operational forecast model would provide a better measure here.

We have considered the reviewer's comments and we fully agree with their concerns. To better capture the thickness tendencies that may have been connected to cloud dissipation or formation, we have analyzed the 1-hr profiles of geopotential height from ERA5 reanalysis. From these profiles, layer thickness tendencies were computed in the 4-hr period leading up to a cloud lifecycle event. The use of reanalysis allowed us to calculate the standard deviation, giving a measure of the climatological, seasonal variability in consecutive 4-hr layer thickness tendencies. We used this variability to quantify the seasons where cloud dissipation/formation events were associated with anomalously large thickness tendencies, suggestive of significant synoptic forcing.

Detailed comments

Line 57: Hines & Bromwich (2017, 10.1175/MWR-D-16-0079.1) also model this case, with similar conclusions to Birch et al.

This relevant reference has been added as suggested.

Line 61: pedantic gammar point 'a myriad of complex processes' should be just 'myriad complex processes' (myriad = countless, so 'there are countless processes' not 'there are a countless of processes'. Or classically myriad = 10,000, with similar implications for the grammar)

Noted and changed. Thank you for identifying this slip.

Line 106: '...measures the number of particles present within a volume of air...' -> 'measures the concentration of particles...'

Changed as suggested.

Line 173: the authors note how low cloud and fog can be distinguished here, but never use this to separate out the cases, which I think is relevant for some of the process identification.

We thank the reviewer for stressing this point. We have taken the reviewer's suggestion and included the separation between low cloud and fog cases in order to distinguish whether systematic differences in aerosol and meteorology can be linked to low cloud versus fog formation processes. We have also updated original Figure 2 to include the monthly number of fog formation cases.

Line 189: the authors note a peak in the variability in backscatter between a few 100 metres and ~1km. This is presumably a result of variability in BL top, and the associated gradient in aerosol & backscatter across it. This is not mentioned here, and throughout the discussion of figure 3 the profiles are discussed in isolation from any consideration of BL depth. I found this frustrating – there are several places where a feature of these profiles is discussed and some inference made, where my first reaction was that this was a result of variation in BL depth and this point was apparently being missed (see notes below). Same with figure 4. Only much later, at figure 5 is this point acknowledged, and profiles normalised to cloud top height. Given the importance of cloud/BL top in relation to aerosol profiles I think too much is made of the results from figures 3 and 4, when it could be stated up front that to properly interpret the profiles they need to be plotted against altitude normalised to BL top – maybe both true and normalised heights are needed to fully interpret them, but the issue needs acknowledging up front.

Original Figure 4 and analysis surrounding it has been updated, taking into consideration the reviewer's concern for variability in aerosol backscatter profiles associated with cloud top/boundary layer variability. The updated figure (see below) looks at backscatter for both the normalized heights (normalized to the median cloud top height observed within 60 minutes of low cloud formation) and the full profile up to 1.5 km for comparison. We find two interesting features in this updated analysis: 1. Aerosol backscatter is largest across the boundary layer/cloud layer, and decreases rapidly above the cloud top (seen also in the full profiles for e-h). This confirms our hypothesis that the primary aerosol concentrations emerge from near the surface and tend to be mixed within the rather shallow cloudy boundary layer. Thus, we find support for low cloud formation based on the profile of available aerosol 2. The backscatter across the soon to be cloudy boundary layer is similar, or slightly smaller, prior to cloud formation (blue) than shortly after the

cloud dissipated (black) for all seasons but winter (a). From this, we conclude it unlikely that advected plumes of increased aerosol concentration (increased to levels above those present shortly after the cloud dissipated) were responsible for supporting low cloud formation; or vice versa, that low aerosol concentration drove cloud dissipation. However, the median and interquartile spread in backscatter during winter is slightly larger where the low cloudy boundary layer would soon form.  We have included such a discussion in the revised manuscript.

[Figure]

Line 201: 'most obvious is a reduction in backscatter in November just before cloud formation (Fig. 4d)' – this doesn't apply at all altitudes, only 200-600m. This might result from, say, subsidence causing BL depth to decrease – change is then not in situ, but movement of layers. It is also not clear that this reduction is relevant to the subsequent cloud formation since we are given no information as to what altitude that cloud/fog formed at.

It is perhaps also worth noting that there are only 6 cases for analysis in November, so a single strong case may dominate the statistics.

The original figure has been revised to not examine 4 representative months, but to examine the seasonal profiles and their associated variability; monthly cases have been combined into seasons to improve the representability of the statistics The updated figure (above) and analysis around it now addresses the reviewer's concern.

Line 204: 'It is interesting that the level where backscatter transitions to its quasi-constant value is at or above where low cloud formation (base < 400 m or surface fog) occurred'
a) this is exactly what we would expect for any scalar quantity with a surface source (eg water vapour in marine environment)...so reassuring rather than interesting? b) to properly assess this you need to plot against a normalised altitude – you know where cloud top was/will-be so don't need to approximate to 'at or above where low cloud occurred'.

The statement regarding "It is interesting..." has been removed from the revised manuscript. The entire text surrounding the original Figure 4 has been updated, which now includes normalized altitude, as well as the full altitude profiles up 1500 m.

Lines 206-209. "Consistency in aerosol backscatter structure from start to end of these clear periods seems to mimic the behaviour of a residual layer of relatively well-mixed aerosol trapped across the lowest few hundred meters of the atmosphere. This mixed layer may have been an artifact of the previous sub-cloud mixed layer prior to dissipation." a) it is not clear what altitude the authors refer to here – assuming they refer to the 'quasi constant' value from 2 lines up, then they refer to the layer above the BL/cloud, i.e. in the free troposphere. Here aerosol profiles depend mostly on advection and conditions upwind, perhaps far upwind. The reference to a previous subcloud layer then seems rather spurious. And again, you know where the cloud layer was (and will be) so you can pin point this, you don't need to speculate. Normalised altitudes would help again.
If the reference is really to within the BL, then this needs making clear.

The original text was confusing and we understand the reviewer's concern. The figure has been updated by normalizing to the formation cloud top height level (panels a-d) and also shown as a function of altitude (panels e-h). The text related to the previous cloud driven mixed layer has been removed in the revised manuscript.

Line 208: "since the transition to a quasi-constant value is occurring at or above cloud base" – physically we expect the transition to quasi-constant free-troposphere values at cloud top, the rather vague, and physically misleading, phrasing 'at or above cloud base' would be unnecessary if the profiles were assessed against a normalised altitude.

We agree with this statement and understand the ambiguity that was introduced in the original phrasing. The new figure with profiles normalized to cloud top height now shows the transition in aerosol backscatter does occur above cloud top, as the reviewer indicates.

The following statement "the data suggest that suface aerosol properties such as number concentration are likely often unrepresentative of aerosol properties at cloud level" I agree with, but not because the 'transition to a quasi-constant value is occurring at or above cloud base' but because there is a general decrease in backscatter with altitude in the lowest levels.

We agree with the reviewer on this statement. The original paragraph has been removed in the revised manuscript, however we do include the discussion regarding the decreasing aerosol backscatter with height across the lower troposphere in the revised Discussion section.

Line 224 & figure 5: only Feb-May are shown in figure 5 'because these months had the most frequent clear sky periods'. This is irritating, since it omits November, the one month in figure 4 which showed a behaviour distinct from the other months shown, and which might be explained by the normalised altitude used here. In general, given the very sparse data set, the limiting of data shown to specific months seems counter productive – better to use all of it all the time – combine months to reduce issues with poor stats in single months. Define season boundaries rather than using whole months to better group consistent seasonal behaviour. If you insist on using only a subset, then at least be consistent and use the same subset throughout.

The reviewer raises a valid point. For the backscatter profile figures, we have now combined monthly data into seasons. All seasons have now been included in the figures and the analysis text.

While the full 2D RFD in figure 5 is useful – it really highlights the variability and that this is clustered (on individual cases?) rather than uniform, it isn't easy to directly compare these plots with figures 3 and 4. The addition of median profiles would help.

The updated figure now clusters the monthly data into seasons (to improve stats for limited number of cases in a single month) and the median profiles normalized to cloud top height have also been included, as suggested (magenta lines – see updated figure below).

[Figure]

Line 233: the words 'and above (fig. 5a-d)' don't fit grammatically with any of the rest of this sentence.

This text has been removed from the revised manuscript.

Line 237: '...cutoff between aerosol and clear sky (Shupe, 2007)' – here 'clear sky' appears to be being used to mean something different than every other occurrence...a complete (?) lack of aerosol? I would rephrase or risk this being interpreted as just 'cloud free'.

The text has been updated to state "...a threshold value determined as pristine (Shupe, 2007)".

Line 241: "Being that aerosol backscatter near and above cloud top ($z_n$=1) was at a minimum suggests that low aerosol concentrations near cloud top could have played a role in its dissipation" – only aerosol below cloud top are directly relevant to its properties, those above can't affect its microphysics. They can only play a role if entrained into cloud, but since the measurements are obtained after dissipation, aerosol above the former cloud top clearly were not entrained. This contradicts the statement on line 239 and is again contradicted (or at least...amended) on line 245.

This text has been removed from the revised manuscript because of the ambiguity introduced, as identified by the reviewer.

Line 266-274: The discussion of aerosol concentration at the surface needs more nuance.
In the case of low cloud - formation should not impact aerosol concentration at the surface - CCN lifted above LCL will nucleate a droplet, but if the drop is moved down again it will evaporate leaving the aerosol particle - number of particles is conserved.
Loss of particles requires:
i) coalescence of droplets - evap would then tend to consolidate all the original aerosol into a single large particle.
ii) scavenging of aerosol by droplets - evap as in (i)
iii) precip - loss of CCN & scavenged aerosol to surface.
All these are possible, but not discussed.

The reviewer raises a number of possible reasons that aerosol concentrations at the surface may decrease. However, as we are unable to test any of these possible processes, adding these possible mechanisms in the discussion to aid in determining the fate of CPC concentrations prior/post cloud lifecycle introduces additional speculation. With this figure and the analysis around it, we are able to show that CPC concentrations have a seasonal dependence around the dissipation/formation times. In particular, we find that aerosol is often as large, or even larger, after dissipation than before dissipation – which shows that aerosol are still present in terms of number. Secondly, we find that surface aerosol concentrations are considerably larger, often with median values twice as large, prior to low cloud formation, compared to after cloud formation during summer and into autumn. We find it likely that the larger concentrations of aerosol near the surface are likely contributing to efficiency of nucleation of a cloud drop.

Further, we have explored the optical properties (see figures below) of these near-surface aerosol prior to, and after, dissipation and formation events. We studied the 500 nm scattering coefficient measured from the nephelometer, as well as the Ångström exponent. Outside of summer, there distributions of scattering coefficient and Ångström exponent did not change systematically around cloud dissipation for formation. In summer, however, especially in July, an increase in scattering coefficient coinciding with a decrease in Ångström exponent was observed. The behavior suggests that the increased concentrations of particles observed during summer are not solely a response of very small particles formed from new particle formation events. Instead these particles appear to have sufficient size (and therefore mass) to provide a source of droplet nucleation.

[Figure]

Median and interquartile distribution of 550 nm scattering coefficient (Mm$^{-1}$) for 2 hours before and after cloud dissipation (a-d) and 2 hours before and after low cloud formation (circles) and fog formation (squares) (e-h). Events where the distributions *were not* significantly different at the 95% confidence level from a Wilcoxon rank sum significance test have a black marker edge color.

[Figure]

Same as above, but for the Ångström exponent.

In fog the CPC might undercount total particles, even when conserved, if droplets don't make it through the inlet into counter (quite probable).
Again, it would be useful here to distinguish between low (but elevated) cloud and fog.

The revised figure and analysis surrounding it now includes the separation between low cloud (base > 400 m) and fog formation episodes in panels e-h. We find that in winter, there is little connection between concentration changes around formation and whether a low cloud for fog layer forms. During summer, it is more apparent that the fog formation episodes are associated with a larger decrease in particles from pre- to post-formation. The decrease likely reflects the fact that some aerosol particles are activated to form cloud droplets in the fog (in connection with the change in relative humidity associated with temperature decreases, as shown in the meteorology analyses).

Line 290: "LWN is primarily proportional to cloud liquid..." – only for liquid water paths below the black body limit of ~50 g m$^{-2}$, above that there is little impact on LW radiation.

We completely agree with the point raised by the reviewer. To address this, we have changed the text to the following:

...LWN is primarily proportional to cloud infrared emissivity (which asymptotes at liquid water paths between 30-50 g m$^{-2}$ (e.g., Shupe and Intrieri, 2004)) and the effective temperature difference between the cloud and surface, ...

Line 318: "These seasonal and sky condition differences in particle concentrations suggest different mechanisms are responsible for aerosol numbers near the surface" – this is interesting. Is this simply a result of having an exposed local surface during summer, which may be a strong source or aerosol, and a snow covered or frozen surface for the rest of the year?

It is intriguing to see different "modes" in aerosol number concentrations emerge for the seasons. The changing surface landscape probably plays a role in the absolute numbers, with the more exposed (and potentially drier) surface during summer contributing to near surface particle concentrations. However, we showed in Fig. 9 that the predominant wind directions during summer were often east-northeast. This would suggest that advection from the land to the south is not primary source but instead the open ocean contribution may be a contributing factor. Reviewer #4 has addressed the potential of new particle formation, a process found to be more frequent and contribute to the larger near-surface concentrations during summer (Freud et al., 2017). A dataset of SMPS size-resolved number concentrations is available for September 2007 through mid-June 2008 at Barrow. We explored the clear sky periods during the few months available in this time frame and were able to identify an example where it loosely appears that a new particle formation event was captured in the hours toward the cessation of a clear sky period, leading to cloud formation (see contour plot below, as well as hourly size-resolved number concentrations for the 5 hours after dissipation compare to 5 hours before formation for this particular June 2008 clear sky event. From these figures and supporting literature (e.g. Freud et al., 2017), we cannot dismiss new particle formation as contributing to the enhanced number concentrations during summer compared to winter.

[Figure]

[Figure]

Line 391: "least squares linear regression of the tendencies between the layers reveal a moderate agreement to the monthly cases" – 'with the monthly cases' or 'for' the monthly cases depending on your intended meaning.

We understand the confusion raised by the reviewer. This statement has been removed from the revised manuscript.

Line 418: "The statistical analyses presented fail to identify a definitive signal in aerosol vertical profiles indicating changes in aerosol partitioning are the primary cause for cloud dissipation." – poor phrasing, this is easily misread as meaning "changes in aerosol partitioning are the primary cause for cloud dissipation" rather than "fail to identify a definitive signal in aerosol vertical profiles that would support changes in aerosol partitioning being the primary cause for cloud dissipation"

We agree with the confusion of the statement; this line has been removed in the revised manuscript.

Line 480: "Here, a similar transformation process has been identified on the northern edge of NSA" – I'm not sure one has been identified, only inferred as a potential mechanism.

The discussion around the air mass transformation has been removed from the revised manuscript, primarily because we are unable to track an air mass in a Lagrangian framework to completely explore its transformation process.

Line 488: 'morphology' is not a verb!

This has been removed from the revised manuscript.

Line 505: 'increased pooling of aerosol particles near the surface' – I'm not sure that an 'increase' in pooling is demonstrated. And none is needed, concentrations rarely fall low enough for aerosol to limit cloud formation, so no pooling of aerosol is required to 'provide the ingredients' for cloud formation.

We appreciate the reviewer's concern. We agree that the ingredients for cloud formation are already present and no additional "pooling" is needed. We do feel that the change in particle

number concentrations between before formation and after formation is an important indicator, especially during fog formation, that a fraction of the particles near the surface that were present have likely now been activated into cloud droplets.

---

## Author Comment (AC3) · 24 Nov 2020

In this study the authors attempt to explore reasons for dissipation and formation of low clouds in the Arctic, using a multitude of data from the ARM site in Utqiagvik (Barrow). They first isolate clear-sky periods using a ceilometer and refine these with additional data. They then proceed to analyze data from lidar aerosol backscatter and from in- situ surface measurements of aerosols, radiation and basic meteorology as well as indicators of atmospheric tendencies from soundings. They do this using composites of data for four years.

Their effort is ungrateful in the sense that it turns out to be very difficult to tease out any solid relationships. This is, while of course frustrating, in itself not a reason to reject a paper; a negative result is also a result, and it all rests with how this is handled. However, the paper could be better organized and more clearly written. I recommend that the paper is accepted after major revision focusing more on the structure and language of the paper, more than on the results themselves.

We thank the reviewer for their detailed review of our manuscript. We have considered each suggestion, comment and criticism and have provided detailed responses to each below (in red).

Major comments: This is an original way to analyze data, and the approach is interest- ing. I commend the use of more than cases studies; while this is likely a reason for the lack of clear results, it represents a way to obtain more general results. Anyone can dig out a single case and speculate about reasons for a given outcome, but this is close to useless in a more general sense unless it can be shown that results are more general.

We appreciate the reviewer's commendation of the methodology of our paper. Our intent was to avoid the "case study pitfall", allowing us to more generally characterize the first order processes critical to cloud lifecycle changes.

While this is a strong case for this paper it is also a bit of a weakness in the present manuscript. The background to the problem and the motivation for the method is pre- sented in a very hand-waiving fashion; the current introduction reads more like a list of previous studies and suggestions than an organized argument. Many examples of suggested aerosol influence is listed, but isn't it quite clear why. While aerosols are certainly important, different clouds form mainly because of dynamics than by aerosol constraints. Different types of clouds form in different situations and differently at dif- ferent locations because of different predominant dynamics; low clouds in the Arctic Ocean, frontal clouds in extratropical cyclones and deep convection in the tropics. All of this is modified but not determined by aerosols.

We agree with the reviewer's concern. We have revised the introduction to more appropriately frame the research question that, as of now, the role of aerosol versus meteorology in its contribution to cloud dissipation or formation on the North Slope of Alaska (NSA) is unknown. We use this as a motivation to provide two hypotheses to this research question: 1) clouds are responsive to aerosol presence, or lack thereof, as has been reported over the central Arctic sea ice. 2) General meteorological processes, such as near surface thermodynamic modification or active transient synoptic forcing, are crucial in determining the lifecycle of low level clouds on the NSA.

Hence, I wish that the authors more deeply criticize and discuss the problem of rep- resentatively, as a motivation to stay away from case studies, and then present more clearly the hypotheses they are attempting to test including potential

effects of atmo- spheric dynamics. As it stands, I get the impression they throw whatever data they can lay their hands on, on this problem in the hope that something might show up. I also miss the motivation to why four years of data is used; why not five – or ten?

Please see our response to the previous comment. Our revised introduction does a much better job at framing the research question and developing hypotheses related to aerosol versus general meteorology in determining the fate of low level Arctic clouds. We use the five years (2014-2018) of data because these are the only years where all the instrumentation analyzed were operable simultaneously. This could have been extended by an additional year to 2019, but the results were produced and the original manuscript was being written in mid-to-late 2019.

The paper – even its title– makes a big deal of the clear periods, but if one is inter- ested in cloud dissipation or formation, presumably the happenings before and after the shoulder times are the interesting things; not the clear period per see. Isn't the clear period in between in itself sort beside the point? Also, when clouds are dissi- pated, presumably new clouds will form at some later time, hours or days later; the formation of the new clouds at the end of the clear period may have absolutely nothing to do with the dissipation of the other clouds hours or days earlier. Calling these "book- ends" is misleading in that the reader is lead to think of this as a coupled sequence of events; they may in fact be entirely different. Hence the focus should have been on either cloud dissipation or cloud formation – or both but separately – and then focusing on before and after cloud dissipation/formation.

We are confused with the reviewer's criticism, but we feel this may be mostly related to the title and scope of the introduction. All the results shown in paper, except for potentially Fig. 3, are devoted to understanding the processes connected with the onset of cloud dissipation and the onset of cloud formation. All of the analysis is focused on the vertical structure and variability of aerosol backscatter, near surface thermodynamics and winds, and thermal advection (geopotential thickness tendencies) in time periods prior to cloud dissipation/formation compared to what those time periods just after dissipation/formation.

To address the reviewer's concern, we have changed the title to better reflect that the actual clear periods themselves are not the focus. Further, we have removed any unnecessary discussion of the clear periods in the introduction, besides the important statements related to a need to understand what controls the processes leading to cloud formation and dissipation. NEED TO CHECK THIS!

This constitutes a problem with the lidar, since it is difficult or even impossible to ob- tain aerosol backscatter in the presence of low clouds, attenuating the lidar signal. This is just a fact of life and is discussed on lines 226-227, as in the passing; this information should be given and discussed up front. The results in Figure 3 should therefore be discussed in the context if being clear skies; not in the context of not being cloudy, since that contrast just isn't there. Of course it may still have some value to look at aerosol backscatter directly after dissipation and directly before formation in a statistical sense, as in Figure 4, but this caveat should be discussed up front; that the one set of plots represent after dissipation has happened while the other set is before cloud formation. Without knowing what the structure was before dissipation and after formation of clouds, the information value is limited. And BTW, is this really cloud dissi- pation/formation; isn't it just a hole in the cloud layer advected past the viewer? Maybe this is why its so hard to get statistically robust results?

As stated by the reviewer, the limitations of the HSRL are the reason we had to rely on studying the vertical distribution of aerosol in the time window just after dissipation and just before cloud formation. We supplemented this with Fig. 3 to show the monthly climatological vertical distribution and its variability during entirety of the clear sky periods. To address the reviewer's concern, we have described the limitations of HSRL backscatter profiles together with the description of the instrument in Section 2.

At the end of the discussion section a hypothesis is formulated, almost like in passing; I'm sorry, but I don't get it. It builds on the Tjernström et al (2019) air-mass transfor- mation hypothesis. But a central tenet in that hypothesis is the fact that over melting sea ice, the surface temperature is locked constant at the freezing point; here there is no analogy. So is cloud dissipation leading to surface cooling, then aerosol pooling, followed by fog formation, fog deepening and lifting to clouds?

That would in essence mean that cloud dissipation leads to cloud formation? If this chain of events is really happening, it should be a testable hypothesis; temperature should drop while aerosol concentrations rise with time, followed by fog formation and cloud base rising from zero to some height; in gact, the very same set of data used here could be used to test this hypothesis. Instead the hypothesis is not even clearly repeated in the conclusions, but brushed over with many words in paragraph two and beginning of paragraph three. If you want to pose a hypothesis, do it; else don't!

We have considered the reviewer's critique and we agree with the reviewer. We have removed this hypothesis description in the revised manuscript, mainly since we do not have the modelling capacity to test this hypothesis. This was similar to a critique from another reviewer.

Finally, the language is sometimes what I would – in lack of a better description – call "flowery". It is important to have a capturing narrative, but unnecessarily complicated sentence structures sometimes lead to confusion and misunderstanding. So maybe sometimes be a bit less imaginative.

We appreciate the suggestions and we have gone through, with multiple "sets of eyes" to remove colloquial language throughout the manuscript.

Minor comments

Line 28: Drop "even".

Removed as suggested.

Line 29: Please rephrase; the temperature of low clouds do not reach "as cold as -34 °C" in "all seasons".

As suggested, we have rephrased the sentence to the following:

Liquid-bearing clouds have been observed at temperatures as cold as -34 °C (Intrieri et al., 2002), but liquid is most common during the warmer, summer months (Shupe et al., 2011).

Line 14: Unnecessarily complicated. Suggest "While clear sky is less frequent than clouds" or even "While clear skies are rare".

The manuscript has been revised to read: "While clear sky periods are relatively rare, ..."

Line 38: Lack of what? "longwave warming" or "Arctic clouds"?

The "lack of" statement has been removed from the sentence to avoid confusion.

Lines 39-40: Only true when the sun is absent or the albedo is high; over bare land and in summer, clear skies usually leads to a surface warming. Even in the Arctic.

This statement is based off of results from Pinto et al. (1997). We have revised the statement to read the following, based on the reviewer's suggestion:

Under cloud free conditions with low solar elevations, effective infrared cooling from the surface results in near-surface temperatures to drop (Pinto et al., 1997).

Lines 41-44: A prime example of when there are too many ideas in the same sentence. Exactly what is it that "is currently understood". I know all this so I understand what you mean, but please rephrase anyway.

Following the reviewer's suggestion, we have changed revised the statement as follows:

The Arctic boundary layer tends to remain relatively shallow following the lack of buoyant mixing because stratocumulus cloud-top generated turbulence is absent during clear skies.

Line 43: "stratocumulus and also"

We assume the reviewer is referring to line 47 and not line 43. However, we argue that the original sentence structure is grammatically correct, whereas updating to "stratocumulus and also" as suggested does not make grammatical sense.

Lines 50-51: I would move up "in the Arctic" in that sentence, or it sounds like the transition everywhere is controlled by Arctic clouds.

We have moved "in the Arctic" from the end of the sentence to the beginning, as suggested by the reviewer.

Line 59-60: So opaque liquid clouds would form out of what? Optically thin ice clouds?

We agree with the reviewer that this statement is relatively vague and difficult to follow. We have revised this statement as follows:

Based on observations from the North Slope of Alaska (NSA) and complementary simulations, Silber et al. (2020) found that clouds forming under low aerosol concentration regimes are incapable of producing the cloud-top turbulence necessary to maintain cloud persistence.

Line 71: In what regard is that?

This statement has been removed in the revised manuscript.

Line 71-72: This is a sentence where the narrative is that clouds dissipate and form at the beginning and end of the clear period, as if the dissipation and the formation where reverse analogs.

We agree with the reviewer's concern, and therefore this sentence has been removed from the revised manuscript.

Lines 74-77: Here is a completely different take; now the formation clear period is at focus, not the dissipation of formation of the clouds.

We respectfully disagree with the reviewer; the text, which has been kept intact from the original submission, has always detailed that the purpose of this paper was to explore common processes or differing processes found around events of formation or dissipation of Arctic lower tropospheric clouds. Below is the actual statement from the original submission, which we have kept in the revised paper:

More specifically, we assess whether the aerosol and the general meteorological variability provide clues to the processes that are important for lower troposphere, below 2 km, cloud

dissipation and cloud formation events. By comparing and contrasting the variability of such properties shortly after cloud dissipation (start of clear period) and shortly prior to cloud formation (end of clear period), we aim to learn how changes in aerosol number, aerosol vertical partitioning, and atmospheric thermodynamics contribute to formation and cessation of clear sky periods in the Arctic.

Lines 106-107; what has "a diameter of 10 to 3000 nm"; the volume of the air or the partciles? I know the answer of course, but the sentence is rather unclear.

The statement has been revised to the following:

At the surface, a TSI 3010 condensation particle counter (CPC) measures the number of particles ranging in diameter from 10-3000 nm present within a volume of air.

Line 107: Do all cloud-relevant aerosols absorb alcohol, or do we miss some?

Aerosol composition has a small effect on the detection limit and therefore the counting efficiency of those very small particles (D < 20 nm). In the size range of cloud-relevant aerosols (D > 40 nm), there is no dependence of the counting efficiency on the aerosol composition (see Pg. 9 in the following instrument handbook: https://www.arm.gov/publications/tech_reports/handbooks/doe-sc-arm-tr-227.pdf).

Line 129: Grater than identically zero?

The statement has been revised to the following:

"…point where a cloudy detection status (greater than zero) re-emerged…" to "…time when the ceilometer once again detected cloud overhead and the cloud persisted for at least 2 consecutive hours."

Line 136: How is the agreement on clouds between the ceilometer and the HSRL?

We found the agreement to be surprisingly good. The HSRL is a more sensitive instrument, and therefore there were instances where the HSRL backscatter exceeded a threshold designated as clear sky. Additionally, the CL31 ceilometer has a maximum range of 7600 m, and as such the HSRL would identify cases when higher clouds were present; these instances were excluded from the analysis.

Because of the overlap of the HSRL, the first effective range level where valid data was returned was around 100 m AGL. The ceilometer has a smaller overlap and therefore instances with fog or very low cloud bases were reported by the ceilometer but not by the HSRL. Combining the HSRL with ceilometer and KAZR cloud radar provided a sufficient means to screen the atmosphere for cloud hydrometeor presence.

Line 146: I assume the base is at 100 m and the top is at 400 m; neither is between 100 and 400 m.

The reviewer is correct, and we apologize for the confusion with the original wording. This statement has been revised.

Lines 172-174: Another long sentence with more than one idea confusing the other. Is there any other way a clear period can end than by the emergence of a cloud? And is the ceilometer ever operating in anything but vertical mode?

Following the suggestion of the reviewer, we have revised this entire paragraph, including updating the figure to show the number of cloud formation events that were identified as low clouds and those that were identified as fog.

In regards to the statement about ceilometer operating in vertical mode:  The original text did not state 'vertical mode' but 'vertical visibility mode'. This is a change in the ceilometer processing retrieval software. When the laser beam is attenuated at a range gate very close to the instrument, the retrieval switches from attempting to estimate the cloud base height and instead provides a measure of the vertical extinction of the laser – providing a measure of the vertical visibility. This is what was meant by 'vertical visibility mode'. Regardless, this statement has been removed from the revised manuscript.

Line 188: Not all months have a clear elevated "level of maximum variability". Figure 4: Why one hour?

The reviewer raises a fair point. The wording has been more carefully discussed to reflect that not all months contain elevated variability in aerosol backscatter.

We choose time windows ranging between 1 and 2 hours around cloud dissipation/formation times in order to closely examine any changes in aerosol (and later in meteorology) that may have had an influence on the cloud lifecycle. The reason the time windows changed in duration was connected to the temporal resolution of the datasets/instrument(s). The HSRL reported data more frequently (30 s) than the other instruments examined (1 min). Therefore we choose longer time windows for the lower temporal resolution data streams.

Lines 226-227: This is really important information to have before looking at Figure 3 & 4.

We agree with the reviewer's statement. As such, we have included statements in Section 2 under the description of the HSRL, as well as in the opening paragraph of Secion 4.2.1. These statements specify that HSRL backscatter is only analyzed when the period has been determined to be completely cloud free using a combination of measurements from the HSRL, ceilometer and KAZR.

Line 239: What type of aerosol particle would not come from "below"; what aerosols do not have an origin at the surface except for those emitted by aircraft?

This statement has been removed from the revised manuscript.

Line 278: "agrees" with what?

We are unsure statement the reviewer is referring to, as original Line 278 did not include the "agrees" terminology.

Figure 6: Why now 2 hours; earlier it was one?

As described above, we have accommodated the analysis periods around dissipation or formation events based on the temporal frequency of the measurements. The extension to 2 hours was to consider the lower frequency observations, but also to better capture whether or not variability in

the distributions of CPC concentrations was connected to broader changes in the air mass properties.

Line 279: You are not exploring "phenomena"; you are exploring variables and trying to infer "phenomena".

Phenomena has been changed to processes.

Line 325: "strongly transparent"? Better say "almost opaque".

Almost opaque is the complete opposite of how the Arctic atmosphere interacts with infrared radiation during clear sky conditions; the atmosphere is transparent, hence its reference as the window region. However, based on this confusion and the suggestion from another reviewer, we have changed "strongly" to "largely" to better reflect our meaning.

Lines 342-344: Not sure I get this; if the dew-point deficit has a positive trend (is increasing) and the temperature has a negative trend (is decreasing), does that neces- sarily mean RH is increasing? Could the dew point not decrease so much more than temperature that RH stays constant or even decrease?

Following the reviewer's criticism, as well as comments from reviewer 2, we have computed the tendencies in relative humidity with respect to ice (for November-May) and with respect to water (for June-October) and compared these to tendencies in the temperature. Calculations of relative humidity with respect to ice or liquid were based on the monthly mean near-surface temperatures observed at the NSA.

Line 383: "in flux"? Maybe chose a different wording?

This description has been removed in the revised manuscript.

Line 423: About the source of aerosols again; isn't this trivial? Moreover, I think aerosols are defined as "airborne . . . particles" so there's one "airborne" to many here. Line 424: "general stable stratification" is probably incorrect, or

The reviewer is correct, and these statements have been removed from the revised discussion section. Further, we have removed the word "general" and left stable stratification.

Line 364-365: This is a bold sentence, supported by only one reference. I'm not nec- essarily disagreeing, but still.

We have toned down the statement here, as it was mainly meant to be a transitional sentence motivating the need to analyze the synoptic situation.

Line 383: "in flux"; is this a good choice of words?

This description has been removed in the revised manuscript.

Line 423: Here are the aerosol sources again; I'm no expert but unless you emit them from an aircraft, don't they have to come from the surface?

The reviewer is correct, and these statements have been removed from the revised discussion section. Further, we have removed the word "general" and left stable stratification.

Line 424: The statement on "general stable conditions" is probably inaccurate or at the very least debatable. Studies have shown that the most common near surface stratification over the whole year is near-neutral, but that stably stratified

conditions prevail in clear conditions especially in the winter when they are also deep and strong. Additionally, is there no ground based convection over Alaska or at Barrow; I get over the ocean but this is on land?

While near-neutral stratification close to the surface may be most common, this is primarily due the high fraction of low level stratiform cloudiness although their mixed layer may not always extend down to the surface and therefore indicate a decoupled stratification. When the clouds clear, however, the lack of downwelling longwave critically impacts the surface energy budget, as we have shown with the stability metric in Fig. 7. There may be instances of near-neutral stratification, depending on the time of day during the clear sky period, but predominantly, the lowest ~500 m are stably stratified.

Lines 485-489: Here's that hypothesis; I would have much liked to have the hypothesis at the front and the paper about testing it, or at the end as a bridge to the next study. Here it isn't even a conclusion; reading a bit hasty one could have missed it.

Being that we do not have the modelling simulation to support the formulation of our original hypothesis, we have decided to remove this discussion in the revised manuscript.

Line 511: Maybe avoid the word "transparent" in this context, as it is so intimately linked to other things in this manuscript.

We understand the reviewer's concern, and we have changed the wording to "apparent".

---

## Author Comment (AC4) · 24 Nov 2020

Review of "Processes contributing to Arctic cloud dissipation and formation events that bookend clear sky periods" by J. Sedlar et al.

This manuscript presents an analysis of the atmospheric state (including aerosol concentrations) right before and after the onset of cloudy and clear periods at Utqiagvik, Alaska. The main motive of the work is to understand the processes that drive low-level cloud formation and dissipation in an Arctic environment.

I find the overall aim of the study and the analysis of available observations interesting and commendable. However, it seems like the manuscript was put together a bit too hastily; the overview and connection to published literature could be expanded (in particular in terms of Arctic aerosols), the presentation of the instrumentation and methods needs more information and the discussion of the results lacks some clarity and depth. On the data analysis side, I also find some issues with the way that the aerosol data from the CPC are treated. As stated in the manuscript, the data from the CPC will give you the total aerosol number concentration, including aerosols down to 6 nm diameter. This is a problem, at least during summer, when the total aerosol number concentration is dominated by smaller aerosols (nucleation and Aitken mode), which have very little influence on cloud droplet formation. Relating the aerosol concentrations from the CPC with cloud formation is therefore dubious.

We thank the reviewer for their insightful review, with particular attention to the aerosol focus of the manuscript. We have considered all the criticisms, comments and suggestions provided by the reviewer, and we have replied with detailed responses to each comment below, in red.

**General comments:**

- • I would suggest that the authors are a bit more careful when they use the term "the Arctic" or when they refer to certain characteristics of "the Arctic". The Arctic is not a homogeneous region where clouds, meteorology and surface properties are the same. Many of the features that the authors mention, in particular in the introduction, may not be true for the lower-latitude parts of the Arctic and/or land areas. For example, are clouds ubiquitous over the whole Arctic during the whole year? Does the longwave radiation dominate the radiative energy budget everywhere and during the whole year? Under cloud-free conditions, does effective infrared cooling from the surface cause extremely cold temperatures everywhere? I am thinking for example of Siberia where you in the summertime can have very different conditions compared to over the Arctic Ocean.

The author raises a valid point. Utqiagvik on the North Slope of Alaska (NSA) is only one station with the Arctic region. However, the NSA is still a part of the Arctic, and on top of that, it is home to long data records which make a statistical study like this one possible. To address the reviewer's concern about representativeness of Utqiagvik as the Arctic, we have changed the title of the manuscript to better reflect the study region. We have kept the motivation in the introduction section regarding Arctic clouds and subsequent clear sky periods as being very important to the surface energy budget of the Arctic; we feel that although the conclusions drawn may not represent the entire high-latitude region, we do feel that they are at a minimum representative of northern Alaska coast.

- • Related to the previous comment, how representative is Barrow as a station for "the Arctic" and the type of cloud formation/dissipation events that you study? I think that the idea that aerosols control cloud formation/dissipation has mainly (only?) been presented for high (>80°N) Arctic clouds, i.e. in pristine environments where (accumulation mode) aerosol number

concentrations are extremely low. Utqiagvik (or Barrow) has rather high (accumulation mode) aerosol concentrations for an Arctic station (cf. e.g. Freud et al., 2017 or Schmale et al., 2018). It may still be an interesting place to study low-level cloud formation and dissipation, but perhaps not so much from the perspective of an aerosol-limited regime?

After working on this study, we agree with the reviewer that Utqiagvik is likely not a representative location for study the aerosol-limited regime. The presence of land and nearby ocean, often ice free for a considerable portion of the year, revealed that a lack of lower tropospheric aerosol is not a common occurrence on the NSA. However, that does not discredit the attempt made here to quantify from the NSA whether or not signatures in near-surface and lower tropospheric aerosol may indicate a relationship or connection to the mechanisms contributing to dissipation and/or formation of lower troposphere clouds. We have made it clear in the revised discussion and conclusions that changes in aerosol presence are not the cause for cloud dissipation. When it comes to cloud formation, especially fog formation which we have separated and paid more attention to in the revised manuscript, the role of large concentrations of both small and also fewer but larger aerosol in summer are linked with air mass transformation. The end result supports the formation of fog following clear sky periods. To our knowledge, this result and mechanisms promoting summer fog formation, has not been identified on the NSA previously.

- • The authors use CPC measurements to relate aerosol concentrations to cloud formation/dissipation events. Firstly, I think that the methodology related to the CPC measurements needs to be better explained. What air is pumped into the instrument? Is it "whole air", "cloudy air" or "clear air"? How are ice crystals and cloud (fog) droplets handled by the instrument? Is the air dried? Does the instrument have any detection limit in terms of number? Secondly, the CPC measures particles down to 6 nm (as stated by the authors). The Arctic is typically dominated by small aerosols in summer (cf. e.g. Freud et al., 2017) but these small aerosols are not efficient cloud condensation nuclei. Figure 3 in Freud et al. shows that in summer, the accumulation mode particle concentration typically goes down drastically while the total concentration of aerosols goes up as new particle formation and growth controls the aerosol population. Why did the authors not use Scanning Mobility Particle Sizer (SMPS) aerosol size distribution or CCN measurements from Utqiagvik? I think these should be publically available (cf. e.g. Schmale et al., 2017).

-

Air is sampled continuously in all conditions, so the inlet would be best described as a whole air inlet. There may be some losses of the big hydrometeors (ice/droplets) due to inlet design, but those are unlikely to have significant effect on the number concentration. Details regarding the inlet and sampling strategy can be found in Quinn et al. (2002; doi:10.1029/2001JD001248, 2002). In terms of detection limit, communicating directly with the responsible instrument PI, even the highest concentrations observed at Utqiagvik have never reached it; the actual detection limit is not certain but the PI was sure it is above 20,000/cc. This is far greater than any of the concentrations that were measured during our study period.

Our study is designed to be a statistical study from 5 consecutive years cloud dissipation/formation events when all of the measurements/instruments described in the methods were operational. Going back to 2008 to examine CCN and/or SMPS measurements for a handful of cases is not the intention of this paper. Furthermore,  unfortunately, the instruments the reviewer suggests to analyze were not in operation during the 2014-2018 study period, following the link to Schmale et al. (2017). We even reached out to the instrument contact PIs from the group (TROPOS) that is

supposedly operating an SMPS currently on Barrow, to see if we could kindly have access to their data to include in this study; we received no response back from them.

However, to satisfy the reviewer's concern with potential new particle formation events, we have looked at the brief amount of SMPS data available during September 2007 to mid June 2008; see responses below in specific comments section. The only case we could find during summer (where NPF events are common) occurred in mid June 2008 and the results do suggest signatures of NPF. We feel this further supports our claim that processes are contributing to enhanced particles numbers during clear sky periods on the NSA – which support cloud formation. We have further explored the scattering and Ångström exponent behavior of these distributions (see figure in specific comments below as well as Figure 7 in the revised manuscript) and these results suggest that while CPC numbers are very large, there is still a component to the aerosol distribution that contains larger particles (more efficient as CCN) based on the scattering and Ångström exponent behavior.

Unfortunately, CCN measurements stopped in 2012 at Utqiagvik.

• I find the discussion about the vertical structure of geopotential height and "synoptic activity" and their relation to cloud formation and dissipation events confusing. In Section 3 (lines 387-398), the authors say that "From May through summer, differential advection amongst the atmospheric layers becomes a more frequent occurrence." From this, they conclude that cloud dissipation events are often associated with baroclinic activity in summer. I would also assume then that the *synoptic activity* is more frequent in summer during cloud dissipation events. The same is also true for cloud formation events (lines 400-409); these are more frequently associated with synoptic activity in summer compared to winter. But in the discussion section, it is stated that (in association with cloud formation events) "Variable dynamics resulting in differential atmospheric advection is most prominently observed during the winter and early spring. Furthermore, in the conclusions, the authors state (in relation to cloud dissipation events) "While we report that all months are subjected to synoptic disturbances, the magnitude of the forcing is weaker during late spring and through early autumn than during winter and early spring."

We appreciate the concerns raised by the reviewer, and we have carefully considered these comments when revising the manuscript. First, we have moved on from using the radiosounding-derived geopotential thickness tendencies, instead using ERA5 reanalysis profiles of geopotential height to derive thickness tendencies. Radiosoundings were nominally every 12 hours, and as Reviewer 2 commented, it is very possible that the two consecutive sounding profiles occurring prior to a cloud dissipation for formation event may not be representative of the synoptic setting that actually impacted the thermodynamics. With ERA5, we are able to use 1-hourly profiles, from which we derive thickness tendencies across a 4-hr period prior to a cloud dissipation or formation event. We assert that the 4-hr timescale is more relevant to the synoptic forcing influencing the event, as well as it provides a sufficient number of data points in which to produce a tendency. Furthermore, we are able to calculate the 4-hr consecutive tendencies for each month of a season during the 5-yr period, which provided a mean and standard deviation of the seasonal layer thickness tendencies. We used this seasonally climatology to understand when specific dissipation/formation events exceeded the 5-year climatological standard deviation. These revised results, and the discussion following them, are now discussed more thoroughly in the revised manuscript. In particular, we have removed confusing and contradictory statements, like those described above by the reviewer.

**Specific comments:**

*Abstract:*

- • Line 2: I would suggest reformulating the sentence including "...lack of downwelling...". It sounds like there is no downwelling radiation at all when the cloud is absent.

We have removed the statement about the lack of downwelling longwave in the revised abstract.

- • Line 18: I am not sure why you emphasize the link to aerosol concentrations here? Isn't any general change in dynamics/radiative cooling more important?

This statement has been removed from the revised abstract.

   *1. Introduction*

- • Line 27: Are there any other studies than Shupe et al. (2011)? Would be interesting to know.

There are many studies prior to Shupe et al. (2011) that document the vertical distribution of clouds, many of these are connected to individual field campaigns or satellite observations that pre-date the Shupe et al. study. Shupe et al. (2011) use pan-Arctic observations (a number of "supersites" containing a variety of active and passive remote sensors) over a number of years to document the vertical partitioning of Arctic clouds. Since our paper relies on observations from one of the observatories analyzed by Shupe et al., we have decided to retain this as the most appropriate reference.

- • Line 27: I suggest changing "These clouds frequently contain concentrations of both..." to "These clouds frequently contain both ...".

Changed as suggested.

- • Line 54. "Simulations of Arctic clouds consistently show that over-abundant ice nuclei or ice crystal concentration can lead to cloud glaciation". I don't think this statement is completely true – it depends on what the authors mean with "over-abundant" and "Arctic clouds". There are several studies that show that mixed-phase clouds in the high Arctic only glaciate at extremely (i.e. unrealistically) high ice crystal number concentrations, e.g. Stevens et al. (2018), Loewe et al. (2018).

We have updated this line to address the reviewer's concern. The revised manuscript now states:

Simulations of Arctic clouds consistently show that enhanced ice nuclei (IN) or ice crystal concentrations can lead to mixed-phase cloud glaciation (Harrington et al. 1999; Jiang et al. 2000; Avramov and Harrington, 2010; Morrison et al., 2011), as ice precipitation acts a net sink of cloud mass (cf. Solomon et al., 2011; Forbes and Ahlgrimm, 2014).

- • Line 56: Related to the previous comment, I think a CCN-limited regime has only been suggested for high Arctic clouds?

We tend to agree with the reviewer in that we also have only seen the CCN limited regime to be present over the central Arctic sea ice. However, with these statements, we are setting the stage for

how our analysis will explore both the aerosol vertical distribution characteristics as well as meteorological forcing properties in the attempt to understand the processes important for cloud dissipation and formation on the North Slope of Alaska.

- • Line 61: In this paragraph, it could perhaps also be worthwhile considering the studies by Young et al. (2018) and Dimitrelos et al. (2020) where they point out the importance of large-scale divergence/convergence (and associated free tropospheric moisture supply) in governing the lifetime of Arctic low-level clouds.

Based on the reviewer's comments, along with 2 other reviewers, we have revised the introduction to better identify the scope of this paper. In this regard, we have made careful effort to identify observational and modeling studies that have highlighted the importance of cloud lifecycle evolution due to microphysical changes (CCN, IN) as well as synoptic forcing. We thank the reviewer for alerting us to the Young et al. 2018 paper, which we have included in the revised manuscript.

- • Line 75: When reading the introduction, I was wondering why you focus on atmospheric properties "after cloud dissipation". It would have made more sense to look the atmospheric state before cloud dissipation. In the methods section you then explain why this is not possible, but I think it could be good to include a short explanation already in the introduction.

We try to be consistent throughout the paper. In this respect, we do look at the atmospheric properties both before and after cloud dissipation and formation events. However, for the HSRL analysis, because the lidar signal dominated by cloud hydrometeors, we cannot study the changes in vertical distribution of aerosol backscatter prior to dissipation, or shortly after formation. To address the reviewer's comment, we have changed the introduction to include the the following sentence:

"By comparing and contrasting the variability of such properties around cloud dissipation (start of clear period) and around cloud formation (end of clear period) events,…"

*2. Instruments*

- • Line 91: The description of the HRSL is very brief and should be expanded. For example, what is the detection limit of the lidar? Is there a limit in terms of how close to the surface the signal can be trusted?

After a literature search, we could not find a standard value listed as the detection limit for this particular HSRL. However, we note the backscatter cross sections below $1x10^{-7}$ ($m^{-1}$ $sr^{-1}$) were generally not observed (See original Fig. 5), and therefore could be considered the lower detection limit for a pristine Arctic troposphere. This backscatter detection limit was also identified by Shupe (2007) as the threshold for a completely clear (clean, pristine) atmosphere. A study by Thorsen et al. (2017, **https://doi.org/10.1002/2017GL074521**) has explored various sensitivities of other HSRLs in comparison with CALIOP onboard CALIPSO. That study was focused on how the sensitivity in aerosol backscatter from lidar would be critical for aerosol optical depth estimates. However, in our study, we are more interested in the vertical presence of aerosol layers, in particular if there are sharp contrasts in the vertical distribution of enhanced or dimishied aerosol backscatter cross sections. Our analysis did not show this to be the case. Following the reviewer's suggestion, we have included information regarding the first vertical range analyzed.

- • Line 1010: How small concentrations of small cloud droplets can the cloud radar observe?

Generally, this will be dependent upon whether ice crystals are present or not. We have included the following line to the revised manuscript:

While the KAZR is capable of observing concentrations of small droplets, its measurement is sensitive volume squared and therefore the signal may be attenuated in by the presence of ice crystals which are typically larger than droplets (e.g., de Boer et al., 2009).

*3. Methods*

- • General: it would be nice to have a map of the location of the station and also a brief description of the typical conditions (closeness to sea, potential pollution sources etc.)

We argue that the Arctic science community is generally well-versed in the geographic location of Utqiagvik and the North Slope of Alaska. Therefore we have decided against including a map. We did however include the following paragraph to the start of Section 2 describing the general cloud conditions, general air mass footprint and connection with pollution from nearby oil fields and wildfires:

The observatory at Utqiagvik is an ideal location to understanding the contribution of meteorological and aerosol processes to Arctic cloud dissipation and formation. Generally, cloud fractions are high, typically between 60 and 95%, and lower tropospheric clouds were common, especially during sunlit months (Shupe et al., 2011; Sedlar, 2014). Having a relatively large cloud occurrence makes the NSA a viable location to further study the process that lead to the formation or cessation of a clear sky period. Utqiagvik is at a coastal site, located within 2 km of the coast line along the NSA. Seasonal climatologies of the back-trajectory footprint of air masses reaching the observatory were predominantly from the high Arctic Ocean, and to a lesser extent from the continent to the south (Freud et al., 2017). Pollution from the oil fields around Prudhoe Bay did not regularly lead to changes in background aerosol or cloud microphysical properties at Utqiagvik (Maahn et al., 2017). However, wildfires may sporadically influence the background aerosol concentrations and chemical composition across the NSA during active fire seasons (Creamean et al., 2018).

- • Line 130: I'm just curious, why 96%?

There is no real scientific reasoning for the choice of 96%. At some point, it was necessary to "allow" occasional observed cloud signatures within our definition of a clear sky period, otherwise

our study would have been limited to very few cases and would not be sufficient as a statisical study.

- • Lines 138-140: I suggest replacing the word "when" with "if".

Changed as suggested.

- • Line 146: Why show times as UTC and not local times? Would make it easier to interpret the radiative fluxes.

This is a matter of preference, and we have decided to keep the hours in UTC time.

• Line 154: It is not completely evident to me that the mixed layer (elevated aerosol backscatter) is shallower during the clear period. How do you see this? Maybe it would help to draw a line at the start of the clear and cloudy periods?

We interpret the mixed layer depth variability as the height where the HSRL backscatter beings to drop off dramatically with height. Prior to cloud dissipation (around 04:00UTC) this backscatter transition occurs above the 300 m level. This level shows a gradual decrease in height up until around the mid-point of the clear period ($\sim$08:00UTC). This is what we refer to as the shallower mixed layer, which is further shown in the inset of equivalent potential temperature profiles in panel c; there we find the mixed layer depth has decreased rather considerably, indicating a mixed layer depth of < 200 m. This layer depth is below the previous cloud base height ($\sim$300 m). We do not have a radiosounding during the cloud dissipation phase, but if we assume the cloud was coupled with the surface just prior to dissipation, this would suggest a decrease in mixed layer depth of more than 100 m had occurred.

- • Line 155: "Evolution in near-surface meteorology showed modest changes...". I interpret "modest" as "not pronounced", but maybe this is not what the authors mean. I would say that the change in wind direction is fairly pronounced at the time of cloud formation? And also the change in dew point temperature?

We have revised the statements to reflect the changes observed in wind direction and near-surface thermodynamics, as suggested.

- • Line 157: It is quite interesting that the particle concentrations increase so dramatically during the clear period. In summer, new particle formation and/or condensational growth of nucleation mode particles often takes place when there is sunlight and (initially) low background concentrations of aerosols (e.g. Freud et al., 2017). Could this be what is happening? Was this a typical pattern or only a one-time feature? Important here is of course also what air the CPC samples, if it is "whole" air or only cloud-free air.

As the reviewer knows from further reading, this was not a one time example, but a relatively consistent process especially during the summer. We have continued with this analysis, following the reviewer's suggestions and questioning below, in subsequent sections of this manuscript. However, we do not feel that it is appropriate to hypothesize on new particle formation at this point in the manuscript, as we are using Fig. 1 to simply show the typical setting of a dissipation event, clear sky period, and formation event.

*4. Results*

- • Line 165: Just out of curiosity, was there any difference in length of the clear periods between the seasons?

Generally, no. Each month tended to have clear periods that ranged from about 3 hours to as many as 26 hours; the longest clear periods were infrequent and therefore skewed the distributions.

- • Line 170: I assume that the clouds with bases below 400m also could include other clouds than fog and low clouds? For example nimbostratus, cumulus and cumulonimbus.

While this is possible, the typical low cloud type across the Arctic is the low level stratocumulus, often mixed-phase. The identified fog events are unlikely to be anything other than fog (which is by definition a cloud with a base level at the surface and reduced visibility). In the revised manuscript, more consideration has been made to separate the low cloud and the fog cases to identify whether different atmospheric processes or mechanisms could be linked to the different cloud formations.

- • Line 188: What is the "1-sigma envelope"?

The 1-sigma envelope referred to the 1 standard deviation around the mean profile at each height. This statement has been removed in the revised manuscript.

- • Lines 190-194: I have several questions/comments regarding this paragraph.

o When is the boundary layer backscatter (which should be dependent on the aerosol surface area, so mainly the accumulation mode) the highest/lowest? How does this agree with other in-situ measurements of CCN and/or aerosol size distribution measurements (e.g. Freud et al., 2017; Schmale et al., 2018; Schmeisser et al., 2018)

These measurements of HSRL backscatter are valid for a limited number of clear sky periods only, not the entire monthly distribution as is the case for the studies listed by the reviewer. We find that in terms of CCN concentrations, the seasonal cycle shown in Fig 3 (along with Fig. 6 later in the manuscript) are very similar to those measured from Utqiagvik (Lubin et al., 2020). We have included this important connection with our analysis of Fig. 6 later.

o Is it really true that the "transition layer" is the shallowest in summer? October and September looks pretty shallow too?

We have revised the statement to reflect the reviewer's point; the revised sentence is:

For example, the summer and early autumn (g-k) mean backscatter decrease happens over a shallower layer above the surface and is more abrupt than during winter and spring (a-f).

o I don't understand the sentence that begins with "Many processes may contribute to ...". Shouldn't this layer just be a result of the vertical depth of the boundary layer/mixed layer?

This is correct, and we did include one of the BL processes that the reviewer is referring to, namely the lower atmosphere stratification. However, we have included 'boundary layer mixing' in the revised manuscript to satisfy the reviewer's concern.

• Line 213: The limitation of the HSRL should be mentioned in Section 2.

The revised Fig. 5 shows the median and interquartile spread of the seasonal aerosol backscatter for low cloud and fog forming events only. Below the cloud layer, it is apparent the backscatter is considerably larger than above cloud top. While we may be approaching the detection limit of the HSRL above the cloud top, and further above into the free troposphere (see panels g-h), it is clear the aerosol backscatter in the layer where cloud would eventually form is above this backscatter. Therefore, we do not see any evidence that the HSRL would miss small concentrations of aerosol particles. Even if it were that small aerosol concentrations were below the detection limit of the instrument, the fact that aerosol backscatter remained larger after the cloud dissipated compared to just before it formed suggests that a sparsity of aerosol (with which to be activated as CCN) was not the reason for the cloud dissipation; it certainly did not inhibit the formation of cloud.

• Line 214: Can you really draw this conclusion from looking at averages? I would think that in order to make this statement, you would have to look at the individual profiles and make sure that the transition layer is always below cloud or within the cloud that the clear-sky period bookends?

This statement has been removed from the revised manuscript. We agree with the reviewer that the climatological profiles may prohibit potentially small scale features which may be ongoing on a case by case basis.

• Line 221: The selection based on a maximum cloud top height below 2km makes sense and should be done from the beginning.

The specification that this paper focuses on cases with clouds below 2 km has been added to the last paragraph in the introduction.

• Line 236: The cutoff backscatter values should be mentioned in Section 2. But I am also wondering what the authors mean with "clear sky"? I assume there should still be aerosols present, it is just that the instrument cannot detect these low concentrations?

In the revised manuscript, this section has been updated. We have specified that the threshold of $1 \times 10^{-7}$ $(m^{-1} sr^{-1})$ was developed by Shupe (2007) as the distinction of pristine Arctic air. While there may be aerosols present, having such a small contribution to the backscatter suggests their cross sectional area must be small meaning the number concentrations should also be small.

• Line 241: What do the authors mean with the sentence "Being that the aerosol backscatter... was at minimum..."? Where and how do you see this?

This statement has been removed from the revised manuscript.

• Line 241: Related to the comment above, how low backscatter values would you need in order to have accumulation mode aerosol concentrations below ~10cm-3?

This would require the use of a forward model of Mie scattering, which is beyond the scope of this paper.

- Line 248: Please define "RFD".

RFD has been defined as relative frequency distribution, as suggested by the reviewer.

- Lines 257-260. I do not think this argument holds. The backscatter will be dependent on surface area. If the aerosol population is dominated by small particles in summer, then the surface area will not be at its maximum, see also Freud et al. (2017).

We agree with the reviewer, in that the original statement was misleading. We have expanded the analysis in this section by the following: 1) we identified a bug in our plotting of Fig. 6, where instead of the interquartile range being plotted, error bars were mistakenly plotting the median value +/- the 25th and 75th percentiles. 2) we have included the median and interquartile range of the 550 nm scattering coefficient around cloud dissipation and formation times. This figure illustrates that the scattering coefficient during summer tends to be as large, or larger, prior to formation than after formation (panel g). That the scattering coefficient has not consistently decreased suggests a sufficient presence of accumulation model particles that much more readily scatter light compared to a distribution dominated by smaller Aitken or ultrafine particles.

We also looked at the Ångström exponent in the same manner; see figure below. We find that the exponent was similar or even smaller prior to formation compared to after formation (g). As the Ångström exponent is inversely proportional to size, it is clear that it's not only very small particles but also larger particles are contributing to the size distributions and the increase in particles observed by the CPC.

[Figure]

Same as in Figure 6 of manuscript, but for the Ångström exponent.

• Lines 269-271: This results is interesting as the increased number of particles in spring/summer could be due to new particle formation and growth during clear periods, please see previous comment (Chapter 3, line 157).

We agree with the reviewer that this result is interesting, and we have based much of our discussion around the formation of fog during summer around a number of processes ongoing near the surface; with the increase in particles, potentially from new particle formation events, playing a role in the formation process. We have revised the discussion section to highlight this.

As discussed above, SMPS data was not available for the time period of this study. However, we did look back at the September 2007-June 2008 SMPS dataset and did a similar analysis of cloud free periods then. Below is a figure that shows one particular clear sky period during June 2008, showing the time evolution of the SMPS particle size distribution. There is evidence of signatures that are consistent with a new particle formation event (potential banana curve) occurring towards the end of the clear sky period.

[Figure]

• Lines 271-274: Does the CPC measure "whole air" or only "clear air"? If it is "whole air", then why would the concentrations decrase?

The CPC measures what the reviewer calls "whole air". The inlet samples air continuously during all conditions. Following the changes observed in the near surface thermodynamics and the lack of wind direction/speed changes for fog events, we have no reason not to believe that the decrease in particles is not from an uptake through activation into a fog droplet.

• Line 290: I do not think this argument is true. The downwelling LW should also be dependent on the temperature, in particular if the LWP is larger than ~$20 gm^{-2}$ (emissivity close to 1).

We have revised to text to highlight the importance of emission temperature once the cloud imitates a blackbody, as follows:

"…in the data since LWN is primarily proportional to cloud infrared emissivity (which asymptotes at liquid water paths between 30-50 g $m^{-2}$ (e.g., Shupe and Intrieri, 2004)) and the effective temperature difference between the cloud (or clear sky) and surface,…"

• Line 293: How is the analysis affected by any presence of a stable surface layer (boundary layer decoupling)?

The methodology of calculating equivalent potential temperature differences between the surface and 950 hPa will include any increases in potential temperature found within this layer. If there is a stable layer at 20 m AGL, or 200 m AGL, a potential temperature difference between 950hPa and the surface will be reflected in this calculation.

• Line 297: I think it should be mentioned in Section 3 that you use the soundings to calculate LTS.

Such a statement was already included in the original manuscript – see line 119-120. We have kept this in the revised version.

• Line 300: Related to figure 7, why is the cooling generally smaller with more stable stratification (for clear sky)?

It is likely that these instances are associated with significant temperature and/or moisture advection at low levels, contributing to a stronger temperature inversion in the lower troposphere. Even though clouds are absent, increased temperatures, especially in the presence of enhance moisture, will cause a relative increase in the downwelling longwave, which will act to offset the LWN deficit. The very strong LTS values are consistent with strong, low-level temperature inversions.

• Line 318: Which mechanisms are you referring to?

The next section explores the relationship of cloud dissipation and formation events to near-surface thermodynamic, winds and synoptic changes, which we link to the changes in aerosol characteristics.

• Line 342: So this means that in summer you mainly have fog formation due to radiative cooling?

This is one of the primary findings that we are asserting in this paper. The subsequent analyses and discussion section further emphasize this point.

• Line 356: Are these results then inconsistent with the geopotential tendencies where you concluded that synoptic activity was more frequent in summer and spring during cloud dissipation events (lines 395-398)?

We don't believe the results to be inconsistent. The original manuscript used 12 hr radiosounding profiles to calculate the layer thickness tendencies. Depending upon the start/end time of a clear period, this meant that the thickness tendencies could be computed a full 12 to 23 hours prior to the actual dissipation or formation time. Using reanalysis has allowed us to reduce the potential for increased time lag between the thickness tendencies and the time periods of interest. The updated figures and results are consistent, namely that abrupt synoptic frontal forcing, while occurring occasionally, is not the primary feature observed from spring through autumn. Instead during these seasons, low cloud and especially fog formation events are connected with the least amount of variability in near surface wind direction (and wind speed differences) and also have the smallest layer thickness tendencies.

• Line 365: For the analysis of geopotential tendencies, I think it could also be interesting to look at these from the perspective of large-scale subsidence and convergence as in Young et al. (2018) and Dimitrelos et al. (2020). It would also be interesting to look at vertical profiles of moisture to see if the layer right above the cloud is a source or sink of moisture.

While these studies are interesting and show a connection to large scale structure, they are not explicitly focused on synoptic scale forcing. Young et al. shows that simulated cloud lifecycle is associated with divergence, although this can emerge through stagnant air mass modification as well as synoptic forcing. The Dimitrelos et al. study also limits the change in cloud to changes in divergence and its impact on cloud top processes.

While we agree with the reviewer that processes occurring near cloud top are important, we simply do not have the temporal availability of profiling in order to match the statistical climatology of our study. This would be more geared toward a case study analysis. We intend to use a cloud resolving model to further explore a handful of these dissipation cases in a future paper.

• Line 372: I would suggest inserting a "vertical" before "structure".

Revised as suggested.

• Line 380: How much was the number of cases reduced?

Because we now rely on 1-hr reanalysis data, the analysis is completed using all available dissipation and formation cases.

• Line 401: You mean in late spring/summer...?

In the revised manuscript, the description of he figures and the analysis of the results show have been completed redone. This statement no longer exists in the revised manuscript.

*5. Discussion*

• Line 430: I am not convinced that differences in horizontal advection is the main reason for the differences in vertical distribution of aerosols, see e.g. Freud et al. (2017).

In terms of seasonal variability, it is likely that changes in vertical distributions of aerosols results from either advection or cloud processing and deposition. Freud et al. demonstrated the footprint of aerosol typically extends from over the central Arctic. Mauritsen et al. (2011) found that over the central Arctic, aerosol concentration can potentially fall well below 10 cm-3. These "pristine" air masses are generally not stagnant and must be transported across the Arctic. Our results have identified that synoptic forcing at the NSA was largest in winter and spring, weakest during summer. Combined with the larger variability in clear sky aerosol backscatter across the lowest ~1 km during winter and spring, we are left to conclude that advection is indeed an important mechanism in aerosol vertical distribution.

However, in the revised manuscript, we have removed the statement that the reviewer has questioned.

*6. Conclusions*

• Line 499: I thought the forcing from synoptic disturbances was stronger in late spring through summer (lines 395-398)?

We understand the confusion that was raised by these contradictory statements in the original manuscript. Following Reviewer 2's concern with using infrequent (12 hr) radiosoundings to understand thickness tendences, we have revised the analysis to use 1 hr reanalysis profiles from the state of the art ERA5 reanalysis. Now using a 4 hr window prior to dissipation or formation events to compute tendences constrains the analysis to focus on the synoptic evolution directly connected with a cloud lifecycle event. The new results continue to reveal that winter is more synoptically active than summer, while spring, and to a lesser extent autumn, represent transitional seasons in synoptic activity. These results have been more carefully described in the Section 4.3.2.

• Line 511: I guess there is also a possibility that the cloud formation and dissipation events does not happen "in-situ" but rather that transport of clouds (and clear air) contribute to the observations made at Utqiagvik?

Absolutely. We only have observations at one point so this paper only attempts to analyze the ongoing processes surrounding the clear sky periods.

**References**

Freud et al. Atmos. Chem. Phys., 17, 8101–8128, 2017 https://doi.org/10.5194/acp-17-8101-2017.

Young et al. Atmos. Chem. Phys., 18, 1475–1494, 2018 https://doi.org/10.5194/acp-18-1475-2018.

Dimitrelos et al., 2020. Journal of Geophysical Research: Atmospheres, 125, e2019JD031738. https://doi.org/10.1029/2019JD031738.

Loewe et al. Atmos. Chem. Phys., 17, 6693–6704, 2017 https://doi.org/10.5194/acp-17-6693-2017

Stevens et al. Atmos. Chem. Phys., 18, 11041–11071, 2018 https://doi.org/10.5194/acp-18-11041-2018

Schmale et al., Sci Data 4, 170003 (2017). https://doi.org/10.1038/sdata.2017.3.

Schmale et al. Atmos. Chem. Phys., 18, 2853–2881, 2018. https://doi.org/10.5194/acp-18- 2853-2018

Schmeisser et al. Atmos. Chem. Phys., 18, 11599–11622, 2018. https://doi.org/10.5194/acp- 18-11599-2018

---

## Referee Report (RR1)

Review of "**Processes contributing to Arctic cloud dissipation and formation events that bookend clear sky periods**" by J. Sedlar, A. Igel, and H. Telg.

Submitted manuscript version 2.

Ian M. Brooks

**Overview**

This revised manuscript is a significant improvement on the original. The major issues raised in my original review have been addressed. The result remain somewhat inconclusive, but nevertheless the extensive documentation of cloud, aerosol, thermodynamic, and large scale dynamic conditions are a useful contribution to the field.

I recommend that the manuscript is suitable for publication after minor revision. Detailed comments to be addressed are noted below.

**Detailed comments**

Line 11 – "A suite of remote sensing and in situ instrumentation from the high-latitude observatory are analysed;…" -> "Measurements from a suite of…are analysed;…" – the measurements are analysed not the instruments.

Line 14-15 – "the clear period bookends" – 'bookends' here is a rather casual, and not entirely clear, term. Maybe rephrase to something like '…aerosol….is relatively invariant during the periods bookending clear sky conditions'

Line 20 – "aerosol particles concentrations changed by a factor" – a factor of what? Need a value (and sign) of the change here

Line 40 – "effective infrared cooling from the surface results in near-surface temperatures to drop" – grammar, 'to drop' doesn't fit with the rest of this statement -> "effective infrared cooling from the surface results in near-surface temperatures decreasing"

Line 108 – "tropospheric clouds were common" – tense doesn't match first part of sentence -> "tropospheric clouds are common"

Line 137 - "although some concentrations may" – a very vague statement, need more detail. 'some' concentrations…high, low, variable but under some particular conditions?

Line 141 – "its measurement is sensitive volume squared" – grammar – "its measurement is sensitive to particle volume squared"

**Methods**

Line 172 – "…condition was not met, the clear period was discarded…" – suggest changing wording to "…condition was not met, the period was discarded…", if the period is discarded because of intermittent cloud then it's not really a 'clear' period for the purposes of this study.

Line 228-238, discussion of figure 3 – the processes mentioned as possible causes of the drop in aerosol backscatter between BL and overlying air are all reasonable. An additional factor may be the typical decrease in humidity across BL top. For hygroscopic aerosol, particle size can change significantly with relative humidity (ballpark values are a doubling between 'dry' and 80% HR, and another doubling between 80% and ~100% RH for particles such as sea salt), this might lead to a drop in backscatter across BL top even for an aerosol population that was uniform in concentration and dry radius across the inversion. This is, of course, highly dependent on aerosol chemistry, and change in RH across BL top, and not quantifiable here, but worth keeping in mind.

Figure 4 – some of the panels show colours (at high backscatter) outside the range indicated on the colour bar.

Line 347 – "…these clouds often modulate the stratification due to cloud top radiative cooling and induced turbulence…"
i) This phrasing is ambiguous – not clear if the meaning is that the stratification itself, or the modulation of the stratification, is due to cloud-top radiative cooling,
ii) The stratification referred to (or implied by the preceding statement) is the 'static stability near the surface' – I'm not sure that cloud-top radiative cooling and associated turbulent mixing impacts strongly (or in some cases at all) on the near surface stratification. That is much more strongly influenced by the simple presence of cloud and whether the surface itself is cooling radiatively (clear skies) or not (cloudy skies). Cloud driven turbulence will certainly impact BL thermodynamic structure as a whole, and might extend to the near-surface layer, but is only one of several factors affecting surface stability.

Line 354 – '950 hPa level is generally around 500 m AGL in the Arctic, which *frequently* encompasses all, or a fraction of, the Arctic atmospheric boundary layer and the sub-cloud mixed layer' – rather loose and partly redundant phrasing. The lowest 500m must always encompass at least part of the BL. It will often encompass at least part of the sub-cloud mixed layer – though since your focus here is on cases where cloud base is <= 400m, it must also always encompass the sub-cloud layer for all cases considered here.

Line 485 – 'Little changes in the vertical structure…' -> 'Little change in the vertical structure…'

Line 525 – '…prior in…' -> '…prior to…'

---

## Author Response (AR2)

Reviewer #1

The authors have generally speaking addressed my posted major concerns. There are two minor comments without an author response (corresponding to - l. 287, - l. 372 in the original submission).

Our apologies to the reviewer for missing responses to these issues raised during the first review. As the reviewer is aware, there were four major revisions to respond and incorporate with the submission, and by mistake these two points got lost in the process. In no way did we intentionally neglect them. In fact, both of these suggestions were dealt with, either during the revised submission, or with this second revised submission.

Reviewer 2

Review of "Processes contributing to Arctic cloud dissipation and formation events that bookend clear sky periods" by J. Sedlar, A. Igel, and H. Telg.

Submitted manuscript version 2. Ian M. Brooks

Overview

This revised manuscript is a significant improvement on the original. The major issues raised in my original review have been addressed. The result remain somewhat inconclusive, but nevertheless the extensive documentation of cloud, aerosol, thermodynamic, and large scale dynamic conditions are a useful contribution to the field.

I recommend that the manuscript is suitable for publication after minor revision. Detailed comments to be addressed are noted below.

Once more, we wish to thank Reviewer #2 for their careful consideration of our manuscript. The reviewer is still concerned with inconclusive results. This study has opened the door, and left it open, for the community to either lend support to our hypotheses and result that support them, or to nullify them and propose alternate hypotheses; the essence of scientific process.

We have considered the reviewer's minor suggestions and incorporated those where appropriate. Responses to the reviewer's comments are listed in red.

**Detailed comments**

Line 11 – "A suite of remote sensing and in situ instrumentation from the high-latitude observatory are analysed;…" -> "Measurements from a suite of…are analysed;…" – the measurements are analysed not the instruments.

Updated as suggested.

Line 14-15 – "the clear period bookends" – 'bookends' here is a rather casual, and not entirely clear, term. Maybe rephrase to something like '…aerosol….is relatively invariant during the periods bookending clear sky conditions'

Changed as suggested.

Line 20 – "aerosol particles concentrations changed by a factor" – a factor of what? Need a value (and sign) of the change here

Added that concentrations changed by a factor of two around summer formation events.

Line 40 – "effective infrared cooling from the surface results in near-surface temperatures to drop" – grammar, 'to drop' doesn't fit with the rest of this statement -> "effective infrared cooling from the surface results in near-surface temperatures decreasing"

Changed as suggested.

Line 108 – "tropospheric clouds were common" – tense doesn't match first part of sentence -> "tropospheric clouds are common"

Changed as suggested.

Line 137 - "although some concentrations may" – a very vague statement, need more detail. 'some' concentrations...high, low, variable but under some particular conditions?

This statement has been revised to reflect "low concentrations of small droplet sizes."

Line 141 – "its measurement is sensitive volume squared" – grammar – "its measurement is sensitive to particle volume squared"

Changed as suggested.

**Methods**

Line 172 – "...condition was not met, the clear period was discarded..." – suggest changing wording to "...condition was not met, the period was discarded...", if the period is discarded because of intermittent cloud then it's not really a 'clear' period for the purposes of this study.

Thank you for catching this ambiguity. Changed as suggested.

Line 228-238, discussion of figure 3 – the processes mentioned as possible causes of the drop in aerosol backscatter between BL and overlying air are all reasonable. An additional factor may be the typical decrease in humidity across BL top. For hygroscopic aerosol, particle size can change significantly with relative humidity (ballpark values are a doubling between 'dry' and 80% HR, and another doubling between 80% and ~100% RH for particles such as sea salt), this might lead to a drop in backscatter across BL top even for an aerosol population that was uniform in concentration and dry radius across the inversion. This is, of course, highly dependent on aerosol chemistry, and change in RH across BL top, and not quantifiable here, but worth keeping in mind.

The reviewer raises a very valid process that may influence the change in backscatter between near surface boundary layer sources and free troposphere. We agree with the reviewer, and have added the following as a potential process of importance:

…"and variability in the relative humidity profile."

Figure 4 – some of the panels show colours (at high backscatter) outside the range indicated on the colour bar.

This is true and is a caveat of holding the color bar limits the same for each seasonal subpanel. By increasing the upper limit, the details in the distributions would be less obvious for DJF and MAM. On the other hand, JJA and SON are dominated by fewer cases and as such the profiles of the PDFs reveal those peaks in backscatter outside the color bar range.

Line 347 – "...these clouds often modulate the stratification due to cloud top radiative cooling and induced turbulence..."

i) This phrasing is ambiguous – not clear if the meaning is that the stratification itself, or the modulation of the stratification, is due to cloud-top radiative cooling,
ii) The stratification referred to (or implied by the preceding statement) is the 'static stability near the surface' – I'm not sure that cloud-top radiative cooling and associated turbulent mixing impacts strongly (or in some cases at all) on the near surface stratification. That is much more strongly influenced by the simple presence of cloud and whether the surface itself is cooling radiatively (clear skies) or not (cloudy skies). Cloud driven turbulence will certainly impact BL thermodynamic structure as a whole, and might extend to the near-surface layer, but is only one of several factors affecting surface stability.

After re-reading the original passage, we agree with the reviewer's concern. To address these valid arguments, we have revised the sentence as:

"Arctic stratocumulus clouds exert a critical influence on the stratification of the lower Arctic atmosphere via their significant greenhouse effect (longwave forcing at the surface) and cloud-generated turbulent mixing.."

Line 354 – '950 hPa level is generally around 500 m AGL in the Arctic, which frequently encompasses all, or a fraction of, the Arctic atmospheric boundary layer and the sub-cloud mixed layer' – rather loose and partly redundant phrasing. The lowest 500m must always encompass at least part of the BL. It will often encompass at least part of the sub-cloud mixed layer – though since your focus here is on cases where cloud base is <= 400m, it must also always encompass the sub-cloud layer for all cases considered here.

This statement has been revised as follows:

"…frequently encompasses all, or a large fraction, of the low cloud driven mixed layer…"

Line 485 – 'Little changes in the vertical structure...' -> 'Little change in the vertical structure...'

Changed as suggested.

Line 525 – '...prior in...' -> '...prior to...'

Changed as suggested.

**Reviewer 3**

**Suggestions for revision or reasons for rejection (will be published if the paper is accepted for final publication)**

This is a clearly improved version of a study where the authors attempt to explore reasons for dissipation and formation of low clouds in the Arctic, using a multitude of data from the ARM site in Utqiagvik (Barrow).

The framing and organization of the study is much improved and has less of a "helpless searching" character; they go in with a hypotheses and invalidates most of them; that's clearly useful results. I'm basically fine with the revision, I only have a few minor points below that the authors can use if they wish to clarify/improve the text more.

Once more, we wish to thank Reviewer #3 for their careful consideration of our manuscript. We have considered the reviewer's minor suggestions and incorporated those where appropriate. Responses to the reviewer's comments are listed in red.

Line 10: Drop "relatively"; it is "limited", full stop.

Removed as suggested.

Lines 31-33: Valid only if albedo is also high enough.

This is true, so we have added "reflective sea ice" to document this important point.

Line 60: No atmosphere anywhere is very stationary; the Arctic atmosphere is no exception.

Absolute true. We have reworded this line to highlight that the atmosphere is not stationary and synoptic forcing, etc., is ongoing during all seasons across the Arctic.

Line 76: Unclear use of "transition of cloud lifecycle". First, a "cycle" implies "transition", so what is a transition of the cycle? Second, does this (= low CCN count) apply also on cloud formation?

To address the first point, the text has been modified as follows:

"…are an efficient mechanism in initiating cloud dissipation…"

For the second point, one could hypothesize that low number concentrations based on the Mauritsen et al. (2011) study which documented a situation where the low particle concentrations likely contributed to cloud dissipation; and since the concentrations remained low, cloud formation was likely suppressed, even in a supersaturated environment. However, this text in the paper is describing results of modeling studies of cloud dissipation processes so we do not wish to speculate on formation inhibition; that is the scientific hyptotheses/basis to test within our study.

Line 142: "… attenuated in by the presence …"?

Updated as follows:

"…the signal may be attenuated by ice crystals…"

Line 140: Unclear formulation in "Near-surface measurements … where observed …" I submit you either "used" near-surface measurements or "observed" near-surface meteorological variables.

The phrase has been updated to "Near-surface measurements…. were made from…"

Lines 162-174: I can think of cases, especially close to a coastline, where there would be substantial but partial cloudiness for extended periods of time while the nadir-pointing instruments would either indicate completely clear or completely cloudy conditions.

Unfortunately, this is the trade-off between spatial coverage and detailed vertical sampling. While it may be the case that variable cloudiness occurs outside the field of view of zenith-viewing instruments, the NSA is well documented as a cloudy environment, which agrees well with observations of cloud occurrence/persistence over the Arctic sea ice.

Line 170: Why 96%? Why not 95% – or 90%?

Since we require at least a 2-hr period of clear sky, 96% corresponds to approximately 115 min of clear sky; meaning we allow for 5-min of intermittent cloudiness or scattered cloudiness within the lowest allowable period (2 hrs). This could have been decreased to only 90%; however, this would inevitable allow longer periods of intermittent cloudiness to be considered within an identified clear sky period.

Lines 198-199: Later you find little evidence that aerosols change much at either "bookend", but for this example there seems to be a large almost "hard-top" change; a little confusing as one goes into the story with this image imprinted on the retina. Maybe this was an odd case; maybe you should drop the CPC results in this figure.

We find the results in Fig 1e to be consistent with the changes reported in CPC concentrations around dissipation/formation shown in Fig. 6. While not consistent across all seasons, the aerosol concentration changes are quite large during events in spring and autumn, and especially large during summer. This is one of the major findings of the paper, and as such we do not think the example in Fig. 1 is misleading or is it an odd case.

Lines 204-205: Not an English expert, but I associate "vast" with space as in a large area; here it is used with "persistence" which is temporal. Is there such a thing as a "vast persistence"?

Vast has been removed from the sentence.

Line 211: Pretty obvious that it would, don't you think?

We believe it is important to relate the periods of clear skies with overall monthly cloudiness. It is not entirely trivial to think that abbreviated clear sky periods would not be more frequent during months with high cloud occurrence. Or vice versa, months with less cloudiness may lead infrequent but longer individual periods of clear skies.

Figure 3: It looks to me that adjacent months are quite similar within seasons. Given the few cases (September & October only six) I wonder why you choose to show this monthly and not by season. Also wonder how representative some of these profiles are.

These profile statistics could have been combined into seasonal plots, but we believe it is important to see where the transition height between lower troposphere mixed layer aerosol backscatter and free troposphere occur each month.

Lines 293-294: Correct me if I'm wrong, but up tpo this point the discussion is general, including both "bookends", so the "cloud lifecycle changes" include both dissipation and formation? Then does the concentration drop across both. Also see my earlier comment about changes (or transitions) in life cycles; the cycle implies a change, so what is a change in the cycle?

SThe reviewer is not incorrect; the paper looks at changes occurring both around dissipation and formation. Figure 6 shows the concentrations both before and after dissipation (a-d) and formation (e-h). As described above, the changes depend on the month, with summer showing the largest change in aerosol concentrations, especially for the low cloud and fog events.

Line 298: Here it seems to shift from general to specific; dissipation. Maybe mark this by a new paragraph.

New paragraph has been added, as suggested.

Line 329: Confusing: "… are not limited to Aitken but include larger particle sizes that can activate." Do you mean smaller?

Correct. We have added the word "smaller" before Aitken for emphasis.

Line 338: Using "… imply that … may not …" indicates unnecessary uncertainty. Stick to only "imply"; that is sufficient.

Changed as suggested.

Line 354: I would use "a large fraction of". With the surface being one anchor point and the other being ~500 m, while "a fraction of" could be interpreted as even only 1% of the whole, the sentence becomes confusing.

Changed as suggested.

Line 367: "… clear-sky LWN and LTS modes …" sounds like these two variables had two separate modes, rather than that the combined pdf of both has different modes. Suggest using "LWN/LTS modes".

Changed as suggested.

Line 415 and onward: The wind direction by itself is a blunt instrument; one can have a small change in a generally southerly wind, say from 165 to 195 degrees, as a cold front passes moving east to west, but still with a substantial change in air mass. The latter may be much more significant than for example changes from 180 to 360 degrees on either side of a passing high-pressure ridge within the same air mass.

The reviewer is correct. However, we believe the results shown for the differences, or lack thereof, in wind around dissipation/formation events is consistent with the humidity and synoptic tendency results, and therefore we kept the text in its current form.

Reviewer 4

**Suggestions for revision or reasons for rejection (will be published if the paper is accepted for final publication)**

I would like the thank the authors for their effort in replying to and considering all comments they have received. This is an interesting study and the manuscript now reads very well. I recommend publication and have just a few minor comments:

We thank the reviewer for once again considering our manuscript for publication. We are happy to hear the reviewer has responding positively our revisions, which were a result of thorough suggestions/comments by this, and 3 other, external reviewers. Our responses are provided below in red.

• Regarding the detection limit of the CPC, in my previous comment (general comment #3 by reviewer 4), I was wondering about the lower detection limit of the CPC and not the higher.

We apologize for the confusion in our response to this comment during the first review. According to the specifications listed in the instrument manual, the lower detection limit is $1 \times 10^{-4}$ cm$^{-3}$. The concentrations observed in our study (e.g. Fig. 6) are orders of magnitude larger than this detection limit.

• Line 537: I would suggest adding "and subsequent wet scavenging/deposition" after "… result of aerosol activation". If there was no wet deposition, then the aerosol number concentration should not change a lot (if both cloudy and clear air is sampled by the CPC). The decrease in aerosol concentration suggests that the occurrence of drizzle, which is interesting in itself.

We do not wish to speculate on drizzle; however coalescence/scavenging is likely and ongoing process. We added "and/or coalescence/scavenging"

• Line 547-548: Please modify the sentence "… onset of clear sky periods, and subsequently the end of clear periods…".

We have modified the text to clarify this statement.

• Line 555: It is not completely clear what the author means with "at the same time" here. I would suggest merging this paragraph with the previous as it would also make it clearer that the increase in aerosol concentrations occurs during the clear periods.

We have clarified this statement by removing "At the same time" and replacing with "In summer".